# Host association and intracellularity evolved multiple times independently in the *Rickettsiales*

Michele Castelli [1], Tiago Nardi [1], Leandro Gammuto[2], Greta Bellinzona [1], Elena Sabaneyeva [3], Alexey Potekhin [4,5], Valentina Serra [2], Giulio Petroni [2,7] ✉ & Davide Sassera [1,6,7] ✉

The order *Rickettsiales* (*Alphaproteobacteria*) encompasses multiple diverse lineages of host-associated bacteria, including pathogens, reproductive manipulators, and mutualists. Here, in order to understand how intracellularity and host association originated in this order, and whether they are ancestral or convergently evolved characteristics, we built a large and phylogenetically-balanced dataset that includes de novo sequenced genomes and a selection of published genomic and metagenomic assemblies. We perform detailed functional reconstructions that clearly indicates "late" and parallel evolution of obligate host-association in different *Rickettsiales* lineages. According to the depicted scenario, multiple independent horizontal acquisitions of transporters led to the progressive loss of biosynthesis of nucleotides, amino acids and other metabolites, producing distinct conditions of host-dependence. Each clade experienced a different pattern of evolution of the ancestral arsenal of interaction apparatuses, including development of specialised effectors involved in the lineage-specific mechanisms of host cell adhesion and/or invasion.

*Rickettsiales* are an early diverging[1] and ancient[2] alphaproteobacterial order. All the experimentally characterised members of this group engage in obligate associations with eukaryotic host cells[3]. The most long-term and thoroughly studied *Rickettsiales* include vector-borne pathogens, e.g., *Rickettsia* and *Anaplasma*, causing various diseases in humans and vertebrates[4–6], as well as *Wolbachia*, that can establish complex interactions with arthropod and nematode hosts[7], chiefly reproductive manipulation and mutualism.

In recent years, our knowledge and understanding of the origin, evolution and diversification of *Rickettsiales*, as well as of the diversity of hosts and of interaction modes, have been remarkably improved. We can identify in particular three main advances. The first is the finding of a plethora of novel lineages (over 30 total genera described,

grouped into seven families[8–11]), living in association with a wide variety of hosts[12–14]. Most of those hosts are diverse aquatic unicellular eukaryotes (e.g., ciliates, amoebae, algae)[1,15–22], which have been deemed as probable ancestral hosts[2,23,24], even though these associations are still poorly investigated.

Second, "*Candidatus* Deianiraea vastatrix" (from now on, *Candidatus* will be omitted from taxonomic names, e.g., *Deianiraea vastatrix*), a fully extracellular *Rickettsiales* bacterium equipped with an unexpectedly large biosynthetic repertoire for amino acids, was discovered[8], opening a new perspective on the evolution of *Rickettsiales*. While previous views implied that obligate intracellular association dated back to the last common ancestor of the order ("intracellularity early" hypothesis), this discovery cast doubt on those.

---

[1]Department of Biology and Biotechnology, University of Pavia, Pavia, Italy. [2]Department of Biology, University of Pisa, Pisa, Italy. [3]Department of Cytology and Histology, Saint Petersburg State University, Petersburg, Russia. [4]Department of Microbiology, Saint Petersburg State University, Petersburg, Russia. [5]Research Department for Limnology, University of Innsbruck, Mondsee, Austria. [6]IRCCS Policlinico San Matteo, Pavia, Italy. [7]These authors contributed equally: Giulio Petroni, Davide Sassera. ✉e-mail: giulio.petroni@unipi.it; davide.sassera@unipv.it

Accordingly, another plausible scenario can be envisioned: obligate intracellularity could have evolved later and multiple times independently in different sublineages ("intracellularity late" hypothesis), together with a stronger dependence on host cells.

Third, metagenome binning recently allowed the discovery of further *Rickettsiales* sublineages, in particular two early-diverging families[10]. Their genetic repertoire (including nutrient uptake, detoxification, and multiple biosynthetic pathways) led the authors to hypothesise that these bacteria could be free-living, implying that adaptation to the interaction with host cells might have occurred in more "derived" *Rickettsiales* lineages.

However, many open points still exist on the origin and evolution of the interactions between *Rickettsiales* and their hosts. In particular, major aspects are whether the process(es) of transition towards obligate association and towards obligate intracellularity were one single or distinct phenomena, and whether each of them occurred "early" or "late". In order to address such salient questions, in this study we collected a large and representative genomic dataset of *Rickettsiales*, thanks to de novo sequencing and selection of metagenomic sequences, thus identifying three novel families and remarkably extending the available diversity within previously known lineages. This allowed us to get a view on the diversity and evolution of host adaptation among *Rickettsiales*, finding multiple and convincing lines of evidence supporting the intracellularity late hypothesis.

## Results

### Novel genomes
In this work, we obtained nine complete genome sequences of *Rickettsiales* bacteria (belonging to nine species, eight genera, two families). These represent the first sequences for the respective species, with the exception of *Megaira polyxenophila*[13]. Thus, the evolutionary representativeness of available *Rickettsiales* genomes was significantly improved, also considering that all the newly sequenced organisms are hosted by ciliates or other protists. All the assemblies were highly curated, resulting in most cases in a very high quality (four genomes fully closed, five in total having L50 = 1, and one L50 = 2) (Supplementary Data 1), as confirmed by the comparison of BUSCO scores with typical ranges in *Rickettsiales* (Supplementary Data 2). In three cases, the quality of the assembly allowed to clearly determine the presence of plasmids (respectively in two *Rickettsiaceae* and one among *Midichloriaceae*, Supplementary Data 1).

### Phylogeny
For the successive analyses, we aimed to capture and analyse the widest available diversity of *Rickettsiales* from published sequences, including under-explored (e.g., *Deianiraeaceae*) and possibly yet uncharacterised lineages. To do so, together with a representative set of 36 available *Rickettsiales* genomes, we selected a total of "high-quality" 314 potential *Rickettsiales* metagenome-assembled-genomes (MAGs) from various sources. Their affiliation to *Rickettsiales* was tested by a multi-step phylogeny-based approach. To this purpose, for each MAG lineage, a customised organism selection was employed, and a site-selection approach was applied to counterbalance compositional heterogeneity[1]. After filtering out phylogenetically redundant MAGs, we identified 68 *Rickettsiales* MAGs, ending up with 113 total *Rickettsiales* for the final analysis, including the nine novel high-quality genomes.

For this final dataset, inferences were performed on the "untreated" concatenated alignment and after applying compositional trimming (Fig. 1; Supplementary Fig. 18). As expected, the most significant differences among those trees pertained the placement of two alphaproteobacterial lineages, namely *Pelagibacterales* and *Holosporales*, which are well-known to be involved in artefacts due to compositional heterogeneity (e.g.,[1,25,26]).

For what concerns the *Rickettsiales*, in the "untreated" dataset the resulting topology was quite robust, with most nodes finding full support, and only five nodes below the ultrafast bootstrap threshold of 90% (Fig. 1; Supplementary Fig. 18). Specifically, the four families that include at least one characterised organism were highly supported, and mostly with a significantly increased representativeness, thanks to novel genomes sequenced and MAGs identified in this study (novel taxa: 14/29 *Rickettsiaceae*; 10/17 *Midichloriaceae*; 1/14 *Anaplasmataceae*; 5/6 *Deianiraeaceae*). The inner relationships within *Rickettsiaceae* and *Anaplasmataceae* are overall consistent with previous phylogenetic and phylogenomic studies[8,10,12,15,16,27]. For *Midichloriaceae*, it was possible to infer a novel inner topology with a much higher support with respect to previous studies[14,19,20,27]. Furthermore, it was possible to determine that the *Midichloriaceae* bacterium associated with *Plagiopyla* represents a novel genus and species (Fig. 1; from now on, *Vederia obscura*, see Supplementary Note 1 for taxonomic description).

The three recently described families composed only by MAGs[10], namely *Gamibacteraceae*, *Athabascaceae*, and *Mitibacteraceae*, were retrieved with a comparatively higher representativeness (Fig. 1). Besides, three further MAG-only families were identified for the first time in this study: (i) *Diomedesiaceae*, (ii) *Jistubacteraceae*, and (iii) *Arkhamiaceae*, the first being sister group of *Deianiraeaceae*+*Gamibacteraceae* (we will name these three families together as the "DDG"-clade), while the latter two forming a sequential branching pattern between *Athabascaceae* and *Mitibacteraceae* (Fig. 1; Supplementary Fig. 18, see Supplementary Note 1 for taxonomic descriptions).

The phyletic relationships among the families of *Rickettsiales* in our "untreated" dataset are in general consistent with the most recent and comprehensive studies[8,10,18], with a single partial exception. The "untreated" dataset indicates a closer relationship between the DDG-clade and *Anaplasmataceae* with respect to *Midichloriaceae*, consistent with 16S rRNA gene phylogenies and some phylogenomic analyses[8,18], while another study placed *Anaplasmataceae* and *Midichloriaceae* as sister groups[10]. Our inferences on the the most pronounced compositionally trimmed datasets (i.e., ≥30% most heterogeneous sites removed) also produced a sister group relationship between *Anaplasmataceae* and *Midichloriaceae*, but with low supports (always below 90% ultrafast bootstrap; Supplementary Fig. 18), thus not being themselves fully conclusive. In the study by Schön and co-authors[10], such alternative topology was obtained as well only after compositional trimming, and found maximal support only with Bayesian inference analyses. Therefore, compositional trimming and inference methods could be major reasons behind those different reconstructions, which may also have been potentially significantly influenced by other factors, such as the progressive and large increase of available members of the DDG-clade in successive studies (in particular, this is the first study considering *Diomedesiaceae*). Overall, it seems that the relationships between *Anaplasmataceae*, *Midichloriaceae*, and DDG-clade are not yet fully resolved, and may require further in-depth investigations. Here, we opted for the topology obtained without compositional trimming as "reference" for the following analyses (Fig. 1), considering it preferable to prevent the loss of phylogenetic signal which is an inevitable consequence of trimming. We posit that the scenario we present below for reconstructing the evolution of *Rickettsiales* is not affected by those, relatively minor, differences in topology.

In terms of origin, the vast majority of samples derived from aquatic environments (both freshwater and marine), especially in deeper nodes of the tree (Fig. 1). The same is true within each family, with the significant exceptions of *Anaplasmataceae* and *Deianiraeaceae*, mostly derived from terrestrial environments. Many characterised hosts resulted to be protists (most abundant in all families, except for *Anaplasmataceae*, all hosted by metazoans). For all MAGs, including in particular deeply branching lineages, no conclusive information is available on potential hosts, while in few cases there is loose indication of association, e.g., as originating from rumen[28,29].

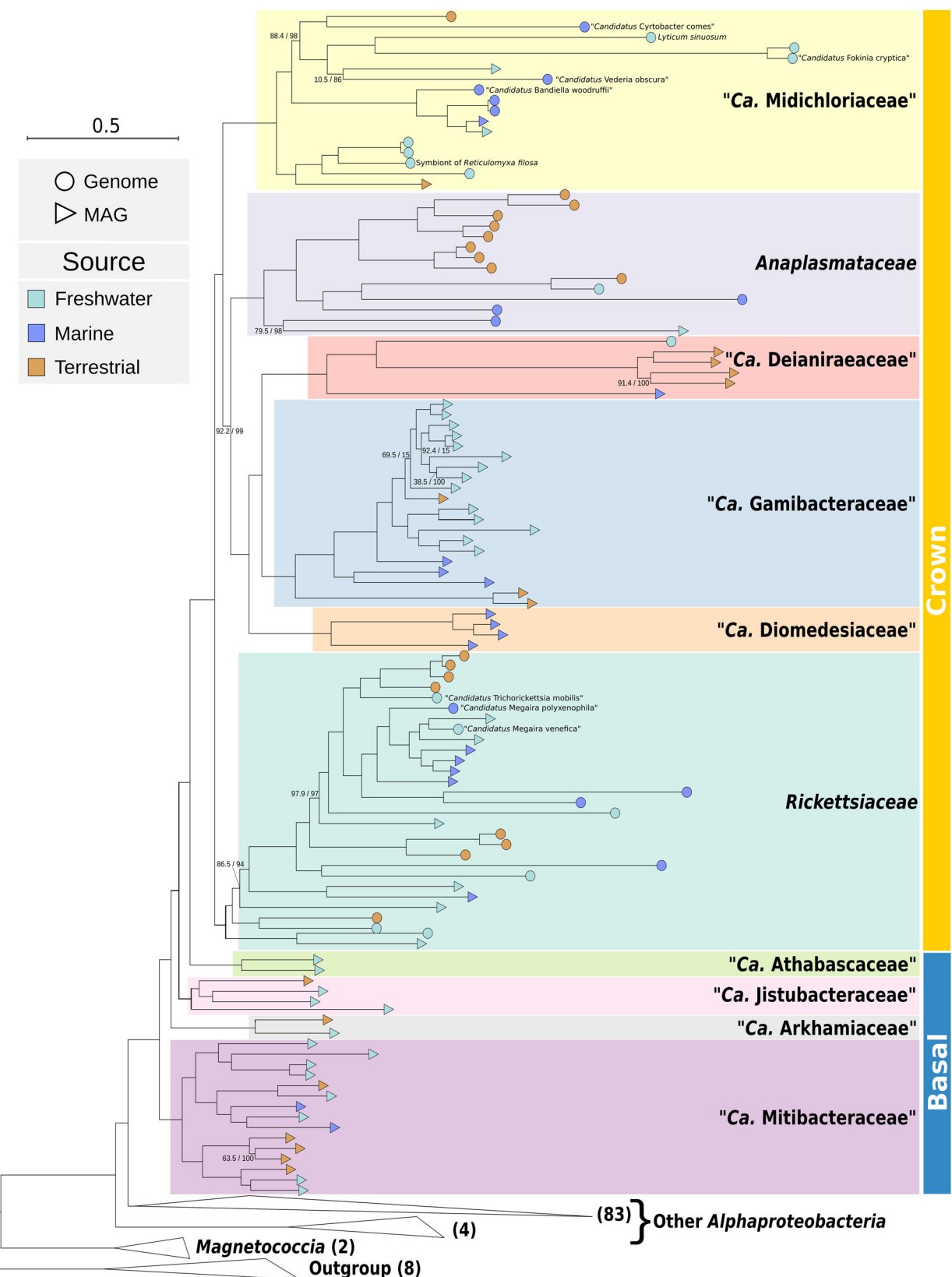

## General genome comparisons

Based on phylogeny, we hereby define as "crown *Rickettsiales*", the smallest monophylum that comprises all characterised organisms (i.e., the one including the six families *Rickettsiaceae*, *Midichloriaceae*, *Anaplasmataceae*, *Deianiraeaceae*, *Gamibacteraceae*, *Diomedesiaceae*), thus corresponding to the classical *Rickettsiales* as defined

previously[10]. Conversely, the four other early diverging families will be defined as "basal *Rickettsiales*".

Genome sizes are quite variable between and within *Rickettsiales* families (Supplementary Data 3). Crown *Rickettsiales* genomes are mostly in the range 1–1.5 Mb, with some appreciable differences between families, in particular on average *Anaplasmataceae* (1.1 Mb)

**Fig. 1 | Phylogenomic tree of *Rickettsiales*.** Maximum likelihood phylogenomic tree of 113 *Rickettsiales* inferred on 179 concatenated orthologs. Each *Rickettsiales* family is highlighted by a differently coloured box. At tips, round shapes indicate genome assemblies, while triangular shapes indicate metagenome-assembled-genomes (MAGs). Shape fillings show the sample source, namely light blue for freshwater, dark blue for marine, and orange for terrestrial. Due to space constraints, only the names of the nine newly obtained genome assemblies were reported, and non-*Rickettsiales* lineages (including other *Alphaproteobacteria*, *Magnetococcia*, and outgroup) are represented by collapsed triangular shapes, with the respective number of organisms reported (full tree is shown in Supplementary Fig. 18). On each branch, support values by SH-aLRT with 1000 replicates and by 1000 ultrafast bootstraps are reported (full support values were omitted). Bars on the right-hand side indicate the crown and "basal" *Rickettsiales*, respectively. The tree scale stands for estimated sequence divergence. *Ca.* is an abbreviation for *Candidatus*.

and *Deianiraeaeaceae* (1.0 Mb), are smaller than others (all averages ≥1.3 Mb). Genomes from basal families are much larger (frequently >2 Mb, and on average ≥1.8 Mb, except *Arkhamiaceae*). It should be taken into account that the probable incompleteness of some MAGs (Supplementary Data 2) may influence the average size estimates, especially in DDG-clade and basal families. GC content is in general inversely correlated with genome size (32–36% on average in classical families and 40–52% in basal ones, with a maximum of 61%, in *Mitibacteraceae*).

As expected, gene numbers are consistent with respective genome sizes (Supplementary Data 3). We aimed to analyse the gene content and its variation along the *Rickettsiales* phylogeny. For this purpose, we reconstructed "homology groups" from the annotated genes and manually inspected the copy numbers in order to get insight into their evolutionary patterns (see Methods for details on the construction of "homology groups" and the differences with respect to the original eggNOG orthogroups). Our analyses supported the notions from previous studies that *Rickettsiales* have globally experienced genome reduction trends (Supplementary Fig. 19), as a putative consequence of adaptation and specialisation to host-associated lifestyles[3,8,10,30,31].

In order to investigate the origin and evolution of such interactions with host cells from a functional and metabolic perspective, we selected a number of relevant traits/functions, by inspecting the global patterns of "homology groups"(Supplementary Fig. 19). The selection included in particular biosynthesis and uptake of metabolic precursors (i.e., amino acids and nucleotides), secretion/adhesion/motility apparatuses and putative effector molecules, which were addressed by further dedicated analyses (see below). A detailed overview of gene content variation and evolution in *Rickettsiales*, with a special focus on general "family-level" trends, as well as on the single newly characterised genomes, is presented in (Supplementary Note 2). Such an approach also allowed to prevent misleading conclusions due to incompleteness of single MAGs (Supplementary Data 2).

**Secretion, attachment, and motility**

Secretion systems are among the components that may exert a central role in regulating interactions with host cells[32], potentially ensuring specific stages of the bacterial life cycle through the delivery of effectors. In *Rickettsiales*, the hallmark apparatus is type IV secretion system (Supplementary Fig. 20), representing a probable ancestral horizontal acquisition[33], and a possible prerequisite for establishing interactions with host cells[10]. Only few genomes among *Rickettsiales* are devoid of this apparatus[16,34], and few others display an incomplete gene set, possibly indicative of ongoing loss, in particular *Bandiella* and the *Midichloriaceae* symbiont of *Reticulomyxa* (Supplementary Fig. 20). Type VI secretion system is very rare, being found only in two *Rickettsiaceae* (Supplementary Fig. 20), in both cases encoded on plasmids, a probable indication of horizontal acquisition. In *Sneabacter* this system has possibly functionally replaced the type IV secretion system[16], while in *Trichorickettsia* both systems coexist.

Among putative secreted effector proteins (Supplementary Fig. 21), the "repeat-bearing" ones (ankyrin, tetratricopeptide, leucine-rich or pentapeptide repeats) are overall abundant and enriched in crown families, with many lineage-specific patterns, but are also present in basal families. On the other hand, RTX toxins[35,36] are quite abundant in basal families, and uncommon in crown ones. Interestingly, proteins involved in the intracellular invasion of eukaryotes, such as hemolysins, patatin-like and other phospholipases[37], are common in some crown families such as *Rickettsiaceae* and *Midichloriacae*, and not uncommon in basal families, while they are rare in DDG clade, especially *Deianiraeaceae* (Supplementary Fig. 21). Several other putative toxins/effectors, characterised in *Rickettsiales*[38] and/or in other bacteria[13,39–45], were found more rarely, showing patterns of presence/absence that appear to be quite lineage-specific, and without sharp differences between basal and crown *Rickettsiales* (Supplementary Fig. 21).

The flagellum might be important especially during horizontal transmission in *Rickettsiales*[20,46]. Flagellar genes are common in basal *Rickettsiales* (Supplementary Fig. 22), likely representing ancestral traits[46]. Conversely, they are absent in the DDG-clade, and are very rare in *Anaplasmataceae* (found just in the aquatic *Echinorickettsia* and *Xenolissoclinum*). Within families *Rickettsiaceae* and *Midichloriaceae*, they are extremely rare in terrestrial representatives (the only cases being *Midichloria mitochondrii* and the *Rickettsiaceae* symbiont of *Amblyomma* Ac37b), but quite common in aquatic ones, in particular basal *Rickettsiaceae* and *Midichloriaceae* in general, thereby also confirming the few experimental observations of flagella[47–49]. The poor correlation of the presence of flagellar genes with *Rickettsiales* phylogeny (including differential cases within the same genus, such as *Megaira* and *Midichloria*[34]) would be indicative of multiple independent reduction/loss events (Supplementary Fig. 22). Similar considerations may be true for chemotaxis, which possibly works in conjunction with flagella for host targeting[50], and is present only in basal *Rickettsiales* and in the basal members of the family *Rickettsiaceae* (Supplementary Fig. 22).

Type 4 pili may be involved in the adhesion/attachment to host cells in *Rickettsiales*[8]. Their components are present in basal *Rickettsiales*, in the DDG clade, and only rarely in the other crown families, especially in the respective early-divergent and/or aquatic representatives (Supplementary Fig. 23). Thus, this apparatus was likely ancestral, experiencing multiple independent losses in crown *Rickettsiales*.

Proteins homologous to the FhaBC two-partner secretion system, which were tentatively linked to the attachment and toxicity towards host cells in *Rickettsiales*[8] and may also participate in bacterial competition[51], were likely ancestral and lost multiple times among *Rickettsiales*, being present in the basal families, *Deianiraeaceae*, *Gamibacteraceae*, and very few representatives of the other crown families (Supplementary Fig. 23).

For what concerns proteins characterised to be involved in adherence/invasion of host cells[52,53] or even immune evasion[54] in *Rickettsiales*, they are present (almost) exclusively in subgroups of the respective families (Supplementary Fig. 23).

**Nucleotide and amino acid metabolism**

Nucleotide biosynthesis is stably present (both purines and pyrimidines) in basal *Rickettsiales* (Fig. 2; Supplementary Fig. 24), while, differently from other biosynthetic pathways (Supplementary Note 2), its presence in crown *Rickettsiales* is "scattered" along the organismal phylogeny. Indeed, it is ubiquitous in the families *Gamibacteraeae*, *Diomedesiaceae*, *Anaplasmataceae*, present in the earliest diverging

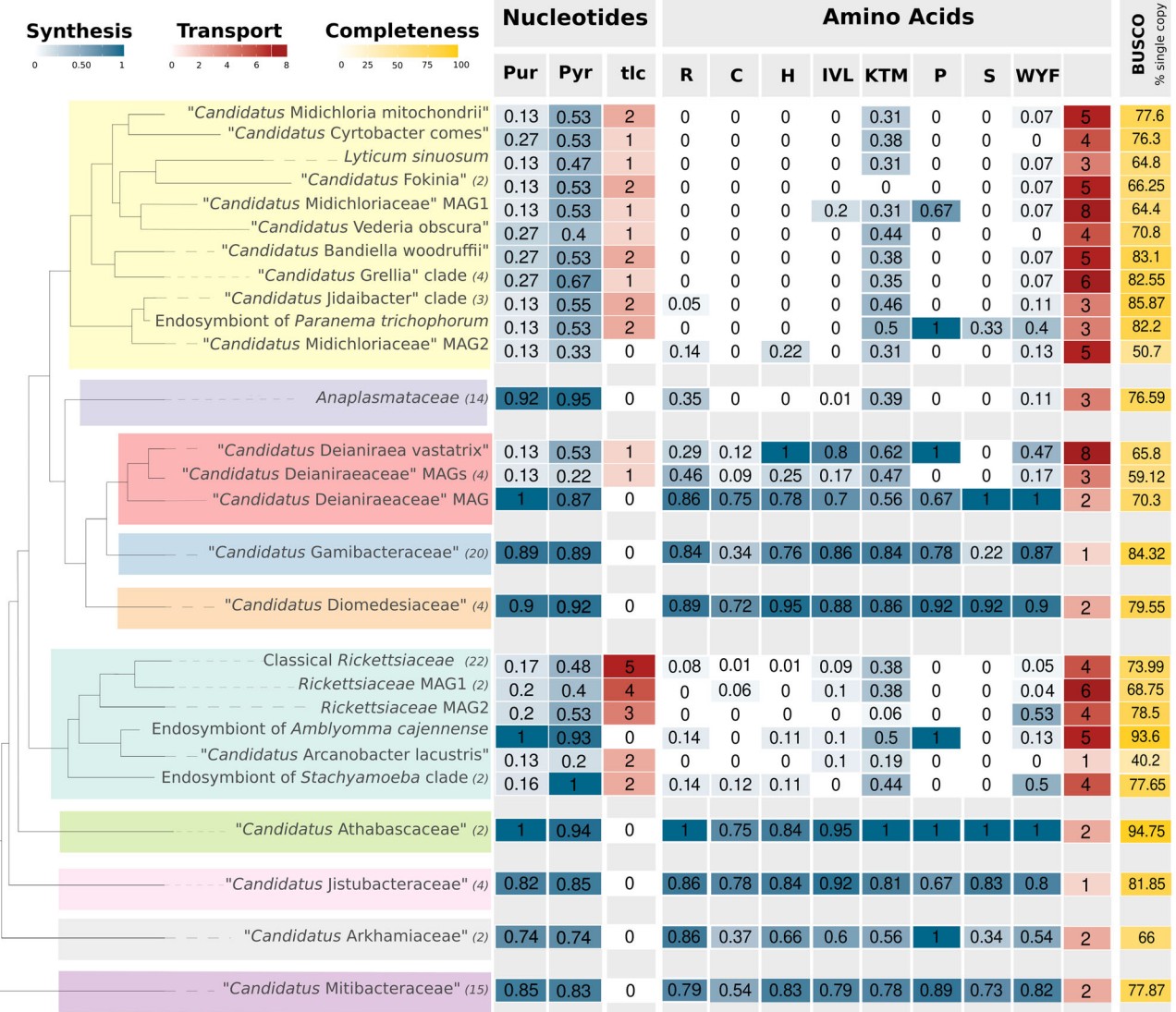

| | Nucleotides | | | Amino Acids | | | | | | | | | BUSCO % single copy |
|---|---|---|---|---|---|---|---|---|---|---|---|---|---|
| | Pur | Pyr | tlc | R | C | H | IVL | KTM | P | S | WYF | | |
| "Candidatus Midichloria mitochondrii" | 0.13 | 0.53 | 2 | 0 | 0 | 0 | 0 | 0.31 | 0 | 0 | 0.07 | 5 | 77.6 |
| "Candidatus Cyrtobacter comes" | 0.27 | 0.53 | 1 | 0 | 0 | 0 | 0 | 0.38 | 0 | 0 | 0 | 4 | 76.3 |
| Lyticum sinuosum | 0.13 | 0.47 | 1 | 0 | 0 | 0 | 0 | 0.31 | 0 | 0 | 0.07 | 3 | 64.8 |
| "Candidatus Fokinia" (2) | 0.13 | 0.53 | 2 | 0 | 0 | 0 | 0 | 0 | 0 | 0 | 0.07 | 5 | 66.25 |
| "Candidatus Midichloriaceae" MAG1 | 0.13 | 0.53 | 1 | 0 | 0 | 0 | 0.2 | 0.31 | 0.67 | 0 | 0.07 | 8 | 64.4 |
| "Candidatus Vederia obscura" | 0.27 | 0.4 | 1 | 0 | 0 | 0 | 0 | 0.44 | 0 | 0 | 0 | 4 | 70.8 |
| "Candidatus Bandiella woodruffii" | 0.27 | 0.53 | 2 | 0 | 0 | 0 | 0 | 0.38 | 0 | 0 | 0.07 | 5 | 83.1 |
| "Candidatus Grellia" clade (4) | 0.27 | 0.67 | 1 | 0 | 0 | 0 | 0 | 0.35 | 0 | 0 | 0.07 | 6 | 82.55 |
| "Candidatus Jidaibacter" clade (3) | 0.13 | 0.55 | 2 | 0.05 | 0 | 0 | 0 | 0.46 | 0 | 0 | 0.11 | 3 | 85.87 |
| Endosymbiont of Paranema trichophorum | 0.13 | 0.53 | 2 | 0 | 0 | 0 | 0 | 0.5 | 1 | 0.33 | 0.4 | 3 | 82.2 |
| "Candidatus Midichloriaceae" MAG2 | 0.13 | 0.33 | 0 | 0.14 | 0 | 0.22 | 0 | 0.31 | 0 | 0 | 0.13 | 5 | 50.7 |
| Anaplasmataceae (14) | 0.92 | 0.95 | 0 | 0.35 | 0 | 0 | 0.01 | 0.39 | 0 | 0 | 0.11 | 3 | 76.59 |
| "Candidatus Deianiraea vastatrix" | 0.13 | 0.53 | 1 | 0.29 | 0.12 | 1 | 0.8 | 0.62 | 1 | 0 | 0.47 | 8 | 65.8 |
| "Candidatus Deianiraeaceae" MAGs (4) | 0.13 | 0.22 | 1 | 0.46 | 0.09 | 0.25 | 0.17 | 0.47 | 0 | 0 | 0.17 | 3 | 59.12 |
| "Candidatus Deianiraeaceae" MAG | 1 | 0.87 | 0 | 0.86 | 0.75 | 0.78 | 0.7 | 0.56 | 0.67 | 1 | 1 | 2 | 70.3 |
| "Candidatus Gamibacteraceae" (20) | 0.89 | 0.89 | 0 | 0.84 | 0.34 | 0.76 | 0.86 | 0.84 | 0.78 | 0.22 | 0.87 | 1 | 84.32 |
| "Candidatus Diomedesiaceae" (4) | 0.9 | 0.92 | 0 | 0.89 | 0.72 | 0.95 | 0.88 | 0.86 | 0.92 | 0.92 | 0.9 | 2 | 79.55 |
| Classical Rickettsiaceae (22) | 0.17 | 0.48 | 5 | 0.08 | 0.01 | 0.01 | 0.09 | 0.38 | 0 | 0 | 0.05 | 4 | 73.99 |
| Rickettsiaceae MAG1 (2) | 0.2 | 0.4 | 4 | 0 | 0.06 | 0 | 0.1 | 0.38 | 0 | 0 | 0.04 | 6 | 68.75 |
| Rickettsiaceae MAG2 | 0.2 | 0.53 | 3 | 0 | 0 | 0 | 0 | 0.06 | 0 | 0 | 0.53 | 4 | 78.5 |
| Endosymbiont of Amblyomma cajennense | 1 | 0.93 | 0 | 0.14 | 0 | 0.11 | 0.1 | 0.5 | 1 | 0 | 0.13 | 5 | 93.6 |
| "Candidatus Arcanobacter lacustris" | 0.13 | 0.2 | 2 | 0 | 0 | 0 | 0.1 | 0.19 | 0 | 0 | 0 | 1 | 40.2 |
| Endosymbiont of Stachyamoeba clade (2) | 0.16 | 1 | 2 | 0.14 | 0.12 | 0.11 | 0 | 0.44 | 0 | 0 | 0.5 | 4 | 77.65 |
| "Candidatus Athabascaceae" (2) | 1 | 0.94 | 0 | 1 | 0.75 | 0.84 | 0.95 | 1 | 1 | 1 | 1 | 2 | 94.75 |
| "Candidatus Jistubacteraceae" (4) | 0.82 | 0.85 | 0 | 0.86 | 0.78 | 0.84 | 0.92 | 0.81 | 0.67 | 0.83 | 0.8 | 1 | 81.85 |
| "Candidatus Arkhamiaceae" (2) | 0.74 | 0.74 | 0 | 0.86 | 0.37 | 0.66 | 0.6 | 0.56 | 1 | 0.34 | 0.54 | 2 | 66 |
| "Candidatus Mitibacteraceae" (15) | 0.85 | 0.83 | 0 | 0.79 | 0.54 | 0.83 | 0.79 | 0.78 | 0.89 | 0.73 | 0.82 | 2 | 77.87 |

**Fig. 2 | Biosynthesis and transport of nucleotides and amino acids.** Heat-map showing the presence and abundance of biosynthetic pathways (blue) of nucleotides (purines and pyrimidines) and amino acids (grouped by to their mutually shared enzymatic steps according to BioCyc[55]), as well as their respective transporters (red). For biosynthesis, the proportion of the total genes of the pathway is shown (Supplementary Data 7), while for transporters, the number of genes is reported, in particular for amino acid transporters the sum of the "characterised hits" (Supplementary Fig. 28). A cladogram of the organisms is shown on the left, with each *Rickettsiales* family highlighted by a differently coloured box. Due to space constraints, selected monophyletic clades with homogeneous gene content were collapsed. For each clade, the number of organisms is shown in brackets (if higher than one), and reported values are averaged, while the complete organism sets are shown in (Supplementary Fig. 24, 28).

*Deianiraeaceae* MAG and in few basal *Rickettsiaceae* (in particular the endosymbiont of *Amblyomma* Ac37b, able to synthesise both purines and pyrimidines), but, besides few genes, absent in *Midichloriaceae* and in the remaining and most numerous *Rickettsiaceae* and *Deianiraeaceae*. Single-gene phylogenies indicate that the respective pathways are likely ancestral in *Rickettsiales*, with quite good correspondence with organismal phylogeny at various levels (up to and even above the families; Supplementary Fig. 25). Few exceptions represented by potential non-orthologs were observed (Supplementary Data 4) and excluded for the phylogenies on the concatenated pathways, which confirmed the same trends observed in single-gene trees (Supplementary Fig. 26). Some relatively minor exceptions to the correspondence with organismal phylogeny (supportive of ancestrality) pertain to the inner relationships among *Rickettsiales* families, namely the placement of the DDG clade (as well as sometimes *Midichloriaceae* or *Anaplasmataceae*) as close relatives of *Rickettsiaceae*, though often not with very high support (Supplementary Fig. 26). This

potentially represents an artefact, due to the fast-evolving rate of the sequences, similarly to the respective organismal phylogeny. In addition, in some cases a few non-*Rickettsiales* sequences (e.g., *Pelagibacterales* and some representatives of *Holosporales* in several concatenated purine biosynthesis trees) were "nested" within *Rickettsiales*, thus representing possible "residual" artefacts even after applying compositional trimming (potentially resulting from long-branch attraction), or, hypothetically, the results of horizontal transfer from *Rickettsiales* to those bacteria.

In the crown *Rickettsiales* lacking nucleotide biosynthesis, this absence is counterbalanced by the ability to obtain final products (or intermediates) from their hosts. Indeed, the presence/absence pattern of tlc nucleotide translocases almost perfectly inversely correlates with nucleotide biosynthesis (Fig. 2; Supplementary Fig. 24), with as noticeable exception a *Midichloriaceae* bacterium MAG (GCA_013288625) that is devoid/depleted in both, but also presents a low BUSCO estimated completeness. This family of transporters

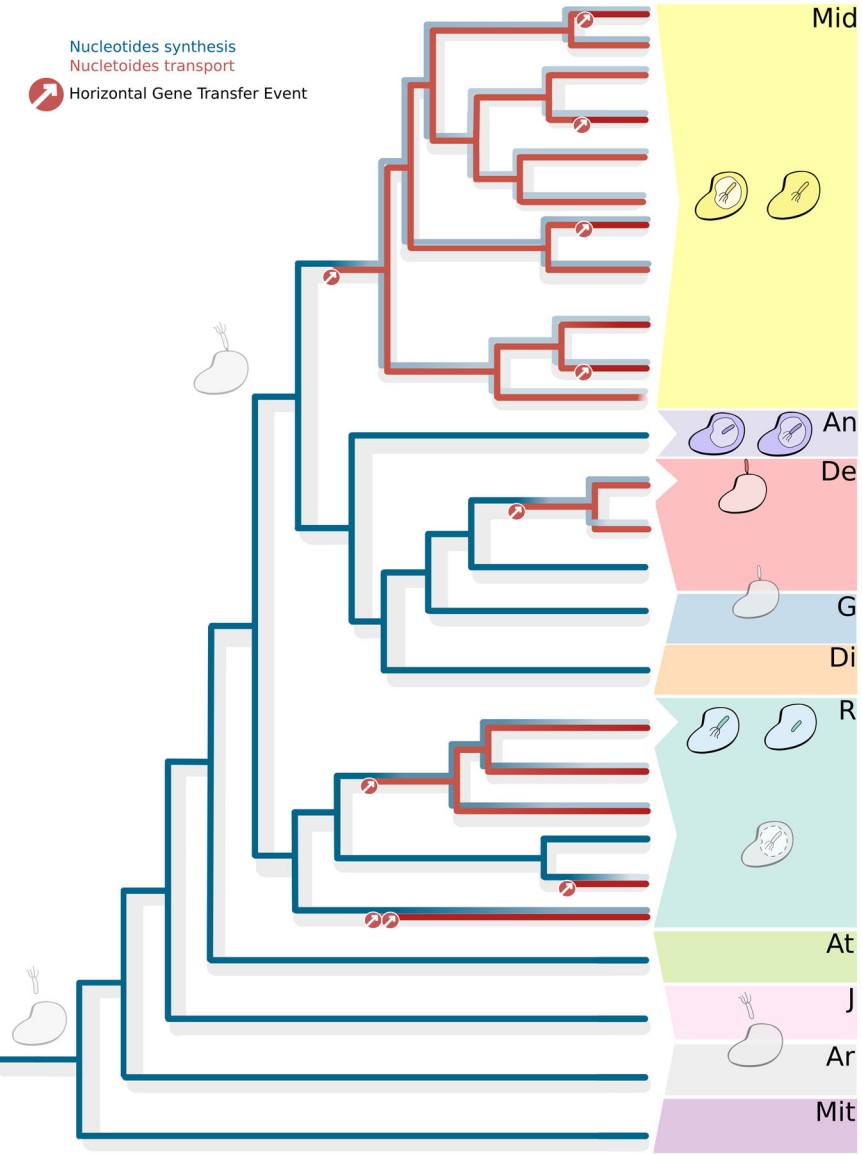

**Fig. 3 | Main steps in the evolution of *Rickettsiales*.** Reconstruction of the main steps of the evolution of *Rickettsiales* with a specific focus on the interactions with and dependence on eukaryotic cells. A cladogram of the main lineages (as represented in Fig. 2) is shown. The case of the biosynthetic pathways (averaged for purines and pyrimidines, blue) and tlc transporters (red) for nucleotides is represented along the tree by a heat-map-like representation, showing the inferred ancestral conditions and hypothesised steps of variation on each branch. In particular, multiple independent acquisitions of transporters by horizontal gene transfer events (red circles with inward arrows) would have led to the progressive reduction/loss of the biosynthesis. *Rickettsiales* families are highlighted by coloured boxes and by abbreviated names (R: *Rickettsiaceae*; Mid: *Midichloriaceae*; An: *Anaplasmataceae*; De: *Deianiraceae*; G: *Gamibacteraceae*; Di: *Diomedesiaceae*; At: *Athabascaceae*; J: *Jistubacteraceae*; Ar: *Arkhamiaceae*; Mit: *Mitibacteraceae*). At tips, groups of tips, and at selected nodes, the drawings represent the known (coloured) or hypothesised (grey) features of the bacteria and their interaction with eukaryotic hosts, in particular, intracellular or extracellular associations, or lack of association, as well as presence/absence of a vacuole and of flagella. Multiple side-by-side pictures represent alternative conditions/reconstructions for the same organisms.

include chloroplast ATP/ADP translocases, as well as a vast array of proteins, able to translocate several different nucleotides[56,57], and previously reported to have experienced multiple horizontal gene transfer events between phylogenetically unrelated host-associated bacteria[58,59].

Comparison of tlc and organismal phylogenies (Supplementary Fig. 27) indicates that these transporters were acquired multiple independent times (up to ten) by different *Rickettsiales* lineages: once among *Deianiraeaceae*, three-five among *Midichloriaceae*, three-four among *Rickettsiaceae* (see Fig. 3). We also detected multiple independent events of duplication leading to several paralogs, namely five copies in classical *Rickettsiaceae*, two-four in three distinct basal *Rickettsiaceae* sublineages, and two copies in the *Jidaibacter* lineage among *Midichloriaceae* (Figs. 2,3; Supplementary Figs. 24, 27).

The presence/absence pattern of biosynthetic pathways for amino acids shows significant analogies with the one we detected for nucleotides (Fig. 2; Supplementary Fig. 28). Indeed, they are quite uniformly present in basal *Rickettsiales*, *Diomedesiaceae*, *Gamibacteraceae*, and partly *Deianiraeceae*. Conversely, they are very rare or fully absent (besides few genes) in the non-monophyletic assemblage of *Rickettsiaceae*, *Midichloriaceae* and *Anaplasmataceae*, with very few exceptions, such as arginine in *Ehrlichia* and *Neoehrlichia* (and partly *Anaplasma*), and proline in the endosymbionts of *Amblyomma* Ac37b and *Peranema* (Supplementary Fig. 28). As in the case of nucleotide synthesis, single-gene phylogenies indicate an overall vertical descent

of these pathways (Supplementary Fig. 25), with few exceptions. The most notable case pertains to cysteine biosynthesis genes, for which basal and crown *Rickettsiales* sequences have different phylogenetic positions, which may suggest recent HGT event(s) in crown *Rickettsiales* or an ancestral paralogy. Among the genes of all the other amino acid biosynthetic pathways, only a limited number of potential non-orthologs among *Rickettsiales* were observed (Supplementary Data 4), which were excluded for the phylogenies on the concatenated pathways (Supplementary Fig. 26). Such phylogenies further corroborated the ancestrality of the biosynthetic pathways for amino acids, with relatively minor differences with respect to the organismal phylogenies, such as the relationships among *Rickettsiales* families. As in the case of nucleotide biosynthesis, these may be residual artefacts after compositional trimming, potentially resulting also from fast-evolving rates of *Rickettsiales* sequences, especially when considering the representatives of the DDG clade.

Besides such potential cases of lineage-specific acquisition of amino acid biosynthesis genes by some representatives of *Rickettsiales* presented above or previously proposed[60], the analyses presented here provide support for "general" vertical inheritance of most amino acids biosynthesis, when present, in most of the *Rickettsiales*, directly from the last common *Rickettsiales* ancestor.

The array of putative amino acids transporters is more complex than that for nucleotides, considering the higher number of amino acids and the multiple independent transporter families with variable substrate specificities, many of which belong to larger gene families of transporters with broader specificity[61,62]. This impairs precise homology-based prediction of substrate specificity in *Rickettsiales*, making it impossible to define with certainty which amino acid is imported by which transporter. The same reasons make an accurate inference of informative gene phylogenies infeasible. Nevertheless, the total number of transporters is much higher in the three families that are more deprived in biosynthesis (Supplementary Fig. 28). This is especially true for those with a higher similarity with ascertained amino acid transporters (Fig. 2). Moreover, as seen for nucleotide transporters, the presence of homologues of different amino acid transporters is scattered along the *Rickettsiales* phylogeny (Supplementary Fig. 28).

We believe that the data presented above clearly indicate a pattern of multiple independent and successive acquisitions of nucleotide and amino acids transporters in different lineages of *Rickettsiales*, events that would have enabled the recipients to efficiently acquire those compounds from their hosts, thus leading to the reduction and eventual loss of the respective biosynthesis, an evolutionary scenario consistent with the intracellularity late hypothesis[8].

## Discussion

The knowledge and understanding of the evolution of the typically host-associated *Rickettsiales*[3,27], in particular its earlier steps, has been hampered by the limited and phylogenetically unbalanced set of genomes available. Here we present a dataset of over one hundred phylogenetically-diverse *Rickettsiales* assemblies, thanks to de novo-sequencing of nine high-quality genomes from underexplored protist-associated representatives, and to an accurate selection of published genomes and MAGs. We thus obtained an enriched representativeness of all known families, including the recently described ones[10], and identified three further ones, for a total of ten families in *Rickettsiales*. Leveraging such an extended taxonomic resolution, we investigated whether the obligate association with eukaryotic hosts and specifically the intracellular lifestyle were "early" conditions with a single origin, or "late" achievements that evolved multiple times independently in different *Rickettsiales* sublineages.

### Evolution of metabolic dependence

It is a generally accepted notion that metabolic dependence on the hosts is a key feature in obligate associations such as those involving multiple independent lineages of bacterial and eukaryotic intracellular parasites[1,63–65], including the *Rickettsiales*[8,10,30,31], that evolved as the consequence of the possibility to efficiently acquire metabolites (including precursors such as amino acids and nucleotides, and, for energy, ATP). Interestingly, such ability is likely due to the acquisition of suitable transporters, making the respective biosynthetic and catabolic pathways dispensable, thus leading to their reduction and eventual loss, concurring in determining a host-dependent condition[63,66]. The pattern of gain of transporters and loss of synthesis pathways can thus be a strong indicator of the state of host dependence through evolution. Based on the herein produced dataset and analyses, the case of nucleotide synthesis and transport is noteworthy among *Rickettsiales*. Indeed, our analysis indicate that the most likely scenario is one of multiple independent horizontal acquisitions (up to ten) of tlc transporters among crown *Rickettsiales*, likely "triggering" independent losses of the ancestral biosynthetic pathways (Figs. 2 and 3; Supplementary Figs. 24, 25, 26, 27). Similar considerations hold for amino acids, even though the impossibility to predict the precise specificity of all transporters impairs a clear reconstruction of single events leading to the multiple independent losses of the ancestral biosynthetic pathways (Figs. 2 and 3, Supplementary Figs. 25, 26, 28).

Besides these more clear-cut cases, detailed analyses of the presence/absence patterns among the genes involved in multiple other pathways (including glycolysis, gluconeogenesis, pentose-phosphate pathway, Krebs cycle and electron transport chain, synthesis of cofactors, lipids, peptidoglycan, lipopolysaccharide, and polyhydroxyalkanoate granules; see Supplementary Note 2) strongly indicate analogous processes of gradual and independent reduction/losses in different crown *Rickettsiales* sublineages. Sharp differences are present even within single families, such as in the metabolically-rich basal *Rickettsiaceae*, as compared to the classical streamlined ones.

Thus, in contrast with the more traditional views[10,20,24], our analyses provide a clear indication that processes of pathway reduction/loss have not taken place just once in *Rickettsiales*, but instead occurred (and are still possibly occurring) multiple times independently in different crown *Rickettsiales* lineages, also in relation with the host features. This hints towards an independent origin of obligate host-associations, and thereby intracellularity, among the *Rickettsiales* (see below section "Evolutionary trajectories of the interactions").

### Evolution of interaction mechanisms

Secretion systems and attachment/invasion molecules are other paramount bacterial components for promoting and actively regulating interactions with host cells[32,33,67]. Interestingly, we found that in crown *Rickettsiales* the repertoire for such systems, as well as for the flagellar apparatus, is a substantial subset of the one of basal *Rickettsiales*. This result fortifies previous notions that the ancestral *Rickettsiales* already possessed a quite rich set of proteins to interact with (unicellular) eukaryotes[10,33]. It seems reasonable to hypothesise that such arsenal could have represented a prerequisite for the establishment of associations with eukaryotic cells, possibly through a partial process of repurposing[10] (e.g., delivery of effectors molecules active on eukaryotes, motility and chemotaxis involved in horizontal transmission). In terms of being instrumental for the evolutionary origin and maintenance of the associations, type IV secretion is a good candidate[10,33], also considering its almost full conservation among *Rickettsiales* (Supplementary Fig. 20). Other apparatuses, e.g., flagellum and type IV pilus/type II secretion, while widespread in "basal" *Rickettsiales*, are more phylogenetically scattered among crown *Rickettsiales* (Supplementary Figs. 22, 23), which is likely a result of multiple independent losses as well. This indicates unique patterns of specialisations along the evolution of *Rickettsiales*, with the concurrent lineage-specific losses of traits that were dispensable for the

interaction with respective host cells. Interestingly, the correlation with phenotypic traits suggests some functional links, such as flagella and chemotaxis in aquatic environments, likely involved in non host-associated stages such as horizontal transmission[20], pili for extra-cellular attachment to host cells in *Deianiraea*[8] and possibly in other members of the DDG clade. At the same time, alternative/additional functions for these apparatuses due to their homologies with secretion systems[68,69] could be possible, and may account for the exceptions to such correlation patterns[34,46].

Conversely, specialisation in the interaction with host cells has likely implied the expansion of other gene families, in particular those of putative secreted effectors, such as the "repeat-containing" ones, or acquisition/development of novel ones. In particular, several proteins characterised in pathogenic *Rickettsiales* (e.g., *Rickettsia*, *Anaplasma*) for their direct involvement in the interaction with the host[38,52–54,70], resulted to be lineage-specific (at the family level or even below) rather than conserved in *Rickettsiales* as a whole (Supplementary Fig. 23). Thus, it seems likely that many other still uncharacterised lineage-specific proteins could exist in the other much less investigated *Rickettsiales* (e.g., *Midichloriaceae*, DDG clade). Such a scenario of lineage-specific sets of "interactors" suggests that the mechanisms and conditions of host-association have evolved independently among different (crown) *Rickettsiales* lineages along with their molecular players.

The absence of experimental data makes it more difficult to precisely infer the condition of basal *Rickettsiales* in terms of potential interactions with eukaryotes. The study discovering the first two basal families (*Mitibacteraceae* and *Athabascaceae*) found that these bacteria are metabolically rich, which is suggestive of independence from possible host cells[10]. Herein, such pattern was fully confirmed in the extended sampling of these two families and in the representatives of the novel ones (*Arkhamiaceae* and *Jistubacteraceae*). At the same time, we found that "basal" *Rickettsiales* bear basically all the putative pre-requisite apparatuses for the interaction. Most significantly, many are also equipped with homologues of effectors typical of crown *Rickettsiales*, such as phospholipases[37] and the "repeat-containing" effectors, and they are even selectively enriched in potential additional effectors[35,36]. Conversely, free-living-like traits such as inorganic nutrient transport and detoxification, previously found only among basal *Rickettsiales*[10], were retrieved also in some crown representatives (Supplementary Note 2). This may be indicative that those crown *Rickettsiales* retain some "primitive" traits, and, more in general, as another hint at more complex evolutionary trajectories than a simple transition towards obligate association at the root of crown *Rickettsiales*.

## Evolutionary trajectories of the interactions

Taken together, the data presented above clearly indicate that obligate host-association was most likely a "late" condition in *Rickettsiales*. Under such a scenario, we propose that at some point in the early *Rickettsiales* evolutionary history their presumably aquatic free-living ancestors were engaged in some kind of facultative interaction with eukaryotes. The starting point could have been defence from protist predators through the release of active effectors, as previously hypothesised[10]. It is possible to envision that such defence mechanisms successively paved the way for the (gradual) development of the (at least occasional) ability to gain further advantages by such interactions. Specifically, they could have become capable of acquiring metabolites from the damaged/killed eukaryotes, somehow reversing and taking control of the predator-prey interactions. The lifestyle of *Deianiraea* could be partly reminiscent of this hypothetical condition[8]. Most likely, such transition towards facultative associations would have occurred prior to the last common ancestor of crown *Rickettsiales*, which are all obligatorily host-associated.

Conversely, it is not straightforward to precisely place an upper bound for such transition, given the complete lack of direct information on the lifestyles of extant basal *Rickettsiales*. It could have occurred sharply in the common ancestor of crown *Rickettsiales*, or could have been more nuanced, involving also the ancestors of some basal lineages, possibly up to the ancestor of all *Rickettsiales*. Nevertheless, it cannot be excluded that any potential host-associated representative of basal *Rickettsiales* could be the result of a convergent and independent evolution with respect to crown *Rickettsiales*.

In any case, for what concerns crown *Rickettsiales*, we can envision that a single initial facultative association would have differentiated in the descendants, becoming tighter and tighter (and at some point, obligate) through parallel successive steps of acquisition/development of metabolite transporters and interactor molecules. Such further transitions would have occurred separately and independently in different *Rickettsiales* lineages, thus supporting the conclusion of a late origin for the obligate association with hosts. The order and kind of such steps, although somehow similar, would have been unique in each of the different crown *Rickettsiales* phyletic lines, as reflected in the present-day lineages, which exhibit differential features of metabolic dependence (e.g., for most amino acids but not nucleotides in *Anaplasmataceae*, and vice versa in most *Deianiraeaceae*), as well as differential mechanisms and conditions of interaction (chiefly intra-cellularity *vs* extracellularity).

Obligate intracellularity is a well-documented condition in most of the characterised crown *Rickettsiales*, namely the non-monophyletic assemblage of *Rickettsiaceae*, *Midichloriaceae*, and *Anaplasmataceae*. Conversely, we previously showed that *Deianiraea*[8], while obligatorily host-associated, is not intracellular, providing the first basis to infer a possible alternative "intracellularity late" hypothesis for *Rickettsiales*. Obligate intracellular bacteria are inherently obligatorily host-associated, and, as such, intracellularity is one among the possible features evolved by bacteria living in obligate host association. Thus, the data and analyses above supporting a "late" obligate host-association similarly represent additional support for the previously proposed[8] "intracellularity late" hypothesis. Accordingly, it seems most probable that obligate intracellularity would have evolved multiple independent times and with differential features only in some of the obligatorily host-associated crown *Rickettsiales*. The members of the DDG clade other than *Deianiraea*, all represented by MAGs, as based in their genome features (e.g., repertoire of metabolic pathways at least equivalent to *Deianiraea*, and comparable set of putative adhesins such as those of the exoprotein family; Supplementary Figs. 23, 24, 25, 26, 28), could be deemed as extracellularily host-associated as well (or even in some cases potentially not obligatorily host-associated), thus being consistent with such a scenario.

Our reconstruction may also provide a novel perspective on the origin and evolution of other host-associated bacterial lineages that, similarly to *Rickettsiales*, present the prerogative to "hold the control" on the interaction and to switch hosts by horizontal transmission, and were thus termed "professional symbionts"[71] (e.g., *Chlamydiae*[72], *Legionellales*[73] and *Holosporales*[1]). These lineages could share similarities in the initial establishment and successive stepwise and "late" evolutionary development.

## Final remarks

From a more general perspective, it seems worth to compare *Rickettsiales* (and possibly other "professional symbionts") with the more "canonical" genome evolution models among obligate symbionts, namely nutritional mutualists in insects (with some parallels also in protists[71]). Such symbionts undergo relatively rapid streamlining as a result of an initial host restriction, followed by a more or less prolonged stasis, and are somehow "doomed" to extinction after replacement. Conversely, "professional symbionts" would be the controllers of the interaction from its evolutionary beginning, retaining the ability to horizontally change hosts, and undergoing much more gradual and "flexible" streamlining processes, depending also on

the external environmental conditions and not just on the features of a single host. We can also observe significant differences in the metabolic interchanges with their hosts, with nutritional symbionts providing metabolites such as amino acids to their hosts, while *Rickettsiales* (and other "professional symbionts") "stealing" the same metabolites from the hosts. Interestingly, the abundance of lineages preferentially associated with marine invertebrates and showing poor co-cladogeneesis with their hosts[74] indicates that there may be more bacteria sharing traits of "professional symbionts" than currently recognised.

Large-scale comparative genomic analyses such as those presented here and elsewhere[10,72,73] have huge potential to provide major advances in our understanding of functional traits and the underlying evolutionary processes. However, they also face inherent predictive limits. In particular, while they are quite suitable for deriving metabolic dependencies, they are not likewise suited for the inference of more complex and not directly documented traits, such as mechanisms of interaction and subcellular (or extracellular) localisation. For example, it could have been burdensome and highly speculative to infer the so far uniquely observed extracellular condition in *Deianiraea*[8] only from its genome. Therefore, considering that we still completely lack any experimental data for six out of ten *Rickettsiales* families (including all basal ones), we strongly invoke the need for further experimental investigations. As in other cases[75], these may provide additional and otherwise unpredictable insights on the lifestyle of present-day organisms, and represent the basis for refining existing hypotheses and inferring novel ones on *Rickettsiales* evolution.

## Methods
### Sample preparation and sequencing
In this work, the nine novel *Rickettsiales* genome sequences were assembled (for a detailed account on sample preparation, sequencing, and genome assembly, see Supplementary Note 3; Supplementary Data 1, 5), starting from eight protist host samples. Six of these samples were characterised in previous studies[47–49,76–78], while two others, namely the ciliates *Plagiopyla frontata* IBS-3 and *Euplotes woodruffi* NDG2, were newly isolated. Each sample was differentially processed. Briefly, for Illumina sequencing most samples were subjected to whole-genome amplification (WGA) with the REPLI-g Single Cell Kit (Qiagen), either directly from few ciliate host cells (four samples: *Paramecium biaurelia* US_Bl 11III1, *Paramecium nephridiatum* Sr 2-6, *E. woodruffi* NDG2, *P. frontata* IBS-3) or from a previously obtained DNA extract (one sample: *Euplotes harpa* BOD18;[76] Supplementary Data 1). Two additional samples (*P. biaurelia* USBL-36I1 and *Paramecium multimicronucleatum* Kr154-4) were processed for bulk DNA extraction (over 200,000 ciliate host cells each), with CTAB and phenol-chloroform protocols, respectively. Each of these seven DNA samples was processed through a Nextera XT library and sequenced by Admera Health (South Plainfield, NJ, USA) on a Illumina HiSeq X machine, producing 2 × 150 bp paired-end reads. Read quality was assessed with FastQC 0.11.7[79]. NDG2 sample was also subjected to Nanopore sequencing. For this purpose, a bulk (~300,000 *Euplotes* cells) extract with the NucleoSpin™ Tissue Kit (Macherey-Nagel™) was processed through a SQK-LSK109 ligation-sequencing library, and sequenced in a FLO-MIN106 flow cell. Basecalling was then performed with guppy 5.0.11. Then, reads were processed with Porechop 0.2.4[80] with default options. Quality of the reads was assessed with NanoPlot 1.23.0[81]. The eighth sample consisted in a *Rickettsiales* bacterium associated with the foraminiferan *Reticulomyxa filosa*, already sequenced together with its host in a previous study[78]. Sequencing reads were kindly provided by the original authors.

### Genome assembly
For each sample, the total Illumina reads were assembled using SPAdes 3.6[82] with default settings, obtaining a "preliminary assembly". Then, a multi-step procedure was applied, in order to select only the contigs belonging to the symbiont of interest and discard those belonging to the host or to additional organisms present in the sample (e.g., residual food, additional associated bacteria), as described previously (e.g.,[8]). For this purpose, the blobology pipeline was applied[83], followed by extensive manual curation. Briefly, preliminary contigs were classified according to their length, GC% content, sequencing coverage, and taxonomy. Reads mapping by Bowtie2 2.4.2[84] on selected contigs were reassembled separately with SPAdes 3.6, or, for NDG2, with Unicycler[85] in a hybrid assembly with the respective Nanopore reads. Two samples (*P. multimicronucleatum* 12 and *P. biaurelia* US_Bl 11III1) were subjected to genome finishing, performing PCR reactions with TaKaRa Ex Taq and reagents (Takara Bio, Japan). Successful results were confirmed by bidirectional Sanger sequencing performed by GATC Biotech (Germany). Quality of the novel assemblies was confirmed by their completeness scores on 219 proteobacterial orthologs predicted with BUSCO 5.0.0[86], as compared with the scores of published *Rickettsiales* (Supplementary Data 2).

### Annotation
The newly obtained genomes were all annotated with Prokka 1.10[87], using the --rfam option. Afterwards, annotation of the genomes of ciliate symbionts was manually curated by a detailed inspection of blastp hits on NCBI nr and on *Rickettsiales* proteins as described previously[8].

### Full *Rickettsiales* dataset construction and phylogenomic analyses
Phylogenomic analyses were aimed to collect a representative and comprehensive view on the evolution and diversity of *Rickettsiales*. All sequences were downloaded from NCBI GenBank via ftp (ftp.ncbi.nlm.nih.gov/genomes/all/GCA), and are updated to July 2021. We manually selected a representative set of 36 *Rickettsiales* genomes, including at least one representative per genus. For the phylogeny, other 89 representative non-*Rickettsiales Alphaproteobacteria*, as well as 8 *Gammaproteobacteria* and *Betaproteobacteria* as outgroup were employed, taking inspiration from the selection by Muñoz-Gómez and co-authors[1]. We then aimed to identify all MAGs (metagenome-assembled genomes) which could be assigned to known "core" *Rickettsiales* lineages as of July 2021 (i.e., the four families *Rickettsiaceae*, *Anaplasmataceae*, *Midichloriaceae*, *Deianiraeaceae*), or to any lineage forming a supported monophyletic group with *Rickettsiales* with the exclusion of other (alpha)proteobacterial orders. Identification and representative selection of *Rickettsiales* MAGs was performed by a multi-step procedure (detailed in Supplementary Note 4). Briefly, all MAGs assigned to *Rickettsiales* by NCBI taxonomy, those assigned to deep-branching alphaproteobacterial lineages[25], plus all additional monophyletic relatives from the gtdb tree[88], were downloaded (394 total MAGs). MAGs were first filtered by assembly quality, retaining only 314 MAGs having ≥50% single-copy and <5% duplicated of 219 proteobacterial orthologs according to BUSCO 5.0.0 (Supplementary Data 2).

All phylogenomic analyses were performed on concatenated alignments of 179 orthogroups (Supplementary Data 6), which were manually selected on purpose (i.e., presence, after manual polishing of paralogs and poorly aligned sequences, in at least 85% of *Rickettsiales* -MAGs excluded-, 85% *Alphaproteobacteria*, 50% outgroup) from the eggNOG v5 assignments[89] predicted with eggnog-mapper 2.0.6[90]. For each organismal dataset (see below), orthogroups were separately aligned with MAFFT 7.475[91], polished with BMGE 1.12[92] and concatenated with AMAS[93]. All phylogenies were performed with IQ-TREE 1.6.12[94], with 1000 ultrafast bootstraps[95] and 1000 SH-aLRT replicates, employing the LG + C60 + F + R6 model unless specified. A first phylogeny was performed on the full organismal dataset for an initial classification of MAGs, employing ModelFinder[96,97] for model

selection. In order to avoid artefacts due to compositional heterogeneity in the dataset (in particular potential "erroneous" phylogenetic proximity of MAG lineages to "core" *Rickettsiales* due GC/AT biases)[1,25,26], the approach by Muñoz-Gómez and co-authors[1] was applied, thus removing 10%, 20%, 30%, 40% or 50% of most heterogeneous sites from the alignment, and performing a separate phylogeny on each trimmed alignment (Supplementary Fig. 29). Based on resulting monophyly and Average Amino acid Identity (AAI) > 0.85, phylogenetically-redundant MAGs were discarded (Supplementary Fig. 30). Thirteen clades were identified, grouping all those MAGs which could not be directly assigned to "core" *Rickettsiales* or other orders. In order to minimise potential artefacts (e.g., due to long-branch attraction), the phylogenetic position of the MAGs belonging to each clade was tested separately, with respect to core *Rickettsiales* lineages and other *Alphaproteobacteria* (Supplementary Fig. 31, 32). Phylogenies were performed accounting for compositional heterogeneity as above, and only MAGs in a monophyletic relationship with *Rickettsiales* were retained. Therefore, 68 total *Rickettsiales* MAGs were selected for all the successive analyses, totalling 210 organisms in the final dataset (113 *Rickettsiales*). For the sake of phylogenetic representativeness, this final dataset included sequences from the AT-rich *Holosporales* and *Pelagibacterales* genomes, and lacked, also due to computational limits, sequences from non-*Rickettsiales* MAG-only alphaproteobacterial lineages (e.g., the MarineAlpha by[255]). Accordingly, the reconstructions of the evolutionary history of *Rickettsiales* genes (see below) could have been potentially affected. Nevertheless, this was taken into account in the interpretation of the results, and potential artefacts due to the AT-rich sequences were directly addressed (see below).

For the final dataset, we applied the same approach as above (IQ-TREE phylogeny accounting for compositional heterogeneity; Supplementary Fig. 18).

### Creation of a set of "homology groups" for gene content comparisons

In order to perform gene content comparisons and infer variations along the inferred species tree, we aimed to obtain a set of broad "homology groups" for the 210 total organisms in the final phylogeny rather than analyse directly the "raw" orthogroups from eggNOG. The aim of this step was to get a comprehensive overview of the homologous genes in the dataset, namely regardless of whether the common ancestor of all sequences within the same "homology group" was younger or older than the ancestor of the investigated organisms. This allowed us a comprehensive inspection of homologues, providing the basis for de novo inference/analysis on the possible duplication and/or horizontal transfer events specific to the *Rickettsiales*, including from distantly related organisms falling outside the taxonomic range herein directly investigated. This was realised by merging the orthogroups resulting from eggNOG assignments that, based on eggNOG itself, shared significant homology (see Supplementary Note 5 for details). Briefly, the eggNOG database is hierarchically organised by taxonomy, namely each lineage (at domain, phylum, class, order, family levels) has a dedicated set of orthogroups, linked to those of higher-level taxa. We aimed to maximise the advantages of such a system (chiefly lineage-specific annotations), and at the same time minimise disadvantages for the intended aim. These disadvantages may include incomplete grouping of homologues (including potential orthologs) due to assignment to eggNOGs belonging to distinct taxa[8]. To overcome such limitations, we designed a "telescopic" approach, in order to merge genes into larger meaningful "homology groups", while still keeping as much as possible lineage-specific refined annotations. Briefly (see Supplementary Note 5 for details), we compared taxonomic paths of all the identified eggNOGs, and grouped into a single "homology group" all the eggNOGs sharing at least a partial taxonomic path, and labelling each "homology group" with the eggNOG at lowest possible

shared rank in the taxonomic path leading to *Rickettsiales* (root; *Bacteria*; *Proteobacteria*; *Alphaproteobacteria*; *Rickettsiales*). Applying such a "telescopic" approach, a total of 444,226 genes were assigned to 20,041 "homology groups", 4009 of which present in at least one member of *Rickettsiales*, and 2990 of those present in 4 or more organisms of the total dataset, and thus considered for the following analyses. The presence/absence and copy number patterns of each homology group were carefully manually inspected, in order to get a general overview of the functional repertoire of the investigated *Rickettsiales*, and to instruct more specific in-depth analyses (see below).

### Phylogenetic analyses on biosynthetic pathways for amino acids and nucleotides

Reference biosynthetic pathways for amino acids and nucleotides were obtained from the Biocyc database[55] (Supplementary Data 7). The datasets for the respective phylogenies were extensively manually curated (see Supplementary Note 5 for details). Briefly, for each gene the corresponding eggNOG "homology group" was identified by a blastp search, and its composition was refined (e.g., excluding clear paralogs, also with the help of NCBI conserved domain search[98], and poorly aligned sequences) by inspection of the respective alignment and of the respective single-gene tree. Preliminary and final single-gene trees were obtained with IQ-TREE and ModelFinder, after aligning and trimming as described above ("Full *Rickettsiales* dataset construction and phylogenomic analyses")[94]. While inspecting the phylogenies, we focused on verifying the support for the monophyly of the whole *Rickettsiales* and of each *Rickettsiales* family, or alternatively on reconstructions suggesting possible HGT events with *Rickettsiales* as recipients. On the other hand, potential events with *Rickettsiales* as donors, such as sequences of other lineages nested within a clade of *Rickettsiales* sequences, were not highlighted, being non-target for the aims of this study.

Based on inspection, whenever appropriate (i.e., all cases except cysteine synthesis, see results), we removed suspected non-orthologs genes, and, in order to get more phylogenetically informative datasets for more robust and reliable inferences, we concatenated together[93] the sequences involved in the same pathway, as well as in pathways sharing common reactions. Each concatenated alignment was also processed as described above ("Full *Rickettsiales* dataset construction and phylogenomic analyses") in order to account for compositional heterogeneity[1]. On each resulting alignment, phylogeny was inferred with IQ-TREE and the LG + C60 + F + R6 model, as described above. Such phylogenies were inspected with the same criteria as above for single-genes.

### Identification of amino acid transporters

All the proteins of the 113 *Rickettsiales* of our final dataset were employed as queries in a blastp search against the full TCDB database[61]. Then, for each *Rickettsiales* genome the number of proteins having a best significant hit (e-value threshold of 1e-5) on each selected entry were counted (see Supplementary Note 5 for details on the selection and refinement of TCDB entries representing putative amino acids transporters and on the rationale for the blastp search).

### Identification and phylogenetic analysis of the tlc nucleotide translocases

Analyses were focused on the tlc nucleotide translocase transporters (see Supplementary Note 5 for details), common in *Rickettsiales* and in other host-associated lineages[59]. Briefly, the corresponding eggNOG "homology group" was identified by a blastp search. Then, it was joined together with a selection of the phylogenetic dataset of tlc translocases by Major and co-authors[59], corresponding to the clade of sequences consisting only of the nucleotide transport protein domain. The sequences were then aligned and trimmed as described above,

and phylogeny was inferred as described above, employing the LG + C60 model as in[59]. Potential HGT events involving *Rickettsiales* were inferred based on the comparison with organismal phylogeny. Namely we inferred potential events for monophyletic groups of genes belonging to (part of) a *Rickettsiales* family with a sister group relationship with non-*Rickettsiales* (or non sister-lineage *Rickettsiales*) sequences. Only events with *Rickettsiales* as recipients were considered, and a range of inferred events is provided, depending on alternative reconstructions involving gene losses.

**Identification of genes involved in the interaction with host cells**
For getting information on the presence and multiplicity of genes involved in multiple features of the interaction of *Rickettsiales* with host cells, the VFDB core reference database was employed[99] (see Supplementary Note 5 for details). The VFDB is quite redundant for orthologs identified in different included pathogens. Thus, in order to make it suitable for analyses on our non-model bacteria, for each VFC (Virulence Factor Class) separately, orthologs were identified within the database with OrthoFinder 2.5.4[100], and manually curated. Then, all proteins of our dataset of *Rickettsiales* were employed as queries in a blastp search on the full VFDB core database. For each *Rickettsiales* genome, proteins were counted as "assigned" to each curated orthogroup if displaying a significant (evalue 1e-5) best blastp hit on any sequence belonging to the orthogroup. Data visualisation was attained with the ComplexHeatmap R package[101].

**Dataset update**
After the full set of analyses had been performed as described above, we aimed to further verify whether any *Rickettsiales* genome recently published in the meantime could provide additional insights on our main conclusions. Thus, we identified and downloaded from NCBI nine novel *Rickettsiales* genomes[13,19,21,22], belonging to novel genera or closely related to those herein newly obtained, and performed some general verifications. Specifically, we compared their BUSCO scores (calculated as above), as well as the respective presence of biosynthetic pathways for nucleotides and amino acids and of tlc nucleotide translocases (in terms of predicted eggNOGs, see above). All the results were highly consistent with those obtained with relatives present in the dataset employed for the whole set of analyses of this study (Supplementary Data 2, 8).

**Reporting summary**
Further information on research design is available in the Nature Portfolio Reporting Summary linked to this article.

# Data availability
Sequences obtained in this project were deposited to NCBI under accession number PRJNA831616. The annotated genome sequences of the symbiont of *Reticulomyxa filosa* is available on Zenodo (https://doi.org/10.5281/zenodo.10324454). Accession numbers of published *Rickettsiales* assemblies (including all initial MAGs) analysed in this study are provided herein (Supplementary Data 9). The eggnog (http://eggnog5.embl.de), BioCyc (https://biocyc.org), TCDB (https://www.tcdb.org), and VFDB (http://www.mgc.ac.cn/VFs) databases were also employed in this study.

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

## Acknowledgements

This project was supported by the Human Frontier Science Program (HFSP) Young Investigator Program grant RGY-0075 to D.S., by the Italian Ministry of Education, University and Research (MIUR): Dipartimenti di Eccellenza Programme (2018–2022) Department of Biology and Biotechnology 'L. Spallanzani' University of Pavia to D.S., by the European Community's H2020 Programme H2020-MSCA-RISE 2019 under grant agreement n° 872767 to G.P., and by EU funding within the NextGenerationEU-MUR PNRR Extended Partnership initiative on Emerging Infectious Diseases (Project no. PE00000007, INF-ACT) to M.C. and D.S. Genomic characterisation of *Trichorickettsia* and *Megaira venefica* was performed partly with support of RSF 20-14-00220 to A.P. The University of Pisa is acknowledged for providing visiting scholarships to E.S. and A.P. The authors would like to thank Venkata Mahesh Nitla for support in culturing *Plagiopyla frontata* IBS-3, Sascha Krenek for DNA preparation of the *Paramecium biaurelia* strain US_Bl 11III1, Umberto Postiglione for assistance in Nanopore assembly, Laura Quattrini and Marco Fagioli for assistance in genome closing by PCR. Gernot Gloeckner and Marco Groth are gratefully acknowledged for sharing genome sequencing data on *Reticulomyxa filosa* and its associated *Rickettsiales* bacterium.

## Author contributions

M.C., G.P and D.S. conceived and designed the study. M.C., E.S., A.P. and V.S carried out the experimental part, which was supervised by G.P. M.C. and T.N. performed data analysis, which was contributed by L.G. and G.B. and supervised by D.S. T.N. and G.B. curated data presentation. M.C and D.S performed conceptualisation and wrote the paper, which were contributed by all authors.

## Competing interests

The authors declare no competing interests
