## [Peer Review File · Nature Communications]

Host association and intracellularity evolved multiple times independently in the RickettsialesREVIEWER COMMENTS

Reviewer #1 (Remarks to the Author):

Castelli et al are interested in the origins of host dependence and intracellularity in the Rickettsiales and to that end manage to greatly increase the sample and diversity of Rickettsiales genomes. Nine genomes (six Midichloriaceae and three Rickettsiaceae) were sequenced by the authors themselves, while they find an impressive 68 Rickettsiales-related MAGs in the public databases. Among these are many new representatives of the recently defined Deianiraeaceae, Gamibacteraceae and Mitibacteraceae, but most importantly representatives of entirely new families (dubbed Diomedesiaceae, Jistubacteraceae and Arkhamiaceae).

This greatly enhanced sampling breaks the branch between other Alphaproteobacteria and the "crown" Rickettsiales (i.e. those that include all experimentally characterized lineages) in multiple places, allowing the authors to peer into the origins of host dependence and intracellularity with unprecedented detail.

The evolutionary histories of molecular complexes and pathways related to host association are investigated, and it is concluded that the overall patterns observed are best explained by multiple independent origins of obligate host association and intracellularity, rather than the single origin that was previously the general belief of the field.

I'm very impressed by the amount of work and the new insights that the authors manage to extract from their analyses. It is a huge step forward into the field. The manuscript is generally well written, though it needs some polishing. I do however have a number of technical concerns and one interpretational concern (see below). My main concern lies with the method used for orthogroup reconstruction (see below), which if the authors agree with my concerns, will require a major revision. Other technical concerns are relatively minor.

SPECIES TREE RECONSTRUCTION

As the authors point out, inferring the Alphaproteobacteria species tree is prone to systemic errors such as the grouping of Holosporales and Pelagibacterales with other Rickettsiales. A common strategy in past studies was to progressively remove the sites that have been biased most by evolutionary shifts in amino acid compositions. However, as the authors rightfully point out, this comes at the cost of signal important to resolve nodes along the backbone of the tree, including possibly those delineating the

relationships between and within the major Rickettsiales families. Rather than removing sites, the authors choose to constrain the tree search by locking in the Holosporales+Rhodospirillales and Pelagibacterales+Rhodobacterales/Rhizobiales/Caulobacterales bipartitions. This is an unorthodox strategy, of to which I have some concerns.

First I would like to see evidence for whether the inner relationships indeed lose their resolution when the most biased sites are removed. I believe the authors but it will make the manuscript stronger if this was shown.

Systemic errors in phylogenies are ultimately failures of the used substitution model to capture certain facets of sequence evolution. In this case, shifts in amino acid compositions across different branches of the tree. If one constrains the tree search such that known artefacts are not allowed, the shifts in amino acid composition are not suddenly properly modelled. The sites that have been most biased by shifts in composition may still support other, if less dramatic, phylogenetic artefacts. There is no guarantee that sites that previously strongly supported the false Holosporales and Pelagibacterales positions now only support the true tree. One way to test for this is to repeat the constrained tree search but with progressively removing the most biased sites. If certain strongly supported groupings are replaced by other strongly supported groupings as more biased sites are removed, we can conclude a residual artefact is at play. If no new strongly supported groupings appear, we can have some more faith the Figure 1 tree is the true tree.

In addition, rather the constraining the tree search, one could simply remove the Holosporales and Pelagibacterales from the analysis entirely. Again, this does not automatically mean there are no other artefacts at play and one would need to test for this with progressive site removal, but it would circumvent the need for constraining tree searches. As they are relatively few genomes and are in the outgroup, I do not expect there is a lot of benefit for including them to gain insights into gene content evolution of the Rickettsiales anyway. I would furthermore omit both lineages from the orthogroup reconstruction and ALE analysis as well, as phylogenetic artefacts also probably perturb the single gene phylogenies on which ALE relies.

The authors find that the resulting species tree is consistent with the tree published by Schon et al 2022, with a single exception: the relationship between the 'DDG' clade and Anaplasmataceae and Midichloriaceae. Whereas Schon et al find a ('DG', (Ana,Midi)) relationship, this study finds (('DDG',Ana),Midi). The authors attribute this difference to the substantially increased 'DDG' sampling. However, upon closer examination, both studies find the same tree under similar conditions. Schon et al namely also get (('DG',Ana),Midi) in their ML analysis when they do not remove biased sites. The increased taxon sampling therefore did not seem to influence this particular topology much. Rather, it seems site selection seems to be more important here. To make a proper comparison with the Schon et al tree, the authors should do a tree inference when biased sites are removed.

Although perhaps not particularly important to the question of the study, I'm wondering why MarineAlpha lineages were omitted from the Alphaproteobacteria outgroup. They were mentioned to be initially collected in Supplementary Text 4, but must have been cut at some point during the various selection procedures. They represent much Alphaproteobacterial diversity and have decent-to-high levels of completeness and low contamination rates, so it seems odd to me they were removed. If the authors do decide to add them in their revision, I would recommend not adding those sensitive to phylogenetic artefacts (see Martijn et al 2018, Nature which lineages these are).

ORTHOGROUP RECONSTRUCTION

Orthologous groups, or orthogroups, are defined as all genes of a given group of species that have evolved from a single ancestral gene in either the last common ancestor of that group of species or one of the more recent ancestors within that group (Gabaldon & Koonin 2013). If a pair of genes has a last common ancestor that predates the last common ancestor of the taxonomic clade in consideration, they should be placed in separate orthogroups. If we take the Rickettsiales as an example, an orthogroup should thus consist of all genes that have descended from a gene present in the last common Rickettsiales ancestor, or in one of its more recent ancestors, e.g. the ancestor of the Anaplasmataceae or any of the other intermediate ancestors.

In this study, the span of taxa considered is that set of taxa used in the main species tree (Figure 1 / Supplementary Figure 1). I assume that this is the tree that is used by the authors when they reconcile gene trees with a species tree using ALE, although this is not explicitly mentioned in their Methods nor Supplementary Text 5. Strictly speaking, the last common ancestor of the species tree was a proteobacterium, and thus ideally, orthogroups should be reconstructed at the Proteobacteria level. However, as the representation of Beta- and Gammaproteobacteria and Magnetococcales (the outgroups) are relatively poor, orthogroups at the Alphaproteobacteria level would be good as well. In that case, however, it may be better to use only the alphaproteobacterial part of the species tree for the ALE analysis.

The authors make use of the eggNOG database (I'm assuming v5? it is not mentioned) to reconstruct their orthogroups. The eggNOG database contains many pre-reconstructed orthogroups at many different taxonomic levels (e.g. the Bacteria level, Metazoa level, Euryarchaeota level etc etc) which are hierarchically linked. This means that for example a single Bacteria NOG can be broken down in many NOGs of one taxonomic rung lower (for example Proteobacteria, Firmicutes etc), which each in turn can be broken down into smaller NOGs (Alphaproteobacteria, Betaproteobacteria) etc etc. To reconstruct their orthogroups, the authors employ the eggnog-mapper. With a predicted protein from a newly sequenced genome as a query, one can use the eggnog-mapper to search the eggNOG database for a

protein sequence that is likely the closest homolog of the query in the database, i.e. the seed ortholog. This seed ortholog is associated with possibly multiple NOGs of different taxonomic ranks (i.e. a protein of a Rickettsiales NOG is also associated with Alphaproteobacteria NOG, Proteobacteria NOG etc etc). The NOG that is most appropriate for your analysis is then assigned to the query. In other words, it is assumed that the query and the seed-ortholog have a last common ancestor that falls within the taxonomic scope of the NOG. Finally, all queries that were assigned to the same NOG are grouped within a reconstructed orthogroup. It is unfortunately not clear what taxonomic level is used by the authors. In Supplementary Text 4 it is stated that the Alphaproteobacteria are used, but in Supplementary Text 5, it is stated that some query proteins were assigned to Metazoa and Firmicute NOGs, which clearly fall outside the Alphaproteobacteria suggesting perhaps that the root level was used.

The eggNOG mapping approach to reconstruct orthogroups has been used by other studies in the recent past and is in my view an appropriate and pragmatic one. However, the authors appear to run into a problem with the standard implementation of this approach. Some proteins of the newly sequenced taxa are assigned to NOGs of lineages as distant as Metazoa and Firmicutes, where (I assume) they expected them to be assigned to (Alpha)proteobacteria or Rickettsiales NOGs. They blame this supposedly false mapping to the higher sequence evolutionary rates in Rickettsiales. Though plausible, I think there are other factors at play. When a Rickettsiales query protein hits a seed-ortholog outside the Alphaproteobacteria, this could be explained in several ways:

(i) this protein is new in Alphaproteobacteria (that is, compared to the Alphaproteobacteria in the eggNOG db) and has previously been only observed in other lineages. The associated gene may have been transferred from some other lineage to a (possibly) very recent ancestor of the newly sequenced Rickettsiales species. In which case, the query protein and the seed-ortholog have a last common ancestor that predates the last common Alphaproteobacteria ancestor, and they belong to separate orthologous groups. The *Aquarickettsia* protein RST68180.1 mapping to Firmicutes may be explained this way. A quick BLASTP search reveals it has a few hits in other Midichloriaceae (to which *Aquarickettsia* belong), but otherwise mostly hits Firmicutes. All queries that hit the same distant NOG could be considered descendants of a single ancestral incoming transfer event and could thus be part of the same true orthogroup. However, it may also happen that multiple independent transfers have occurred from that distant NOG into the Rickettsiales, in which case they should be in distinct orthogroups.

(ii) the seed-ortholog is in fact Alphaproteobacterial, but is a contaminant of a genome in the distant lineage and hence mislabeled as e.g. Metazoan. I strongly suspect this to be the case for the seed-ortholog of *Aquarickettsia* protein RST68169.1. A quick BLASTP search vs nr reveals that it has in fact many alphaproteobacterial homologs, but one of the top hits is to *Trichoplax adhaerens*. This placozoan is known to harbor a Rickettsial endosymbiont belonging also to Midichloriaceae (see Driscoll et al 2013, GBE). Furthermore, the Metazoa NOG 3C1KH only contains four other sequences, three of which are from the antelope genome *Pantholops hodgsonii*, a genome unfortunately notorious for its high level of contamination. The Metazoan NOGs 3BT1X and possibly 3C01C have similar red flags. For query proteins

that hit contaminating seed-orthologs, the NOG assignment can not be trusted. Ideally, the best hit after eliminating contaminated seed-orthologs should be used for NOG assignment.

(iii) the query protein is in fact not Alphaproteobacterial, but a contaminant of the new Rickettsiales genomes. I do not expect this to happen, if at all, as the authors have done a very thorough job sequencing and curating their genomes and selecting MAGs with minimal contamination levels.

(iv) the query protein has undergone highly elevated substitution rates and has no close relatives in the Alphaproteobacteria NOGs, such that true more distant Alphaproteobacteria relatives appear as more distant to the query protein than proteins from even more distant lineages. In this case, the query protein should be assigned an Alphaproteobacteria NOG, but are assigned to NOGs of distant lineages and would therefore be wrongly separated from the true NOG it belongs to. This is the explanation the authors use to explain their example deviant NOG assignments. I suspect however that this is a relatively minor issue, though I could be wrong. Virtually all "basal Rickettsiales" do not have long branches, and most of the newly sequenced genomes and identified MAGs do not either. Several taxa, such as Fokinia, Lyticum, Xenolissoclinum and others of similar length may pose problems though. The authors could opt to simply omit them from the orthogroup reconstruction and prune them from the species tree. The Deianiraeaceae are unfortunately all very long branching, but are critical to understanding early Rickettsiales evolution, so they should be kept in.

Issues (i) and (ii) could for a large part be resolved simply by strictly using only the Alphaproteobacteria NOGs for orthogroup assignment and reconstruction: you make sure you don't hit any genomes contaminated with alphaproteobacterial genes, and you also make sure you don't hit possible distant gene transfer donor lineages. If such transfer-descendants have no orthologs in the current eggNOG db set of Alphaproteobacteria NOGs, i.e., they don't have hits in the Alphaproteobacteria NOGs, one can always do a de novo clustering of all unmapped queries, with Silix + Hifix or some other clustering algorithm. And as suggested, issue (iv) could be ameliorated by omitting the longest branching lineages from the analysis. I do not think the system of the eggNOG database lies at the heart of the issue the authors encountered, at least not with the discussed Aquarickettsia examples. Another suggestion would be to use the HMMER search strategy of the eggNOG-mapper instead of the default DIAMOND. DIAMOND is appropriate for taxa that have close relatives in the eggNOG database, whereas HMMER is more appropriate for more distant taxa (Huerta-Cepas, 2017 GBE). Since especially the "basal Rickettsiales" are not represented and are relatively distant to the Rickettsiales that are present in eggNOG, the HMMER option seems a better choice.

Regarding the "telescopic" approach, I have some concerns here as well. If I understand correctly (the method is not explained very well, it could be much clearer), query proteins that hit a NOG with a taxonomic path that falls outside the Rickettsiales path will in the first step be assigned to the most recent common ancestor between both paths. For example, a query protein that hit a Rhizobiales NOG would be labeled as the Alphaproteobacteria NOG that is the parent of that rhizoNOG, and for a query

protein that hit a Metazoa NOG, it would be labelled as a root NOG. Then, in the second step, for all those query proteins labeled with a "root" NOG, it was checked which shallower NOGs along the Rickettsiales path (e.g. Bacteria NOGs, Proteobacteria NOGs, Alphaproteobacteria NOGs and Rickettsiales NOGs) were linked to that "root" NOG and all query proteins assigned to any of these shallower NOGs were pooled together with those assigned to the root NOG.

This seems to me a dangerous approach. A single root NOG may have multiple "daughter" Bacteria NOGs, Proteobacteria NOGs etc. This can happen if a duplication happened in some deep ancestor. For example if two Alphaproteobacteria NOGs are linked to the same root NOG, it means that a gene duplication occurred before the last common Alphaproteobacteria ancestor. A pair of query proteins with such a deep paralogy thus belong in truth to two separate orthogroups at the Alphaproteobacteria level and they may be wrongly lumped together in the same orthogroup using the described telescopic approach, in effect violating the definition of an alphaproteobacterial orthogroup.

In summary, I believe that the problems described by the authors (exemplified with the *Aquarickettsia* ribosomal proteins) can be better explained by more straightforward technical phenomena than elevated evolutionary rates, and that the telescopic approach, designed to address these issues is perhaps overly complicated and brings with it its own set of issues. I would recommend the authors to address the technical issues and stick to the simpler orthogroup reconstruction method (for example as described above). I hope I have properly understood the method used by the authors and that my feedback makes sense.

ALE ANCESTRAL RECONSTRUCTION

Many of the new MAGs selected for the species tree and orthogroups have good completeness estimates, but a good amount also have relatively poor ones. This may affect the ALE results if left unaccounted for. The accuracy of inferred evolutionary histories depends greatly on the representation of the extant diversity of genes, after all. ALE has a 'fraction_missing' option, which as far as I could tell was not used by the authors.

Despite the large amount of work that went into reconstructing orthogroups and the corresponding ALE analyses, they are hardly discussed in the main text. The Results paragraph of lines 182-191 could use a brief description of the used orthogroup reconstruction method and ancestral reconstruction method with ALE to allow the reader a better understanding of the analyses employed. For many molecular machineries (T4SS, Flagella, Type 4 pili, fhaAB, biosynthesis, transporters etc) the authors infer evolutionary histories (independent losses, horizontal acquisitions, ancestral presence/absence states etc) but do not check whether the inferred ALE histories of the corresponding orthogroups match with these interpretations. If all these inferences are available, why not make use of them?

PHYLOGENETIC ANALYSIS OF NUCLEOTIDE AND AMINO ACID BIOSYNTHESIS GENES

It is my understanding that ultimately the authors wish to find out whether the identified biosynthesis genes were present in the Rickettsiales ancestor or not. To check for this, one could simply identify which of the reconstructed orthogroups correspond to these genes and check the history inferred by ALE. The authors do indeed (if I understand correctly) identify the orthogroups (though this could be better explained in the Supplementary Text 5 and Methods; the way it is written could still be interpreted as eggNOG COGs/orthogroups referring to the actual eggNOGs, rather than the reconstructed orthogroups of the study. Also, SuppText 5 makes it seem as though entirely new orthogroups were constructed by BLAST search of the Biocyc reference against all proteins of all 210 taxa, which is confusing), but do not discuss the relevant ALE results. I imagine that these ALE histories had too little signal to say anything about ancestral presence / absence due to poor signal in single gene phylogenies, and that this was the reason the authors chose to infer a "concatenated" phylogeny and compare that with the species tree. This is OK, but I would like to see this explained in the manuscript, to make it more coherent.

The concatenation approach is a sound solution, but can only be done if there is reason to believe that all genes concatenated together share the same underlying tree. I see that short and poorly aligned sequences and putative paralogs were removed, but this does not automatically mean the remaining sequences were all generated with the same underlying tree. Yes, they do share the same pathway supposedly, but metabolic genes are generally more prone to HGTs. Are there any strongly supported major conflicts between these single gene phylogenies? If not, then this analysis is OK. I would like to see this discussed, however.

It may also be interesting to see what happens when the concatenated tree is reconciled with the species tree using ALE. Does it still support the same conclusion as the comparing the concatenated tree with the species tree by eye?

ANALYSIS OF NUCLEOTIDE AND AMINO ACID TRANSPORTERS

The authors rather convincingly show that in those lineages where biosynthesis pathways are reduced, the transporters (tlc for nucleotides, and various others for amino acids) are more prevalent. This is a very cool result and what we would expect to see given our current understanding of endosymbiosis evolution. The authors further make the claim that this pattern supports the idea that these transporters

have been acquired throughout Rickettsiales evolution multiple times independently (as opposed to the single-origin-with-consequent-multiple-independent-loss scenario proposed by Schon et al 2022). I agree that is what the presence / absence pattern and single gene tree suggests. But is this also reflected in the reconstructed orthogroups? If the orthogroups fulfill the condition I outlined above, each independent acquisition would correspond to a distinct orthogroup. Do the authors indeed see for example an orthogroup for tlc with only Midichloriaceae, a separate one for Deianiraceae, and three separate ones for sublineages of Rickettsiaceae, as Figure 3 suggests? What about the orthogroups corresponding to the amino acid transporters? If so, this would strengthen your conclusion. It may be that due to the "telescopic" approach, such orthogroups that should be distinct are merged together, in which case I am curious if they become distinct with the suggested orthogroup reconstruction approach described above.

Regarding the amino acid transporter counting approach described in Supplementary Text 5, I do not entirely understand why doing a BLASTP search against "trusted entries" is considered problematic, whereas doing a BLASTP search against the entire TCDB and then selecting matches with "trusted entries" is considered OK. Does it have something to do with the relation between E-value and database size? This should be clarified, if you wish to keep the description of your initial strategy. To keep things simple, I would honestly just describe the second search strategy, and omit the first. I don't think the second search strategy needs a particular justification.

ON THE INTRACELLULARITY-LATE HYPOTHESIS

In 2019 Castelli et al put forward the hypothesis that intracellularity has evolved not once, as previously thought, but multiple times independently throughout the course of Rickettsiales evolution. The main lines of evidence for this include (i) the extracellular nature of Deianiraea, whose phylogenetic placement breaks the monophyly of known intracellular lineages and (ii) the fact that intracellular lineages (to their knowledge) never revert to extracellular lifestyles. These are in my opinion the strongest arguments and already sufficient to deduce independent origins of intracellularity (I therefore find the title of this manuscript a little odd, as it gives the impression that this is a new finding of this study).

This study aims to find additional evidence for this hypothesis. It confirms the paraphyly of intracellular lineages with a phylogeny based on a greatly expanded dataset, and analyzes the gene content and their evolutionary histories of the new lineages. Unfortunately gene content is presently unable to perfectly differentiate between intra- and extracellular lifestyles. Even tlc, generally considered a strong predictor for intracellularity, is present in the extracellular Deianiraea, and absent in the Anaplasmataceae, endosymbiont of *Amblyomma* (the absence in the intracellular Midichloriaceae MAG is most likely explained by its mere 50.7% completeness). One of the most striking results of the study, that when aa/nt transporters are horizontally acquired, vertically inherited biosynthesis genes are lost. This is a strong marker for host-association / host-dependence (as the authors point out in the Discussion) but is

not a perfect predictor of intracellular lifestyles, simply because we do not know the lifestyles of the MAG organisms that retained the biosynthesis genes and lack these transporters. It is in my opinion therefore insufficient to serve as independent support for "intracellularity-late". This observation, as well as the observed patterns for other molecular systems (secretion systems, effectors, flagella, and pili), is what we expect to see given the hypothesis, but it does not support it. This distinction is in my view important, and the authors often thread this needle correctly, but are occasionally too strong (most notably in the title and abstract, but also in lines 433/434 and 467 in the Discussion) and should be adjusted.

OTHER GENERAL REMARKS

Many of the supplementary figures are difficult to read and require the reader to zoom in to read the text, but as a consequence lose sight of the overall picture. I understand that phylogenetic trees and presence-absence maps with these levels of taxon sampling can grow to unyieldy large sizes, but I believe nonetheless that in many cases font sizes can easily be increased. This is particularly true for ALE results (SuppFigure2), in which also the ODTL counts are near impossible to read. It would also be very nice (but optional) if the ODTL counts and node copy numbers could be scaled according to counts, along the lines of Figure 3 of Martijn et al 2020, Nature Communications. Though it lacks a tutorial, code for this can be found on https://github.com/maxemil/ALE-pipeline/blob/master/templates/visualize_ALE_results.py. The phylogenies of the biosynthetic pathways (SuppFigure8) also suffer from miniature fonts. If increasing the font size to a readable level makes the tree unreadable, it may work better to collapse certain clades and provide the complete trees as supplementary files.

Please also ensure that phylogenetic trees are "ordered", similar to Figtree's "Order nodes" option or the "ladderize" function in the ETE python package.

Regarding the heatmaps, gene counts are often "outshined" by few outliers with lots of copies, making it practically impossible to appreciate the differences between the more "regular" copy numbers. It should be possible to ignore outliers when coloring these maps.

For some Supplementary Texts, their file name does not match the content (e.g. Supp Text 4 is denoted as Supp Text 3, etc).

Regarding the running of single gene trees with PhyloBayes, what model was used? How many chains were run, and what was the burnin used? If multiple chains were used, did they converge?

Though BLAST is so pervasive in modern bioinformatics, and we use the term "blasting" on the workflow, in manuscripts I think we should stick to the more formal "performing a BLASTP search using query X against a database Y" while also mentioning E-value cutoffs etc.

Reviewer #2 (Remarks to the Author):

This manuscript examines genomic diversity and patterns of genome evolution in the Rickettsiales. The advances include a) de novo sequencing of nine new strains b) Extraction of diverse novel MAGs that represent new taxonomic groups/diversity c) Analysis of patterns of gene loss/gain across the phylogeny. I think aspects b and c are probably most significant for the field. I did enjoy the manuscript, and certainly felt I learned new things about Rickettsiales evolution. It felt a 'significant' manuscript.

The manuscript could be improved in two ways in my opinion.

First, the manuscript could be written with stronger focus, particularly in the introduction - establishing the aims and objectives of the study clearly. In general I could understand the manuscript, but there were a number of points where expression could be improved.

Second, the manuscript suffers somewhat from 'time' - that is to say, data had to be frozen at a point in time and then analysed, but since that point quite a few high quality genomes have come out that would have been great to include. I guess this is just a thing in genomic analysis - you have to stop somewhere and the analysis can take a good year, but there are cases where we know there are more complete genomes than those studied. However, it probably doesn't impact conclusions greatly - but the authors perhaps should check that their genomes reflect others that are more complete.

Other major concerns:

I have a wider concern over the quality of the genomes analysed; for the de novo genomes, all are fragmented and some have low BUSCO scores indicating elements are missing from the assembly, and it was a bit of a disappointment not to see long read technologies applied to improve quality. Fragmentation is even more true (inevitably) for the MAGs, any of which are really rather partial.

The authors do, cleverly in my opinion, tend to analyse across a group rather than individual genomes, but I was still left wondering whether absence of evidence was evidence of absence for key systems.

However, this brings its own problem, as Figures such as Figure 2 compare completeness in single genomes with multiple genomes (and I suspect counteracting this, relatively complete genomes vs quite incomplete ones). I'd like to hear the authors validation for this.

Incompleteness also means I am not clear how useful analysis such as supplementary figure 3 is. I would recommend the first column of this figure should be BUSCO Completeness just so interpretation can be made more easily.

I did also feel a little circularity in argument that basal were all environmental/extracellular - it feels it is asserted and then inferred at different points in the manuscript.

Dear reviewers,

Please find below our point-by-point response to your comments to our manuscript (answers in red). We feel that we were able to answer to all concerns, and that this has greatly improved the quality of the manuscript.

We thank you for your work.

Davide Sasseria and Giulio Petroni, on behalf of all authors.

Reviewer #1 (Remarks to the Author):

Castelli et al are interested in the origins of host dependence and intracellularity in the Rickettsiales and to that end manage to greatly increase the sample and diversity of Rickettsiales genomes. Nine genomes (six Midichloriaceae and three Rickettsiaceae) were sequenced by the authors themselves, while they find an impressive 68 Rickettsiales-related MAGs in the public databases. Among these are many new representatives of the recently defined Deianiraceae, Gamibacteraceae and Mitibacteraceae, but most importantly representatives of entirely new families (dubbed Diomedesiaceae, Jistubacteraceae and Arkhamiaceae).

This greatly enhanced sampling breaks the branch between other Alphaproteobacteria and the "crown" Rickettsiales (i.e. those that include all experimentally characterized lineages) in multiple places, allowing the authors to peer into the origins of host dependence and intracellularity with unprecedented detail.

The evolutionary histories of molecular complexes and pathways related to host association are investigated, and it is concluded that the overall patterns observed are best explained by multiple independent origins of obligate host association and intracellularity, rather than the single origin that was previously the general belief of the field.

I'm very impressed by the amount of work and the new insights that the authors manage to extract from their analyses. It is a huge step forward into the field. The manuscript is generally well written, though it needs some polishing. I do however have a number of technical concerns and one interpretational concern (see below). My main concern lies with the method used for orthogroup reconstruction (see below), which if the authors agree with my concerns, will require a major revision. Other technical concerns are relatively minor.

We thank the Reviewer for the appreciation of our work. We highly acknowledge the detailed and insightful comments received, which allowed us to greatly improve our manuscript accordingly.

We believe that the following premises may be useful to fairly envision our point of view in relation to some comments received. We hope that, in the refined re-description and with the implementations suggested, our approach will be now seen as scientifically sound and supportive for the main findings of this study. Please see the point-by-point answers below.

We are prone to think that some of the technical concerns, including the main one on orthogroups (please see below), may have stemmed from our insufficient description of our methodology and its rationale, and we apologise for that. Accordingly, we carefully examined each suggestion received, and followed those that could be directly implemented within our study design in order to improve its quality. On the other hand, considering the huge computational (and non-computational) work required, we judged unfeasible to substitute major steps of our work when the changes would not impact (in our view) the main findings of the study. In such cases, we did our best to fairly and

more clearly present the rationale and (in our view) validity of our approach, both in the response and in the manuscript.

SPECIES TREE RECONSTRUCTION

As the authors point out, inferring the Alphaproteobacteria species tree is prone to systemic errors such as the grouping of Holosporales and Pelagibacterales with other Rickettsiales. A common strategy in past studies was to progressively remove the sites that have been biased most by evolutionary shifts in amino acid compositions. However, as the authors rightfully point out, this comes at the cost of signal important to resolve nodes along the backbone of the tree, including possibly those delineating the relationships between and within the major Rickettsiales families. Rather than removing sites, the authors choose to constrain the tree search by locking in the Holosporales+Rhodospirillales and Pelagibacterales+Rhodobacterales/Rhizobiales/Caulobacterales bipartitions. This is an unorthodox strategy, of to which I have some concerns.

We understand the Reviewer's concern on this point. Indeed, to our knowledge, certain artefacts due to amino acids composition among *Alphaproteobacteria* are well acknowledged in the specialist literature, in particular the grouping of the unrelated (but AT-rich) *Rickettsiales*, *Pelagibacterales*, and *Holosporales*.

A number of previous studies developed and applied different strategies to valuably tackle this issue, typically by ranking sites by their compositional bias, and progressively removing most biased ones (e.g., 10.1371/journal.pone.0078858; 10.1038/s41586-018-0059-5; 10.7554/eLife.42535). Depending on how the compositional bias is defined and the corresponding metrics obtained, each of such methods will inevitably (and differentially) lead to the loss of phylogenetic information (and possibly to unintentional introduction of different kinds of biases). Importantly, it seems that the extent and impact of such drawbacks cannot be fully predictable, especially in case a significant amount of sites (50% or even more) is removed.

While designing our study, we could not identify any consensually orthodox strategy from the debates of the (quite specialist) scientific community. We reasoned that, for our study aims, it was paramount to make sure that any close relationship between deep-branching MAG lineages and crown *Rickettsiales* was "trustworthy", and not artefactual due to compositional biases, such as independently evolved high AT-content. Accordingly, after extensive preliminary tests (not shown), we opted for a published approach (10.7554/eLife.42535) that was designed on purpose to tackle putative artefactual grouping of AT-rich *Alphaproteobacteria*, and used it in the initial phylogenetic steps of our workflow (selection of MAGs, see Supplementary text 4).

In a successive stage of our workflow, we did not find an unambiguous empirical and theoretical support for the employment of the same strategy for defining the inner relationships within *Rickettsiales*. At the same time, we aimed to keep other lineages such as *Holosporales* and *Pelagibacterales* in our dataset as part of the alphaproteobacterial phylogenetic backbone for further analyses. We thus reasoned that, although not conventional, the constrained search was a convenient choice for our needs (please see more specific answers below for further rationales and details).

First I would like to see evidence for whether the inner relationships indeed lose their resolution when the most biased sites are removed. I believe the authors but it will make the manuscript stronger if this was shown.

The Reviewer is right, we have now added those trees as "Supplementary figure 2", and we added a paragraph in the main text describing the thought process and preliminary results (lines 154-166,

666-670 of the track-mode manuscript). As compared to the not compositionally-trimmed (and unconstrained) tree, when applying compositional filters some inner relationships partly lose support, going even below the recommended IQTREE threshold of 90% UFB. Moreover, in some trees alternative reconstructions were obtained for the relationships between some families, namely among “basal” *Rickettsiales* and among the *Anaplasmataceae*, *Midichloriaceae*, and DDG clade. However, such alternative reconstructions never reached 90% UFB. While we acknowledge that the relevance of finding alternative reconstructions is debatable (please see the response to a specific point below), we underline that our considerations on the opportunity of using compositional trimming for defining inner relationships was not based solely on those specific trees inferred on the final dataset, but also on multiple preliminary tests with smaller datasets, which produced trendless and non-repeatable alternative reconstructions between the families, with even more pronouncedly low supports.

Systemic errors in phylogenies are ultimately failures of the used substitution model to capture certain facets of sequence evolution. In this case, shifts in amino acid compositions across different branches of the tree. If one constrains the tree search such that known artefacts are not allowed, the shifts in amino acid composition are not suddenly properly modelled. The sites that have been most biased by shifts in composition may still support other, if less dramatic, phylogenetic artefacts. There is no guarantee that sites that previously strongly supported the false Holosporales and Pelagibacterales positions now only support the true tree. One way to test for this is to repeat the constrained tree search but with progressively removing the most biased sites. If certain strongly supported groupings are replaced by other strongly supported groupings as more biased sites are removed, we can conclude a residual artefact is at play. If no new strongly supported groupings appear, we can have some more faith the Figure 1 tree is the true tree.

We thank the Reviewer for the insightful suggestion. We agree on the complexity of the outcomes and influences of each potentially biased site in different reconstructions. At the same time, we are not fully convinced that applying simultaneously constrained tree search and compositional filtering would provide fully reliable supports for any specific reconstruction on the inner relationships within *Rickettsiales*.

Compositional trimming approaches are chiefly dependent on the strength of the bias and on how biased sites are identified. Thus, any valuable but inherently empirical approach will allow to counteract the strongest and sharpest biases (if any), such as grouping of unrelated lineages with convergently evolved comparable AT-shift, especially the on-purpose established approach herein used (doi: 10.7554/eLife.42535). In principle, the same strategies may be less powerful (and reliable) when hypothetically aiming to counteract more “subtle” biases (if any) between more closely related organisms (such as, in this instance, *Rickettsiales* sublineages). Consequently, it seemed not so straightforward to predict their reliability on the inferred inner topologies after compositional trimming, when considering also the relative impact of the loss of information and potentially introduced biases due to the trimming itself.

In sum, while we acknowledge the relevance of the issue raised, we considered that the most appropriate set of analyses would be to provide the constrained but untrimmed tree (as in the original manuscript), with the addition of the trimmed but unconstrained trees, as suggested by the Reviewer in another comment above (please see also below for interpretation of the differences observed between reconstructions).

In addition, rather than constraining the tree search, one could simply remove the Holosporales and Pelagibacterales from the analysis entirely. Again, this does not automatically mean there are no other artefacts at play and one would need to test for this with progressive site removal, but it would circumvent the need for constraining tree searches. As they are relatively few genomes and are in the outgroup, I do not expect there is a lot of benefit for including them to gain insights into gene content evolution of the *Rickettsiales* anyway. I would furthermore omit both lineages from the

orthogroup reconstruction and ALE analysis as well, as phylogenetic artefacts also probably perturb the single gene phylogenies on which ALE relies.

We thank the Reviewer for this suggestion. We did keep on purpose *Holosporales* and *Pelagibacterales* in our final phylogenetic dataset. We aimed for a representative selection of the described alphaproteobacterial orders for the successive analyses, including ALE (requiring the backbone species tree) and the more specific phylogeny on pathways of interest. We agree on the risk of potential artefacts also in single-gene phylogenies, but we judged that, provided a careful and “conscious” examination of the results, the robustness and informativeness of our reconstructions on *Rickettsiales* would have overall benefited from such inclusions. For example, especially *Holosporales* (sharing comparable niches with *Rickettsiales*) may have been potential donors in hypothetical HGT events (later not found for the main pathways examined). Moreover, retrieving independent lineages for the biosynthetic genes of *Rickettsiales* and each of the other two orders (as later found among the main pathways examined) would have provided, by comparison, indirect support that the monophyly of *Rickettsiales* genes was not artefactual but due to actual vertical inheritance.

The authors find that the resulting species tree is consistent with the tree published by Schon et al 2022, with a single exception: the relationship between the 'DDG' clade and Anaplasmataceae and Midichloriaceae. Whereas Schon et al find a (('DG', (Ana,Midi)) relationship, this study finds (('DDG',Ana),Midi). The authors attribute this difference to the substantially increased 'DDG' sampling. However, upon closer examination, both studies find the same tree under similar conditions. Schon et al namely also get (('DG',Ana),Midi) in their ML analysis when they do not remove biased sites. The increased taxon sampling therefore did not seem to influence this particular topology much. Rather, it seems site selection seems to be more important here. To make a proper comparison with the Schon et al tree, the authors should do a tree inference when biased sites are removed.

We apologise that this point was not clearly and adequately presented in our previous version of the manuscript, we have now modified it accordingly (lines 154-166 of the track-mode manuscript).

The Reviewer is right in considering that compositional site selection seemingly has a relevant influence in the topology. Indeed, some of the trees with the compositional filters applied, now included as suggested by the Reviewer (Supplementary figure 2; see above), did present the (('DDG', (Ana,Midi)) topology, which would be consistent with the findings by Schon and colleagues.

As detailed in the responses above, we considered that there is not yet a consensual orthodox strategy for the application of compositional site selection for determining such more subtle inner relationships. We also observed that, in any case, even with the deepest and thus potentially most debatable site trimming (50%), the alternative topology never reached a fully adequate support for our dataset (<90% UFB).

Accordingly, we judged that this point is still somehow arguable according to the available data, and somewhat beyond the aims of the current study, potentially deserving further in-depth analyses. Thus, while we do not favour in principle any specific topology, we had to “select” one as reference for the successive analyses (in particular ALE), and we felt that the fairest option was the (('DDG',Ana),Midi), being based on compositionally “untreated” dataset. Indeed, a priori we considered any potential bias in the untreated dataset as “preferable” to unpredictably introduced ones and loss of signal due to compositional trimming.

Regarding the relevance of those differences in topology for the current study, their impact for our successive reconstructions would be inevitably not null. However, we think that it does not affect our main conclusions on the evolution of dependence on host nucleotides and amino acids (and by

extension, other metabolites) and of intracellularity, as these are not based on the specific topology of those nodes.

Although perhaps not particularly important to the question of the study, I'm wondering why MarineAlpha lineages were omitted from the Alphaproteobacteria outgroup. They were mentioned to be initially collected in Supplementary Text 4, but must have been cut at some point during the various selection procedures. They represent much Alphaproteobacterial diversity and have decent-to-high levels of completeness and low contamination rates, so it seems odd to me they were removed. If the authors do decide to add them in their revision, I would recommend not adding those sensitive to phylogenetic artefacts (see Martijn et al 2018, Nature which lineages these are).

We thank for this suggestion. We initially focused on checking not fully determined relationships of alphaproteobacterial MAGs with *Rickettsiales*, and thus during this phase, for the sake of completeness, we included a number of MAG lineages, including MarineAlpha. After the preliminary analyses, we determined/confirmed that many of these lineages, including MarineAlpha, are not related to *Rickettsiales*, and we thus discarded them (also due to computational limits). We acknowledge that including a selection of MarineAlpha would have provided an even more representative dataset of *Alphaproteobacteria*. However, considering that they are not expected to change the main findings, we would prefer to keep the initial selection and take the suggestion for future studies.

ORTHOGROUP RECONSTRUCTION

Orthologous groups, or orthogroups, are defined as all genes of a given group of species that have evolved from a single ancestral gene in either the last common ancestor of that group of species or one of the more recent ancestors within that group (Gabaldon & Koonin 2013). If a pair of genes has a last common ancestor that predates the last common ancestor of the taxonomic clade in consideration, they should be placed in separate orthogroups. If we take the *Rickettsiales* as an example, an orthogroup should thus consist of all genes that have descended from a gene present in the last common *Rickettsiales* ancestor, or in one of its more recent ancestors, e.g. the ancestor of the *Anaplasmataceae* or any of the other intermediate ancestors.

The Reviewer is right in stating that orthogroups are by definition groups of orthologous genes. Actually, we purposely started not from orthologous groups, but rather from potentially larger “homology groups”. We apologise for misusing the term “orthogroup” (now replaced by “homology group” in the text) and for the unclear description of the rationale of our choice and procedure (now modified in the novel version of the manuscript, see lines 681-709 of the track-mode version, and Supplementary text 5, in track-mode).

The rationale of our choice was that we believe that trying to define REAL orthogroups in a novel dataset such as ours starting from a general database and using a pre-defined pipeline (eggNOG, NCBI COGs or any other) is inherently going to result in a number of wrong assignments.

We thus decided to start from a preliminary assignment (eggNOG-based), and proceed with dataset-specific refinements (“telescopic approach”, see below), to obtain curated “homology groups”, that in our opinion are the best type of data for phylogeny-based (including ALE) investigations and interpretations.

In detail, we considered any assignment to pre-computed clusters of orthology (such as eggNOGs) as inevitably “preliminary” from the perspective of orthology. Indeed, while eggNOGs were chosen as a convenient starting point for their high resolution and “refinement” (as compared e.g. to NCBI

COGs), they are inherently at best optimised for (a selection of) already characterised organisms, and thus will predictably fail to capture any peculiar feature of newly characterised organisms and lineages, such as those targeted in this study.

With this in mind, we aimed to test potential paralogy/orthology (or xenology) on each “homology group” by phylogeny-based methods, more informative and “tailorable” for the dataset in use. These include the “gene-tree to species-tree” reconciliation performed with ALE, which we believe would have been redundant if we had started from already fully ascertained (or presumed) orthologous groups.

We hope that, with the added edits (see lines 681-709 of the track-mode manuscript and also responses to more detailed Reviewer comments below) the rationale of our strategy will be clear and agreeable.

In this study, the span of taxa considered is that set of taxa used in the main species tree (Figure 1 / Supplementary Figure 1). I assume that this is the tree that is used by the authors when they reconcile gene trees with a species tree using ALE, although this is not explicitly mentioned in their Methods nor Supplementary Text 5. Strictly speaking, the last common ancestor of the species tree was a proteobacterium, and thus ideally, orthogroups should be reconstructed at the Proteobacteria level. However, as the representation of Beta- and Gammaproteobacteria and Magnetococcales (the outgroups) are relatively poor, orthogroups at the Alphaproteobacteria level would be good as well. In that case, however, it may be better to use only the alphaproteobacterial part of the species tree for the ALE analysis.

Yes, the main species tree (Figure 1 / Supplementary Figure 1) was used. We apologise for the lack of information, now corrected.

Regarding eggNOG assignments, we chose those for *Alphaproteobacteria* as a starting point, being more representative for our dataset. Nevertheless, given the early-diverging position of *Rickettsiales* among *Alphaproteobacteria*, we considered inappropriate to remove the outgroup gene sequences from further reconstructions such as ALE, as this could have impaired (or even prevented) the identification of ancient horizontal-gene-transfer events with *Rickettsiales* as recipients and distantly related donors (ideally such HGTs should be placed between the *Magnetococcia* and the *Beta/Gammaproteobacteria*, or within the *Beta/Gammaproteobacteria*). In order to define the “homology groups” to be investigated, all resulting assignments (including those on the outgroup) were effectively combined together, applying the “telescopic approach” (please see below further details).

The authors make use of the eggNOG database (I'm assuming v5? it is not mentioned) to reconstruct their orthogroups. The eggNOG database contains many pre-reconstructed orthogroups at many different taxonomic levels (e.g. the Bacteria level, Metazoa level, Euryarchaeota level etc etc) which are hierarchically linked. This means that for example a single Bacteria NOG can be broken down in many NOGs of one taxonomic rung lower (for example Proteobacteria, Firmicutes etc), which each in turn can be broken down into smaller NOGs (Alphaproteobacteria, Betaproteobacteria) etc etc. To reconstruct their orthogroups, the authors employ the eggno-mapper. With a predicted protein from a newly sequenced genome as a query, one can use the eggno-mapper to search the eggNOG database for a protein sequence that is likely the closest homolog of the query in the database, i.e. the seed ortholog. This seed ortholog is associated with possibly multiple NOGs of different taxonomic ranks (i.e. a protein of a *Rickettsiales* NOG is also associated with Alphaproteobacteria NOG, Proteobacteria NOG etc etc). The NOG that is most appropriate for your analysis is then assigned to the query. In other words, it is assumed that the query and the seed-ortholog have a last common ancestor that falls within the taxonomic scope of the NOG. Finally, all queries that were assigned to the same NOG are grouped within a

reconstructed orthogroup. It is unfortunately not clear what taxonomic level is used by the authors. In Supplementary Text 4 it is stated that the Alphaproteobacteria are used, but in Supplementary Text 5, it is stated that some query proteins were assigned to Metazoa and Firmicute NOGs, which clearly fall outside the Alphaproteobacteria suggesting perhaps that the root level was used.

We thank the Reviewer for the insight. Yes, we launched the eggNOG mapper on the v5 of eggNOG, via the following options (`--tax_scope 28211 --go_evidence non-electronic --target_orthologs all --seed_ortholog_evalue 0.001 --seed_ortholog_score 60 --query_cover 20 --subject_cover 50`), and selected from the output the assigned “eggNOG OGs” for the following analyses. We apologise for the information that was previously missing, now added (in track mode) in the Supplementary text 5.

The eggNOG mapping approach to reconstruct orthogroups has been used by other studies in the recent past and is in my view an appropriate and pragmatic one. However, the authors appear to run into a problem with the standard implementation of this approach. Some proteins of the newly sequenced taxa are assigned to NOGs of lineages as distant as Metazoa and Firmicutes, where (I assume) they expected them to be assigned to (Alpha)proteobacteria or Rickettsiales NOGs. They blame this supposedly false mapping to the higher sequence evolutionary rates in Rickettsiales. Though plausible, I think there are other factors at play. When a Rickettsiales query protein hits a seed-ortholog outside the Alphaproteobacteria, this could be explained in several ways: (i) this protein is new in Alphaproteobacteria (that is, compared to the Alphaproteobacteria in the eggNOG db) and has previously been only observed in other lineages. The associated gene may have been transferred from some other lineage to a (possibly) very recent ancestor of the newly sequenced Rickettsiales species. In which case, the query protein and the seed-ortholog have a last common ancestor that predates the last common Alphaproteobacteria ancestor, and they belong to separate orthologous groups. The *Aquarickettsia* protein RST68180.1 mapping to Firmicutes may be explained this way. A quick BLASTP search reveals it has a few hits in other Midichloriaceae (to which *Aquarickettsia* belong), but otherwise mostly hits Firmicutes. All queries that hit the same distant NOG could be considered descendants of a single ancestral incoming transfer event and could thus be part of the same true orthogroup. However, it may also happen that multiple independent transfers have occurred from that distant NOG into the Rickettsiales, in which case they should be in distinct orthogroups.

(ii) the seed-ortholog is in fact Alphaproteobacterial, but is a contaminant of a genome in the distant lineage and hence mislabeled as e.g. Metazoan. I strongly suspect this to be the case for the seed-ortholog of *Aquarickettsia* protein RST68169.1. A quick BLASTP search vs nr reveals that it has in fact many alphaproteobacterial homologs, but one of the top hits is to *Trichoplax adhaerens*. This placozoan is known to harbor a Rickettsial endosymbiont belonging also to Midichloriaceae (see Driscoll et al 2013, GBE). Furthermore, the Metazoa NOG 3C1KH only contains four other sequences, three of which are from the antelope genome *Pantholops hodgsonii*, a genome unfortunately notorious for its high level of contamination. The Metazoan NOGs 3BT1X and possibly 3C01C have similar red flags. For query proteins that hit contaminating seed-orthologs, the NOG assignment can not be trusted. Ideally, the best hit after eliminating contaminated seed-orthologs should be used for NOG assignment.

(iii) the query protein is in fact not Alphaproteobacterial, but a contaminant of the new Rickettsiales genomes. I do not expect this to happen, if at all, as the authors have done a very thorough job sequencing and curating their genomes and selecting MAGs with minimal contamination levels. (iv) the query protein has undergone highly elevated substitution rates and has no close relatives in the Alphaproteobacteria NOGs, such that true more distant Alphaproteobacteria relatives appear as more distant to the query protein than proteins from even more distant lineages. In this case, the query protein should be assigned an Alphaproteobacteria NOG, but are assigned to NOGs of distant lineages and would therefore be wrongly separated from the true NOG it belongs to. This is the explanation the authors use to explain their example deviant NOG assignments. I suspect however that this is a relatively minor issue, though I could be wrong. Virtually all "basal Rickettsiales" do

not have long branches, and most of the newly sequenced genomes and identified MAGs do not either. Several taxa, such as Fokinia, Lyticum, Xenolissoclinum and others of similar length may pose problems though. The authors could opt to simply omit them from the orthogroup reconstruction and prune them from the species tree. The Deianiraeaceae are unfortunately all very long branching, but are critical to understanding early Rickettsiales evolution, so they should be kept in.

Issues (i) and (ii) could for a large part be resolved simply by strictly using only the Alphaproteobacteria NOGs for orthogroup assignment and reconstruction: you make sure you don't hit any genomes contaminated with alphaproteobacterial genes, and you also make sure you don't hit possible distant gene transfer donor lineages. If such transfer-descendants have no orthologs in the current eggNOG db set of Alphaproteobacteria NOGs, i.e., they don't have hits in the Alphaproteobacteria NOGs, one can always do a de novo clustering of all unmapped queries, with Silix + Hifix or some other clustering algorithm. And as suggested, issue (iv) could be ameliorated by omitting the longest branching lineages from the analysis. I do not think the system of the eggNOG database lies at the heart of the issue the authors encountered, at least not with the discussed Aquarickettsia examples.

We thank the Reviewer for the detailed insight and suggestion. Again, we apologise for the lack clarity in the description of the rationale of our procedure (now edited lines 681-709 of the track-mode manuscript), and for the misuse of term “orthogroups” in the place of more descriptive “homology groups”.

Regarding the assignments of proteins to eggNOGs of different lineages, to our experience we considered that fast-evolving sequences in *Rickettsiales* (Option “iv” in the Reviewer’s list) a reasonably likely one in multiple potential cases. However, we fully agree with the Reviewer that multiple other reasons could lie behind such partly unexpected assignments, and we apologise that our description was incomplete (now edited in the novel version of the manuscript lines 681-709 of the track-mode manuscript, as well as, in more detail, in the Supplementary text 5, in track mode).

In any case, our strategy based on “homology groups” and on “telescopic approach” (see replies to points above and below, respectively) was designed to be robust with respect to multiple other potential explanations. We consider that this made it unnecessary to further refine the dataset case-by-case, or, as suggested by the Reviewer, to perform clustering of unmapped queries or to exclude fast-evolving sequences, which we judged “a priori” inconvenient for this study, where we aimed at a genus-level representativeness of the *Rickettsiales* dataset, to preserve the possibility to capture relevant lineage-specific features.

In particular, HGTs from distant organism (option “i”) were expressly included in our homology groups (to be tested for being xenologs by phylogeny), while assignments misled by mislabelled/contaminated reference sequences in the eggNOG database (option “ii”) would be simply circumvented by “pruning” to eggNOGs of higher taxonomic ranks by the telescopic approach (see below). Contaminations in the employed *Rickettsiales* assemblies (option “iii”) cannot be fully excluded, but, as also the Reviewer noticed, we are prone to think they are very minor, due to our initial dataset filtering.

Another suggestion would be to use the HMMER search strategy of the eggnog-mapper instead of the default DIAMOND. DIAMOND is appropriate for taxa that have close relatives in the eggNOG database, whereas HMMER is more appropriate for more distant taxa (Huerta-Cepas, 2017 GBE). Since especially the “basal Rickettsiales” are not represented and are relatively distant to the Rickettsiales that are present in eggNOG, the HMMER option seems a better choice.

We thank for the reasonable suggestion. In this work, to optimise computational efforts, we opted for the default DIAMOND, which represents the best balance between speed and memory consumption (<https://doi.org/10.1093/molbev/msab293>). Considering the “homology group” and

“telescopic” approach (see above and below, respectively), as well as the additional independent (e.g. TCDB, VCDB) and/or manually curated investigations for the main pathways/functions of interest, we believe that our results are solid enough. We will consider the suggestion for future work.

Regarding the "telescopic" approach, I have some concerns here as well. If I understand correctly (the method is not explained very well, it could be much clearer), query proteins that hit a NOG with a taxonomic path that falls outside the Rickettsiales path will in the first step be assigned to the most recent common ancestor between both paths. For example, a query protein that hit a Rhizobiales NOG would be labeled as the Alphaproteobacteria NOG that is the parent of that rhizoNOG, and for a query protein that hit a Metazoa NOG, it would be labelled as a root NOG. Then, in the second step, for all those query proteins labeled with a "root" NOG, it was checked which shallower NOGs along the Rickettsiales path (e.g. Bacteria NOGs, Proteobacteria NOGs, Alphaproteobacteria NOGs and Rickettsiales NOGs) were linked to that "root" NOG and all query proteins assigned to any of these shallower NOGs were pooled together with those assigned to the root NOG.

We believe that the Reviewer has correctly interpreted our procedure. We apologise that the description was not straightforward to follow, we have now edited to make it clearer (lines 696-709 of the track-mode manuscript, see also Supplementary text 5 in track-mode).

This seems to me a dangerous approach. A single root NOG may have multiple "daughter" Bacteria NOGs, Proteobacteria NOGs etc. This can happen if a duplication happened in some deep ancestor. For example if two Alphaproteobacteria NOGs are linked to the same root NOG, it means that a gene duplication occurred before the last common Alphaproteobacteria ancestor. A pair of query proteins with such a deep paralogy thus belong in truth to two separate orthogroups at the Alphaproteobacteria level and they may be wrongly lumped together in the same orthogroup using the described telescopic approach, in effect violating the definition of an alphaproteobacterial orthogroup.

We agree with the Reviewer on this interpretation. Actually, this was a desired outcome, in order to reach the large “homology groups”, to be directly tested by more accurate phylogeny-based methods without previous assumptions on orthology/paralogy (see also above). Again, we apologise for our wrong and misleading usage of the term “orthogroups”, which, as explained above, was now replaced by the more descriptive “homology groups”.

In summary, I believe that the problems described by the authors (exemplified with the Aquarickettsia ribosomal proteins) can be better explained by more straightforward technical phenomena than elevated evolutionary rates, and that the telescopic approach, designed to address these issues is perhaps overly complicated and brings with it its own set of issues. I would recommend the authors to address the technical issues and stick to the simpler orthogroup reconstruction method (for example as described above). I hope I have properly understood the method used by the authors and that my feedback makes sense.

We thank again the Reviewer for the sensible insights. We hope that we were now able to clarify the rationale of our strategy and why we consider it appropriate for the current study.

ALE ANCESTRAL RECONSTRUCTION

Many of the new MAGs selected for the species tree and orthogroups have good completeness estimates, but a good amount also have relatively poor ones. This may affect the ALE results if left unaccounted for. The accuracy of inferred evolutionary histories depends greatly on the

representation of the extant diversity of genes, after all. ALE has a 'fraction_missing' option, which as far as I could tell was not used by the authors.

We did not use the “fraction_missing” option, and we thank for the suggestion. At the same time, we think that it would have not been obvious how to apply it to our dataset. The degree of genome complexity is highly variable among *Rickettsiales*, even within the same family, which is reflected in the broad BUSCO completeness ranges even for selected reference *Rickettsiales* genomes (from ~60% up to over 90%, see Supplementary table 2, in the tab “BUSCO_selected_Rickettsiales_genomes”). It was also for this reason that we kept a rather “permissive” inclusion threshold of 50% for the MAGs. Accordingly, we believe that this would have made it difficult to provide sensible estimates of fractions of missing genes per species, as required for the “fraction_missing” option of ALE. Thus, we overall think that the approach used, while not ideally optimal, represents an overall adequate strategy in relation to the dataset features. We underline that ALE results did not represent the “final” conclusions for our study, being instead preliminary and instructive for selecting the successive in-depth analyses after careful inspection of the ALE results. Thus, while ALE may in principle have overestimated losses being “unaware” of incomplete MAGs, we believe that the impact on our conclusions has been likely negligible, since we were aware of this and kept it into account during our inspection of the results of ALE (and of successive analyses).

Despite the large amount of work that went into reconstructing orthogroups and the corresponding ALE analyses, they are hardly discussed in the main text. The Results paragraph of lines 182-191 could use a brief description of the used orthogroup reconstruction method and ancestral reconstruction method with ALE to allow the reader a better understanding of the analyses employed.

The Reviewer is right. As suggested, we have now added a brief description of the methods for obtaining the homology groups and for ALE in the Results (lines 197-211 of the track-mode manuscript).

For many molecular machineries (T4SS, Flagella, Type 4 pili, fhaAB, biosynthesis, transporters etc) the authors infer evolutionary histories (independent losses, horizontal acquisitions, ancestral presence/absence states etc) but do not check whether the inferred ALE histories of the corresponding orthogroups match with these interpretations. If all these inferences are available, why not make use of them?

The suggestion by the Reviewer is in principle sensible. However, for the dataset of this study we judged the “raw” ALE results not enough informative for presenting and discussing reliable inferences on each single gene for multiple reasons (see below). This was actually the rationale that prompted us to perform further in-depth analyses, rather than relying on ALE results only. Accordingly, we considered that the most appropriate usage of the ALE results was as a preliminary (but still relevant) step in our work, allowing a comprehensive and largely automated screening, instructive for selecting the most relevant genes and functions for the successive in-depth analyses. We have more explicitly underlined the preliminary usage of ALE results in our study, we apologise that it was not clearly stated before (see lines 204-210 of the track-mode manuscript).

As pointed out above, ALE inferences were made on “homology groups” rather than strict orthologs, differently from the manually curated putative orthologs employed for the successive phylogenies on amino acid and nucleotide synthesis genes. Also, we considered that, whenever possible, direct inspection of concatenated phylogenies is more informative than single-gene analyses (including reconciliation-based such as in ALE), due to the comparably higher phylogenetic signal, fundamental when inferring relationships of the predictably fast-evolving *Rickettsiales* sequences.

Such single-gene analyses could be especially misleading for short and/or highly variable genes, such as those of flagella. More specifically, for flagella and T4SS several previous extensive phylogenetic studies have already indicated their ancestral origin (e.g. 10.1093/molbev/msr159; 10.1111/1462-2920.12881; 10.1128/IAI.01384-09; 10.1371/journal.pone.0004833), and further dedicated investigations were beyond the aims of the present study.

Regarding single genes that could not be meaningfully concatenated based on related functions, such as transporters, when HGT was suspected, the *Alphaproteobacteria*-centered dataset of ALE analyses did not seem fully appropriate for conclusive inferences. For this reason, for the phylogeny of the tlc nucleotide translocases we used a set of sequences and organisms tailored on purpose in a previous study (10.1093/gbe/evx015). For amino acid transporters, the broadness of the respective gene families (also considering the inability to determine substrate specificity) impaired the possibility to build meaningful datasets for phylogeny. This point is now more clearly specified in the novel version of the manuscript (lines 361-367 of the track-mode manuscript).

PHYLOGENETIC ANALYSIS OF NUCLEOTIDE AND AMINO ACID BIOSYNTHESIS GENES

It is my understanding that ultimately the authors wish to find out whether the identified biosynthesis genes were present in the Rickettsiales ancestor or not. To check for this, one could simply identify which of the reconstructed orthogroups correspond to these genes and check the history inferred by ALE. The authors do indeed (if I understand correctly) identify the orthogroups (though this could be better explained in the Supplementary Text 5 and Methods; the way it is written could still be interpreted as eggNOG COGs/orthogroups referring to the actual eggNOGs, rather than the reconstructed orthogroups of the study. Also, SuppText 5 makes it seem as though entirely new orthogroups were constructed by BLAST search of the Biocyc reference against all proteins of all 210 taxa, which is confusing), but do not discuss the relevant ALE results. I imagine that these ALE histories had too little signal to say anything about ancestral presence / absence due to poor signal in single gene phylogenies, and that this was the reason the authors chose to infer a "concatenated" phylogeny and compare that with the species tree. This is OK, but I would like to see this explained in the manuscript, to make it more coherent.

We believe that the Reviewer has correctly interpreted our reasoning and strategy. As suggested, we added a clearer explanation to the methods (lines 723-728 of the track-mode manuscript) and supplementary material (Supplementary text 5, in track mode).

The concatenation approach is a sound solution, but can only be done if there is reason to believe that all genes concatenated together share the same underlying tree. I see that short and poorly aligned sequences and putative paralogs were removed, but this does not automatically mean the remaining sequences were all generated with the same underlying tree. Yes, they do share the same pathway supposedly, but metabolic genes are generally more prone to HGTs. Are there any strongly supported major conflicts between these single gene phylogenies? If not, than this analysis is OK. I would like to see this discussed, however.

The Reviewer is right, single-gene phylogenies should be presented and considered. We had performed and carefully inspected single-gene phylogenies prior to concatenation. Those have now been added as "Supplementary figure 10", and more explicitly presented/discussed (lines 288-290, 329-331, 733-734 of the track-mode manuscript).

Those trees were overall consistent with the concatenated organismal phylogeny and the respective concatenated pathway phylogenies. Some of those single-gene phylogenies had slightly different

topologies, and overall low supports. This was not unexpected, due to the comparably low phylogenetic signal of single genes, and was indeed the reason for the concatenation.

We agree with the Reviewer that metabolic genes (including putatively “core metabolism” such as the amino acids and nucleotide synthesis) can be subjected to HGT even in the host-associated *Rickettsiales*. Thus, we were not surprised to find a few conflicts in some genes of the cysteine and histidine synthesis pathways, which were reflected in the respective whole concatenated pathway phylogenies (Supplementary figure 9). As already mentioned in the manuscript, these may be seen as indicative of HGT with some representatives of *Rickettsiales* (but not all) as recipients (or as artefacts due to high sequence divergence).

Nevertheless, we underline that in the current study we were interested in the “main evolutionary picture” of those pathways in *Rickettsiales*, which our data clearly indicate as ancestral and, when present, (almost always) vertically inherited, or otherwise loss as a consequence of the HGT receipt of suitable transporters. In our view, such a picture may be at most “confounded”, but seems not at all “compromised” by small events pertaining single/few genes and single/few organisms, such as those potentially suggested by some of our results and by previous studies (e.g. 10.3389/fmicb.2022.867392).

It may also be interesting to see what happens when the concatenated tree is reconciled with the species tree using ALE. Does it still support the same conclusion as the comparing the concatenated tree with the species tree by eye?

In principle, we agree with the Reviewer that applying a tree reconciliation with ALE for the concatenated alignment could represent an interesting potential comparison with the concatenated tree phylogenies for the inferences on the evolutionary ODTL events. To our knowledge, such an approach, while logical, would be unusual, since ALE was conceived for reconciliation of single genes phylogenies, rather than concatenated pathways. We also believe that the high supports in the concatenated phylogenies at deep *Rickettsiales* nodes, consistent with species phylogeny, represent an already convincing evidence supporting the presented conclusions. In a hypothetical ALE reconciliation, we would expect that the majority of sampled concatenated pathway trees would have highly comparable topologies to the one already obtained, thus eventually supporting the same conclusions. Taken together, we considered that the predicted added value of the novel analyses do not outweigh their potential drawbacks, and opted not to add them.

ANALYSIS OF NUCLEOTIDE AND AMINO ACID TRANSPORTERS

The authors rather convincingly show that in those lineages where biosynthesis pathways are reduced, the transporters (tlc for nucleotides, and various others for amino acids) are more prevalent. This is a very cool result and what we would expect to see given our current understanding of endosymbiosis evolution. The authors further make the claim that this pattern supports the idea that these transporters have been acquired throughout *Rickettsiales* evolution multiple times independently (as opposed to the single-origin-with-consequent-multiple-independent-loss scenario proposed by Schon et al 2022). I agree that is what the presence / absence pattern and single gene tree suggests. But is this also reflected in the reconstructed orthogroups? If the orthogroups fulfill the condition I outlined above, each independent acquisition would correspond to a distinct orthogroup. Do the authors indeed see for example an orthogroup for tlc with only Midichloriaceae, a separate one for Deianiraceae, and three separate ones for sublineages of *Rickettsiaceae*, as Figure 3 suggests? What about the orthogroups corresponding to the amino acid transporters? If so, this would strengthen your conclusion. It may be that due to the “telescopic” approach, such orthogroups that should be distinct are merged together, in which case I am curious if they become distinct with the suggested orthogroup reconstruction approach described above.

We thank the Reviewer for the insight. Indeed, as presented in detail in the responses to other comments above, we considered that the reference eggNOGs inherently could not be fully representative for the actual orthogroups in a dataset including many novel and diverging lineages as in this study. For this reason, thanks to the “telescopic approach” we aimed for “homology groups” rather than orthogroups, to be tested on purpose by ALE and by dedicated analyses. This is the case of the tlc transporter, for which the “final” phylogeny included multiple non-alphaproteobacterial sequences, necessary for evaluating HGT and duplications events, such as the multiple putative instances herein detected.

Regarding the amino acid transporter counting approach described in Supplementary Text 5, I do not entirely understand why doing a BLASTP search against "trusted entries" is considered problematic, whereas doing a BLASTP search against the entire TCDB and then selecting matches with "trusted entries" is considered OK. Does it have something to do with the relation between E-value and database size? This should be clarified, if you wish to keep the description of your initial strategy. To keep things simple, I would honestly just describe the second search strategy, and omit the first. I don't think the second search strategy needs a particular justification.

We apologise that the description of our strategy was not clear. The reason for blasting on the whole TCDB and then selecting only queries with best hits on the “trusted” amino acid transporter entries of interest resides in the homology of multiple distant paralog transporters with different specificity present in the database. We initially restricted the blast search to the “trusted entries”, but, while inspecting the results, we noticed that, among the *Rickettsiales* sequences hitting to amino acids transporters, several displayed much higher identities with transporters of other substrates. A “simple” approach to get rid of those unspecific hits could have been to apply e-value (or identity) thresholds to restrict the blast search. However, due to high sequence variation and evolutionary distances in the *Rickettsiales*, we judged this strategy as unsuitable to optimise sensitivity and specificity. On the other hand, we empirically found convenient to employ the full TCDB, and “filter out” those *Rickettsiales* sequences having better hits on distant paralog transporters with different specificity present in the database (than the hits the same query sequences had on amino acids transporters in the same database).

In any case, following the Reviewer’s suggestion, we simplified our Supplementary text 5 (in track mode) and kept only the description of the final used strategy.

ON THE INTRACELLULARITY-LATE HYPOTHESIS

In 2019 Castelli et al put forward the hypothesis that intracellularity has evolved not once, as previously thought, but multiple times independently throughout the course of *Rickettsiales* evolution. The main lines of evidence for this include (i) the extracellular nature of *Deianiraea*, whose phylogenetic placement breaks the monophyly of known intracellular lineages and (ii) the fact that intracellular lineages (to their knowledge) never revert to extracellular lifestyles. These are in my opinion the strongest arguments and already sufficient to deduce independent origins of intracellularity (I therefore find the title of this manuscript a little odd, as it gives the impression that this is a new finding of this study).

We thank the Reviewer for the precise comment. We did not mean to overstate our findings, we apologise if this was the outcome.

Regarding physical location of extant *Rickettsiales* with respect to eukaryotes and its evolutionary implications, we agree that our previous finding of the extracellular *Deianiraea* is a highly informative discovery for the “intracellularity late” hypothesis, considering that other “unusual” and/or “early-divergent” lineages are only MAGs, with no “real” data on a quite complex lifestyle trait such as intra- vs extracellularity.

Anyhow, as we stated in the 2019 paper, at the time we considered the “intracellularity late” as a plausible hypothesis, but not yet strictly “preferable” over a traditional “intracellularity early” one. We believe that the “shift” in likelihood of the two hypothesis is now realised, thanks to the analyses of the current study, taking profit of a much larger dataset (including, besides the single and still potentially “unusual” *Deianiraea*, multiple genomically-comparable representatives of the DDG-clade,). Please see also below the answer to the following point for additional comments.

This study aims to find additional evidence for this hypothesis. It confirms the paraphyly of intracellular lineages with a phylogeny based on a greatly expanded dataset, and analyzes the gene content and their evolutionary histories of the new lineages. Unfortunately gene content is presently unable to perfectly differentiate between intra- and extracellular lifestyles. Even tlc, generally considered a strong predictor for intracellularity, is present in the extracellular *Deianiraea*, and absent in the Anaplasmataceae, endosymbiont of *Amblyomma* (the absence in the intracellular Midichloriaceae MAG is most likely explained by its mere 50.7% completeness). One of the most striking results of the study, that when aa/nt transporters are horizontally acquired, vertically inherited biosynthesis genes are lost. This is a strong marker for host-association / host-dependence (as the authors point out in the Discussion) but is not a perfect predictor of intracellular lifestyles, simply because we do not know the lifestyles of the MAG organisms that retained the biosynthesis genes and lack these transporters. It is in my opinion therefore insufficient to serve as independent support for "intracellularity-late". This observation, as well as the observed patterns for other molecular systems (secretion systems, effectors, flagella, and pili), is what we expect to see given the hypothesis, but it does not support it. This distinction is in my view important, and the authors often thread this needle correctly, but are occasionally too strong (most notably in the title and abstract, but also in lines 433/434 and 467 in the Discussion) and should be adjusted.

We thank the Reviewer for the insight. We understand the concern on some overemphasised sentences, which have now been toned down accordingly (see abstract, lines 475-476, 510-529 of the track-mode manuscript). We apologise that in multiple passages of the manuscript we were unable to rightly evidence the relevant distinctions between host-dependence/association and intracellularity and their implications. We have now edited it accordingly (lines 60-62, 73-76; 476-477; 511-531).

We fully agree also on the distinction between consistent and supportive data.

We have now modified the text to better point out more clearly in the manuscript that we also considered that obligate intracellularity is after all one of the possible outcomes for an obligate host association. Thus, we believe that the support we found for a “late” obligate host association is inherently further supportive for a “late” obligate intracellularity as well. Besides, although no single perfect predictor for obligate intracellularity can be identified, we believe that the finding of multiple consistent traits, although not directly supportive of the “intracellularity late” hypothesis, can be considered as quite strongly indicative, thus representing some kind of further “indirect” support. Such consistent traits include in particular the large scale genome consistence of the whole DDG-clade with *Deianiraea*, including biosynthetic pathways and exoproteins/adhesins, as well as the lineage-specific evolution of interactors among intracellular *Rickettsiales*.

OTHER GENERAL REMARKS

Many of the supplementary figures are difficult to read and require the reader to zoom in to read the text, but as a consequence lose sight of the overall picture. I understand that phylogenetic trees and presence-absence maps with these levels of taxon sampling can grow to unyielding large sizes, but I believe nonetheless that in many cases font sizes can easily be increased. This is particularly true for ALE results (SuppFigure2), in which also the ODTL counts are near impossible to read. It

would also be very nice (but optional) if the ODTL counts and node copy numbers could be scaled according to counts, along the lines of Figure 3 of Martijn et al 2020, Nature Communications. Though it lacks a tutorial, code for this can be found on https://github.com/maxemil/ALE-pipeline/blob/master/templates/visualize_ALE_results.py. The phylogenies of the biosynthetic pathways (SuppFigure8) also suffer from miniature fonts. If increasing the font size to a readable level makes the tree unreadable, it may work better to collapse certain clades and provide the complete trees as supplementary files.

The Reviewer is right on the poor readability of some of our supplementary files, we apologise for that. Following also suggestions by Reviewer 2 (see below), we improved the readability of our supplementary figures, including font sizes.

We also thank for the suggested script to render ALE results, while, due to our already re-formatting of the ALE results, we found more convenient to start from and improve our previous data visualisation approach. In any case,

Please also ensure that phylogenetic trees are "ordered", similar to Figtree's "Order nodes" option or the "ladderize" function in the ETE python package.

We thank for the suggestion, which we followed in all the appropriate trees in the supplementary materials (Supplementary figure 2, 9-11, 15, 16).

Regarding the heatmaps, gene counts are often "outshined" by few outliers with lots of copies, making it practically impossible to appreciate the differences between the more "regular" copy numbers. It should be possible to ignore outliers when coloring these maps.

We thank for the suggestion, we edited our heatmaps so to "mask" outliers and better show the variability ranges (see Supplementary figures 4-8).

For some Supplementary Texts, their file name does not match the content (e.g. Supp Text 4 is denoted as Supp Text 3, etc).

We apologise for the mistake, which was now corrected.

Regarding the running of single gene trees with PhyloBayes, what model was used? How many chains were run, and what was the burnin used? If multiple chains were used, did they converge?

We used the default CAT model. Due to computational limitations, we inferred a single chain per gene, for 10000 iterations, applying a burn-in of 10% (at the ALEobserve step). We added the missing information to the Supplementary text 5 (in track mode).

Though BLAST is so pervasive in modern bioinformatics, and we use the term "blasting" on the workflow, in manuscripts I think we should stick to the more formal "performing a BLASTP search using query X against a database Y" while also mentioning E-value cutoffs etc.

We agree with the Reviewer. The terms blasting/blastp/etc were edited throughout the manuscript and Supplementary texts into more formal phrasing.

Reviewer #2 (Remarks to the Author):

This manuscript examines genomic diversity and patterns of genome evolution in the Rickettsiales.

The advances include a) de novo sequencing of nine new strains b) Extraction of diverse novel MAGs that represent new taxonomic groups/diversity c) Analysis of patterns of gene loss/gain across the phylogeny. I think aspects b and c are probably most significant for the field. I did enjoy the manuscript, and certainly felt I learned new things about *Rickettsiales* evolution. It felt a 'significant' manuscript.

We thank the Reviewer for the appreciation of our work and for the helpful suggestions.

The manuscript could be improved in two ways in my opinion.

As detailed below in our point-to-point responses, we did our best to improve our manuscript according to the feedback received.

First, the manuscript could be written with stronger focus, particularly in the introduction - establishing the aims and objectives of the study clearly. In general I could understand the manuscript, but there were a number of points where expression could be improved. We apologise for the insufficient focus of the introduction and for the insufficient expression quality of some passages in our previous version of the manuscript. Following the Reviewer's recommendation, we extensively and carefully revised the manuscript to provide a stronger focus and more clear expression. Please see in particular lines 47, 60-63, 72-76, 476-477, 511-531, 556-562 in the track-mode manuscript.

Second, the manuscript suffers somewhat from 'time' - that is to say, data had to be frozen at a point in time and then analysed, but since that point quite a few high quality genomes have come out that would have been great to include. I guess this is just a thing in genomic analysis - you have to stop somewhere and the analysis can take a good year, but there are cases where we know there are more complete genomes than those studied. However, it probably doesn't impact conclusions greatly - but the authors perhaps should check that their genomes reflect others that are more complete.

We perfectly understand the Reviewer's concern on time. As stated in the manuscript, our genome data are updated to July 2021, then, as noticed by the Reviewer, the full analysis took approximately one year (bioRxiv preprint posted on 14th October 2022: <https://doi.org/10.1101/2022.10.13.511287>, first submission to this journal on 15th December 2022: NCOMMS-22-51656-T). Unavoidably, further *Rickettsiales* genomes were published in the meantime.

Following the Reviewer's suggestion, we selected recently published (up to 18th May 2023) high-quality genome assemblies from novel species closely related to ours, and performed some general checks for the presence of "key" genes, namely nucleotide synthesis and transport, amino acid synthesis (described in lines 777-787 of the track-mode manuscript). We found highly consistent results with those of previously analysed closely related genomes (see Supplementary table 2 tab "BUSCO_checking_novel_genome_ass" and "BUSCO_New_Rickettsiales_genomes", and Supplementary table 7). Regarding the completeness of our assemblies, please see answer below to another Reviewer's comment.

Other major concerns:

I have a wider concern over the quality of the genomes analysed; for the de novo genomes, all are fragmented and some have low BUSCO scores indicating elements are missing from the assembly, and it was a bit of a disappointment not to see long read technologies applied to improve quality. Fragmentation is even more true (inevitably) for the MAGs, any of which are really rather partial. We apologise that the manuscript was not clear regarding quality and completeness of the novel genomes. We added some edits to point this more clearly (see lines 93-96, 609-612 of the track-mode manuscript and below).

To our experience, for highly reduced genomes such as those of *Rickettsiales*, BUSCO scores are rarely very high, and might thus be misleading at first sight. For this reason, as comparison we

provided in Supplementary table 2 also the BUSCO scores for reference published *Rickettsiales* genomes included in our study. These data clearly show that our newly obtained genomes are in the typical range for *Rickettsiales* complete genomes, indicating their high-quality. This finding fully confirmed our expectations, considering that we obtained all those genomes from high-coverage sequencing (mostly >100x, see Supplementary text 3), and subjected them to a careful extensive manual revision, as in our previous studies (e.g., 10.1038/s41396-019-0433-9).

Besides completeness, for most of the nine novel genomes, we were also able to obtain “highly compact” and even fully “closed” assemblies, thanks to multiple strategies, including Oxford Nanopore long-read sequencing (see Supplementary table 1 and Supplementary text 3 for details).

Specifically, two genomes (*Megaira polyxenophila* and *Bandiella woodruffii*) were fully closed thanks to Oxford Nanopore, other two (*Fokinia cryptica* and *Trichorickettsia mobilis*) by PCR, and two others (*Vederia obscura* and *Lyticum sinuosum*), although not fully closed, were highly compact ($L50 \leq 2$). The other three more fragmented assemblies include the *Midichloriaceae* bacterium associated with the foraminiferan *Reticulomyxa*, assembled from short reads from a previously published study (10.1016/j.cub.2013.11.027), which nevertheless showed the highest BUSCO score among our newly obtained assemblies, and one of the highest among the “crown” *Rickettsiales* in general (Supplementary table 2, tab “BUSCO_New_Rickettsiales_genomes”).

Regarding the MAGs, as noticed by the Reviewer, on average lower BUSCO scores were inevitably lower. We chose to apply a “permissive” threshold for assembly quality filtering of MAGs, in order to be inclusive with respect to phylogenetic diversity, and in successive analyses we “carefully” took into account the potential incompleteness of the MAGs (please see replies to other Reviewer’s points below).

The authors do, cleverly in my opinion, tend to analyse across a group rather than individual genomes, but I was still left wondering whether absence of evidence was evidence of absence for key systems. However, this brings its own problem, as Figures such as Figure 2 compare completeness in single genomes with multiple genomes (and I suspect counteracting this, relatively complete genomes vs quite incomplete ones). I’d like to hear the authors validation for this.

We thank the Reviewer for the appreciation of our analysis strategy. We agree that part of the missing functions in some MAGs may be at least potentially due to incompleteness. As noticed by the Reviewer, we carefully tried to counterbalance this unavoidable potential source of by focusing on complete genomes, as well as, especially for clades including multiple MAGs, on “collective” lineage features and/or evolutionary trends with multiple occurrences (e.g. acquisition of tlc translocases, followed by loss of nucleotide synthesis). Thus, we considered those cases as highly unlikely to be misled by incompleteness (or contamination) in each single MAG. This was made clearer in the manuscript (lines 213-215 of the track-mode version).

We also apologise that in our Figures the data were not presented with sufficient clarity. In Figure 2 and 3, for the sake of readability, when respectively reporting gene/presence absence and presenting evolutionary reconstructions, we “collapsed” the members of some monophyletic clades, reporting average values per clade rather than individual ones. The choice on which clades to collapse was made after verifying that, for the main functions considered (namely nucleotide and amino acid synthesis and transport), their members showed quite homogeneous repertoires, and thus somehow “redundant” for the presentation purposes. As reported in the Figure 2 legend, values for individual organisms are available for direct inspection by readers in Supplementary Figures 8, 12.

Incompleteness also means I am not clear how useful analysis such as supplementary figure 3 is. I would recommend the first column of this figure should be BUSCO Completeness just so interpretation can be made more easily.

The Reviewer is right, completeness score is useful for a more direct interpretation of the heatmaps on gene presence/absence. We have now added it to our Supplementary Figures 4-8, 12 and to Figure 2.

I did also feel a little circularity in argument that basal were all environmental/extracellular - it feels it is asserted and then inferred at different points in the manuscript.

We apologise that our manuscript was not clear on this point, we have now rephrased it for a more careful presentation (see lines 458-474 of the track-mode manuscript).

Briefly, the status of “basal” *Rickettsiales* is inherently hypothetical, as their current members are all MAGs. In the first identification of two “basal” *Rickettsiales* families by Schön et al. (doi.org/10.1038/s41564-022-01169-x), it was proposed that they were not host-associated. Here, we investigated a much larger phylogenetic sampling, including two new “basal” *Rickettsiales* families. We found rather consistent general features of “basal” *Rickettsiales*. We thus agree that, as per their biosynthetic metabolism, “basal” *Rickettsiales* are likely able to survive in the absence of eukaryotic hosts. At the same time, we found that the distinction between “basal” and “crown” *Rickettsiales* is more nuanced for other genome traits. We thus propose a slightly updated reconstruction, suggesting that “basal” ones might be potentially involved in facultative interactions with eukaryotes.

REVIEWER COMMENTS

Reviewer #1 (Remarks to the Author):

I apologize in advance for my verbose review reports. Being verbose helps me organize my thoughts.

SPECIES TREE RECONSTRUCTION

In my original review report, I was mainly concerned about the constraints that were used during the tree search. The search was constrained such that the Pelagibacters could only branch with Rhodobacterales/Rhizobiales/Caulobacterales, and the Holosporales could only branch with Rhodospirillales. Both Pelagibacters and Holosporales are known to artefactually branch with Rickettsiales, and constraining the tree search this way would ensure that these artefacts would not occur. However, this strategy cannot guarantee that other artefacts are suppressed, as the biased sites responsible for the abovementioned artefacts remain part of the analysis. I suggested to use a more conventional site-removal strategy.

Despite my concerns, the authors have not revised their methodology. Essentially, two reasons are given:

- Loss of phylogenetic signal. Indeed, the natural consequence of removing data is the reduction of signal. Their dataset is also affected by this, as shown in their new trees shown in Supplementary Figure 2. Starting at 20% site removal, key bipartitions ((DDG,Ana),(all_others)) and ((DDG,Ana,Midi,Rick,Atha),(all_others)) drop below the 90% ultra-fast bootstrap threshold, and as more sites are removed, more bipartitions (or branches) follow suit. However, this does not necessarily mean that these branches can no longer be trusted. The main aim of the site removal strategy is to identify artefactual branches. If branches merely lose support, and are not replaced by other strongly supported branches, we can have some faith that they are real. This branches discussed above seem to behave this way, and thus we can have some faith in them.

- The unintentional introduction of new artefacts as sites are removed. As far as I am aware, this is not a phenomenon recognized by the current phylogenetic literature. Perhaps I have missed it, in which case please list me the references. But from the literature I know and my own experiences, removing sites that poorly fit the used phylogenetic model have never introduced new, strongly supported artefacts.

I have closely studied the new trees shown in Supplementary Figure 2.

On a side note, these trees are very difficult to read. Different clades are not annotated and/or colored, and branch lengths are not rendered. It was quite a chore to properly assess these trees and extract meaning out of them. The authors need to improve this in their next revision.

As expected, the Holosporales and Pelagibacterales branch artefactually with maximum support in the untrimmed (and unconstrained analysis), but are resolved as soon as 10% (in case of Pelagibacterales) or 20% (in case of the Holosporales) of the sites are removed. This shows that, for this dataset, only a small fraction of the sites need to be removed to resolve these artefacts. Unfortunately, at 20%, two key bipartitions within the Rickettsiales drop to respectively 86% and 85% support, below the 90% threshold. However, (as explained above), because these branches were significantly supported in the untrimmed analysis, and were not replaced by other strongly supported branches with more progressive trimming, we can have some faith that these branches are real.

It appears that the topologies at the class/family level are exactly the same between the constraint tree of Figure 1 / Supplementary Figure 1 and the the 20% site removal tree in Supplementary Figure 2. Since I lacked the newick files, I could not check whether the more fine-grained branching patterns are also identical.

If they are, I would suggest that the authors rewrite the narrative of their manuscript such that they select the 20% site removal tree as their species tree. This means no new analyses need to be done. But it would prevent the need to describe the constraint strategy as their tree inference strategy. While here it appears that the constraint strategy did not lead to other artefacts, I fear it sets a dangerous precedent to the field that it is OK to use constraints for inferring species trees. Constraints should strictly be used for tree topology testing. Therefore, if it is possible to present the species tree without mentioning the constraint strategy, I strongly encourage the authors to do so.

If they are not exactly identical, it would be OK with me if the authors do not change their narrative. A change in the species tree means re-running ALE for all single gene trees and re-interpreting the results, which altogether is a massive amount of work. And as the 20% trimmed tree would in that case only be slightly different from the constraint tree, it would not lead to strikingly different results. However, I would like to see a revision of the main text -at the start of paragraph line 126- that includes (1) a brief explanation of how the species tree was inferred, including an explanation of why the constraint strategy was chosen over a site-trimming strategy and (2) a note that explains that the constraint strategy itself does not ensure that the entire tree is free of artefacts. I think it is important that the field understands that constraints for species tree inferences should be avoided if possible. Then, to show that no new major artefacts were present in the constraint tree, the authors could present their Supplementary Figure 2 analysis and show that the major bifurcations in the tree remained the same.

Regarding the decision to keep Holosporales and Pelagibacters in the species tree and single gene trees, this is understandable. It would indeed be interesting to see HGT events between Holosporales and Rickettsiales, and keeping the Pelagibacters for representation sake makes sense as well. However, a note should be included in the ALE part of the main text Results section that known phylogenetic artefacts affecting the placements of Pelagibacters and Holosporales relative to Rickettsiales could affect single gene trees (this is very apparent in the single gene trees of the biosynthesis genes) and in turn the reconciliations as well. Any potential grouping of a Holosporales with a Rickettsiales in a single gene tree could be explained by a HGT, but also by a phylogenetic artefact. It should also be mentioned that

MarineAlpha lineages (and possibly other non-Rickettsiales alphaproteobacterial MAG lineages) were not included, and that this lack in representation could also affect reconciliations.

In my report, I argued that the increased taxon sampling most likely did not cause the shift from the Schon et al topology -(('DG'),(Ana,Midi))- to the one observed by the authors -(('DDG',Ana),(Midi))- . In their revision, the authors still argue that their increased taxon sampling may be responsible. There is no support this. First, this topology is, as they mention themselves, in line with most other studies with fewer taxa. Hence, adding more taxa did not alter this topology. Second, Schon et al also obtain the (('DG',Ana),(Midi)) tree in their ML untrimmed analysis. Hence, adding the Diomedesiaceae and the additional Gami's and Deia's did not alter the topology. The topology only changed when biased sites were removed. In both their own analyses and that of Schon et al, the (('DDG',Ana),(Midi)) tree fell below significance threshold. Of course, that does not mean the addition of taxa is not valuable. If we get the same tree with better taxonomic representation, our faith in that tree increases.

I do agree with the authors that whether (('DG'),(Ana,Midi)) or (('DDG',Ana),(Midi)) is correct is not yet clear and is outside the scope of this study. Still, I think that the study contributes some insight to this issue. It suggests that, regardless of the level of taxon sampling, one topology occurs with untrimmed alignments, and the other (albeit unsupported in ML analyses, but supported in Bayesian analyses) when biased sites are removed. This could be an important clue for future studies and should be made more clear.

RECONSTRUCTION OF "HOMOLOGOUS GROUPS"

In my previous report, I outlined my concerns regarding the methods used to reconstruct orthologous groups (or 'orthogroups'). One distinct orthogroup includes all genes of a given group of species that have evolved from a single ancestral gene in either the last common ancestor of that group or one of the more recent ancestors within that group. It turns out however that the authors did not intend to reconstruct orthogroups, but what they call 'homology groups'. This means that any two genes within a homology group can have a last common ancestor gene that was present in an ancestor that predates the last common ancestor of the given group of species. For example, let's say a gene in an ancient bacterial ancestor experienced a duplication. One copy was retained only in the lineage to Firmicutes, and the other was retained only in the lineage to Alphaproteobacteria and Rickettsiales. Then, the Firmicute copy was recently transferred to a Rickettsiales lineage. The Firmicute (transferred) copy and the Rickettsiales (inherited) copy belong to different orthogroups at the Alphaproteobacteria or Rickettsiales level, but belong to the same homology group. The authors should more clearly define what they mean exactly with 'homology group' to aid the reader in understanding their methodology.

I understand from the author's rebuttal that they do believe that you should ideally aim to reconstruct the true orthogroups, but that this is not possible when you use a pre-existing dataset of orthogroups as a base. A number of arguments are given:

First, they argue that this approach inherently leads to wrong assignments. It is not fully clear to me why this approach is "inherently" wrong. As far as I understand, the eggNOG-mapper was explicitly designed for identifying orthogroups of proteins of novel organisms, more or less precisely what the authors would like to do. This statement should be explained further.

Second, some Rickettsiales protein queries could be wrongly assigned to non-alphaproteobacterial (e.g. Metazoa, Firmicutes, Sphingomonadales) eggNOGs. In their revision, the authors list four scenarios that could explain this:

(i) the Rickettsiales gene was recently acquired through HGT from a non-alphaproteobacterial lineage. The firmicute and sphingomonadales hits for the Aquarickettsia protein could potentially be explained this way. With the simple eggNOG-mapper approach for orthogroup reconstruction, this gene would be entirely missed by downstream analyses and this is indeed problematic. However, it does correctly indicate that this gene does not belong to any of the pre-existing orthogroups. The conventional approach here is to de novo cluster such unmapped sequences into groups that approximate such new orthogroups. The HGT scenario is thus not necessarily an issue with the simple eggNOG-mapper approach.

(ii) high sequence evolutionary rates in Rickettsiales cause distantly related non-alphaproteobacterial sequences to have better scoring diamond hits than more closely related alphaproteobacterial sequences (they have acquired so many substitutions since their divergence with the reference eggNOG taxa that the sequences have essentially become saturated, and true close relatives appear as or more distant compared to true distant relatives). The authors write in their rebuttal that this scenario is reasonably likely in multiple potential cases. However, no examples are given. I would like to see some examples, and some indication that this is a pervasive enough issue that it is a major issue for orthogroup reconstruction. As I explained in my first report, almost none of the species tree lineages seem to have extraordinarily long branches relative to Rickettsiales and other alphaproteobacterial lineages present in the reference eggNOGs to warrant such saturation worries. In addition, I would like to re-iterate here that a more sensitive search algorithm (hmmer) could be used with the eggNOG-mapper, compared to the default diamond one, to minimize the saturation issue.

(iii) the non-alphaproteobacterial hit is a taxonomically mislabelled reference sequence (in other words, the hit is actually from an alphaproteobacteria, but this bacterium contaminated the reference genome of the non-alphaproteobacteria). This is indeed an issue, but can be circumvented by restricting the eggNOG search to alphaproteobacteria NOGs (a hit to an alphaproteobacterial ortholog is still sufficient to identify the correct orthogroup, even if it is not the best hit in the entire eggNOG database)

(iv) the protein query itself is not actually alphaproteobacterial, but a contamination. This is however most likely not an issue because the novel genomes have been thoroughly checked.

In summary, it seems to me the conventional 'eggNOG-mapper + de novo clustering unmapped queries' approach is quite adequate for the task of orthogroup reconstruction in this context.

In the rebuttal, the authors also state that the gene tree species tree reconciliation with ALE would have been redundant if they had used fully ascertained orthologous groups. I do not understand what they mean with this. If you have managed to reconstruct perfectly an orthogroup, you still do not know for each possible pair of genes inside that orthogroup what their exact relationship is. It could be an orthologous relationship (i.e. their last common ancestor was a speciation event), a paralogous one (the lca was a duplication event) or even a xenologous one (at least one of the pair experienced an HGT in its past since the lca of the pair) (see also Gabaldon & Koonin 2013, Nature Review Genetics). To be clear, not necessarily all pairwise relationships in an orthogroup are orthologous ones. One can use ALE to infer to sort out pairs of paralogs, pairs of orthologs etc etc. ALE is thus not redundant.

Due to the issues perceived by the authors (outlined above), they implement their so-called 'telescopic approach' to reconstruct homology groups. If I understand the 'telescopic' approach correctly, homology groups are created at different levels. The last common ancestor of all genes in a homology group can be either at the Root (i.e. LUCA) level, the Bacteria level, the Proteobacteria level, the Alphaproteobacteria level, or the Rickettsiales level. Homology groups at the root, bacteria and proteobacteria level may thus in effect contain multiple orthogroups at the Alphaproteobacteria and Rickettsiales levels. The homology groups are constructed with the ultimate aim to be used for gene tree species tree reconciliation analyses with ALE, to infer ancestral copy numbers, duplications, transfers, originations and losses, and as a base for later phylogenetic analyses of genes of interest.

For protein queries that, because of reference genome contamination or sequence saturation, hit non-alphaproteobacterial seed-orthologs during the preliminary eggnoG-mapping, this telescopic approach would indeed ensure that the queries group with their true orthogroup (alphaproteobacteria or rickettsiales levels) relatives in the same homology group. For queries that hit non-alphaproteobacterial seed-orthologs because of horizontal gene transfer, the approach will merge multiple true orthogroups into a single homology group.

As far as I'm aware, ALE has only been used in conjunction with orthogroups in the literature. The reconciliation method for ancestral reconstruction is still in its infancy and whether the use of homology groups that contain multiple orthogroups is appropriate here is unclear. It could lead to some unforeseen artefacts, if one is not careful. However, we can perhaps predict some behaviour with logical reasoning. In theory, if one reconstructs a tree from a homology group with multiple orthogroups, each orthogroup should form its own monophyletic group. In reality, each of these orthogroups has (by definition) a separate, independent origin in the species tree. Now the question is whether the reconciliation of this homology group tree with the species tree can actually recognize these independent origins. By definition, ALE only recognizes one 'origination' event per reconciliation. However, the other true originations (one for each orthogroup) could still be recognized as 'transfer' events or even as 'duplication' events in specific cases (see below).

I can envision several types of errors:

(i) two orthologous groups branch as sisters in the gene tree, and their independent origins are in sister branches of the species tree. Hence, the gene tree fits perfectly into the species tree by accident. The origination event is falsely inferred in the parent branch of the species tree, and both true origination events are missed.

(ii) two orthologous groups branch as sisters in the gene tree, and their independent origins happen to be on the same branch of the species tree. A single origination event is correctly inferred on this branch, but the other is missed. Instead, a false duplication event will be inferred.

(iii) two or more orthogroups branch as sisters in the gene tree, but have independent origins in unrelated areas of the species tree. The behaviour ALE is more difficult to predict, which is unfortunate because this seems to me the most frequent scenario. Even though there are multiple true origination events, only one will be inferred per reconciliation. ALE reports a 100 replicate reconciliations per gene tree-species tree comparison, and one can envision that each true origination event is recognized as such in only a subset of these 100. Its signal is thus 'diluted'. If a true origination event is not recognized as one, it may still be recognized as an incoming transfer. Each of these transfers are not representative of the true historical transfers, as their origin (from outside the species tree) is entirely missed. Recognition of a transfer origin can still be useful though, even if the source is not clear. However, each of the true origination events may still be difficult to recognize (as origination or transfer) in the ALE result, as the signal is heavily diluted among the 100 ALE replicates.

In summary, the use of homology groups with multiple orthogroups can lead to unforeseen and difficult to predict ancestral reconstruction artefacts. If the authors wish to stick to their telescopic approach, I would be OK with that as long as it is made clear to the reader what the pro's and cons of homology groups with multiple possible orthogroups are in this context, and how it affects their interpretation of the results.

ALE ANCESTRAL RECONSTRUCTION

In my original report I noted that the authors did not make use of the 'fraction_missing' option of ALE, despite the good chance that many new genomes are not complete. The authors reply in their rebuttal that assessing completeness is inaccurate for Rickettsiales, even for known complete genomes. This is a fair point. This argument should be incorporated in the next revision of the manuscript, however.

The authors also stress that their ALE analysis merely functions as a preliminary analysis, a comprehensive screening, to identify and select (sets of) genes for more detailed downstream analyses. It is however very much unclear what criterion, derived from the ALE analysis, was used for selection. Supplementary Figure 3 is referenced in the main text where the selection is made, but this figure just shows the number of inferred ancestral events and copy numbers. What patterns exactly lead the authors to select their systems of interest (secretion attachment motility, nucleotide and amino acid metabolism and transporters)? These systems were already of interest prior to this study, so in principle the ALE analysis was not necessary to select them. Furthermore, the only ALE result discussed was the

global trend of genome reduction in the Rickettsiales, something that is already well established. In other words, what was the added value of the ALE analysis?

PHYLOGENETIC ANALYSIS OF NUCLEOTIDE BIOSYNTHESIS GENES

The authors are interested in the evolutionary origin of the nucleotide biosynthesis genes. If it can be confidently inferred to be present in the Rickettsiales ancestor, it gives more weight to the idea that it was a free-living organism. To do so, the authors identified the homology groups associated with these biosynthesis genes, and concatenated their alignments into two supermatrices, one for the purine biosynthesis pathway, and one for the pyrimidine synthesis pathway. Given that these are metabolic genes, I was concerned that individual genes may have had alternate gene histories, and could therefore not be concatenated. The authors have now provided single gene trees for each of the 30 genes (15 for purine, 15 for pyrimidine) to show that their histories are largely congruent with each other, and to that of the inferred species tree.

Upon close inspection I agree that indeed, the trees mostly agree with the species tree. That is, I could not see any strongly supported major incongruencies. The one exception being the placement of the Rickettsiaceae (this is also visible in the concatenated phylogenies of Supplemental Figure 9, see discussion below). However, I have also noted some, perhaps minor, issues that should be resolved.

The *pyrF*, *pyrB*, *purL* and *purB* still include paralogs. For *pyrF*, this includes all taxa of the clade on the top of the tree with Magnetococcae and Gamibacteraceae. For *pyrB*, it is all taxa in the clade with Gamibacteraceae, Deianireaceae, Athabascaceae, Dasania, Congregibacter, Alteromonas and Enterobacter. For *purL* and *purB*, it is the outgroups. The branch connecting the ingroup with the outgroup is vastly longer than all other branches in the tree, indicating that these are most likely not orthologs of the Rickettsiales and Alphaproteobacteria in the ingroup. All these paralogs can lead to significant errors in the concatenated tree and should therefore be removed.

The adk tree lacks support values.

The authors use the homology groups to construct two types of datasets. An "all organisms" or "full organisms" (quick side note: I think it would benefit the manuscript if one name would consistently be used instead of two) and an "organism selection" or "selected organism" dataset. The former simply includes all taxa with at least one gene present, whereas the latter only includes the taxa that have at least 50% or some significant number of genes of the pathway present. The reason for this second dataset is explained as follows: some organisms displayed incomplete pathways according to their predicted gene complement. The genes that are present may either still function in that pathway, with the missing steps being filled in by other enzymes, or they may themselves have other 'non-specific' roles in other pathways. The authors regarded them as possible 'false positives'. Furthermore, incomplete pathways lead inherently to missing data in the phylogenetic datasets which the authors

believe could hamper phylogenetic accuracy. To resolve these issues, the authors select only those taxa that have a 'significant' proportion of genes of the pathway present for their second dataset.

In my opinion, this second dataset is entirely unnecessary. The biochemical role the current day genes play in their respective organisms is not relevant to the question of whether or not they were present in the Rickettsiales ancestor. What is important is that they are orthologs to the biosynthesis genes in question, and this has been confidently shown with the single gene trees. They are thus not 'false positives' in an evolutionary sense. Also, missing data in phylogenetic dataset, as far as I know, does not lead to phylogenetic inaccuracies (that is, strongly supported inaccurate topologies), only to reduction in signal. At the worst, these taxa may, because of missing data, have uncertain placements in the tree. The trees generated with the second dataset look cleaner and are easier to read, but at the cost of not using perfectly valid data. This could lead to wrong conclusions. I would simply omit this dataset and all of its associated phylogenies.

This was something that was already visible in the original manuscript. My apologies for not bringing this up in my original report.

The trees inferred from the concatenated alignments of both the purine and pyrimidine genes placed the Rickettsiaceae in a different spot relative to the species tree. Whereas in the species tree, they are placed as the first diverging lineage among the classical Rickettsiales families, in these biosynthesis trees they are more nested within a clade consisting of DDG and Anaplasmataceae (with strong support). This placement persists with progressive z-score filtering. While this would not interfere with the main conclusion that the biosynthesis genes were most likely present in the Rickettsiales ancestor, it is a major observation and worth mentioning. What could explain this phylogenetic incongruency? It may also be worth mentioning that the Pelagibacteraceae remain (most likely) artefactually placed with Rickettsiales even with the most stringent z-score filtering.

PHYLOGENETIC ANALYSIS OF AMINO ACID BIOSYNTHESIS GENES

The authors were interested in the origin and evolution of the amino acid biosynthesis genes as well. As per my request, they have included all the single gene trees in Supplementary Figure 10. I have also checked these trees carefully, and it indeed seems that most of these genes that share the same pathway are concatenable (i.e., they seem to have largely the same underlying tree). However, many gene groups seem to include possible distant paralogs or isolated cases of gene transfers. I have flagged many suspicious cases below. A good deal of them probably are indeed non-orthologs. The authors should investigate them and decide which ones are paralogs or transfers (to be removed), and which ones are orthologs with much higher substitution rates and/or artefactual placements (to be kept). It would strengthen the manuscript if the quality of these datasets were enhanced.

Serine:

- The clade consisting of Gami's and Deia's seems to be very distant from the other Rickettsiales relatives and other alphaproteobacteria, indicating they may be paralogs

Tyrosine, Tryptophan, Phenylalanine:

- *aroK*: the clade consisting of an unnamed rickettsiaceae, an unnamed deianireaceae and two *Wolbachia*'s seem very suspicious, as well as the two Magnetococcae. These two taxa are usually placed near the outgroup and have branch lengths comparable to all non-Rickettsiales taxa

- *trpD*: the clade consisting of Mitibacters, Gammaproteobacteria, Sphingo's and Parvularcula. It is very odd to see the Gammaproteo's there, as well as the Mitibacters far removed from the other Rickettsiales

- *tyrA*: the unnamed rickettsiaceae nested within the Mitibacters. Also this tree is not properly rooted.

Isoleucine, Valine, Leucine

- there is a clade consisting of Caulobacters, Sphingo's and Parvularcula that has an odd taxonomic composition, a longer branch, and is often placed relatively far away from the other non-Rickettsiales alphaproteobacteria. These may be some paralogs that perhaps have replaced the true orthologs in these taxa. The clade occurs in a number of these genes. See below some additional suspects:

- *ilvB*: Occidentia

- *ilvH*: Two unnamed Gamibacters

- *ilvC*: Two Anaplasmataceae and a Midichloriaceae

- *cimA*: Deianiraea vastatix

- *leuB*: Sphingo's in the outgroup

Proline

- *proB*: Midichloriaceae endosymbiont of *Peranema trichophorum*, Rickettsiaceae endosymbiont of *Amblyomma cajennense* Ac37b and *Ca Deianiraea vastatix*. It may be that these are contaminations of their eukaryotic hosts? I've seen cases of the *Amblyomma* genome contaminations before.

- *proA*: the same taxa, as well as the freshwater MAG, and the two Gammaproteo's

- *pro/03_COG0345@2*: the clade of Pelagibacters and Bartonella.

Cysteine

- The single gene trees seem to be very incongruent with each other. As the authors note in line 330 of Main Text, HGT may be the underlying cause for incongruent patterns. It should be noted however that the inclusion of paralogs and/or contaminating sequences can not be excluded either. In any case, concatenation of these genes is unjustified. I would therefore omit the concatenation tree in the next revision. Oddly, the authors still conclude "global" vertical inheritance, even though for Cysteine this is clearly not the case.

- One could still however deduce from the single gene trees that they were present in the Rickettsiales last common ancestor, however: the copies of Mitibacters, Jistubacters, Atha's and Ark's always branch together, implying that their last common ancestor, which is essentially the last Rickettsiales common ancestor, had them.

Histidine

- Most single gene trees seem to be somewhat congruent with each other and with that of the species tree. The authors see evidence of possible HGT, however. I suspect that this is deduced (although it is not explained) from the odd placement of the clade consisting of several Deianireaceae, Gammaproteobacteria, Caedimonas, an unnamed Gamibacteraceae, several Caulo's, Puniceispirillum, Geminicoccus and Parvularcula in the concatenated tree. It seems to me that this clade represents a distant paralog. The unusual composition, the inclusion of an outgroup, the longer branch, and the placement relatively away from close species tree relatives are strong hints. This paralog clade, or a subset of it, is visible in all histidine single gene trees

- hisG: the clade sister to the abovementioned clade most likely represents a paralog as well

Other general remarks for the single gene trees

- Rickettsia sp. GCA 015657545, Rickettsia prowazekii and Rickettsiales bacterium GCA 014132315 are not colored

- Support values are indicated at the nodes, they should be indicated at the branches. Support values indicate numerical support for bipartitions, i.e. branches of the tree, not the nodes. (this is also true for the tlc single gene tree)

- Several taxa are solely indicated by some accession numbers. Taxonomic labeling is missing. This does not seem to be an issue in the species tree.

ANALYSIS OF NUCLEOTIDE AND AMINO ACID TRANSPORTERS

In my original report I was curious whether the multiple independent origins of nucleotide and amino acid transporters was also reflected in the orthogroups. As the authors point out in their rebuttal, their

groupings of gene do not actually constitute orthogroups, but 'homology groups'. Hence, the one origin - one orthogroup rule can not be used to estimate the number of independent origins from the homology groups alone. The authors inferred the number of independent origins/acquisitions by comparing the tree inferred from the homology group with that of the species tree by eye. This is OK, but perhaps the authors could briefly explain how each independent origin was inferred from this gene tree - species tree comparison.

ON THE INTRACELLULARITY LATE HYPOTHESIS

In my original report I was concerned that the authors had concluded too much from their results. The manuscript occasionally regarded the predicted gene content of the newly discovered genomes and MAGs (specifically genes related to nt/aa biosynthesis, nt/aa transporters etc) as independent support for the 'intracellularity-late' hypothesis put forward by Castelli et al 2019. The idea being that the gene content of the basal Rickettsiales (Atha's, Ark's, Miti's & Jistu's) indicates that they are not intracellular and therefore the Rickettsiales ancestor was not intracellular, and hence the paraphyletic intracellular lineages must have had independent intracellular origins. Unfortunately gene content alone can not serve as independent evidence or support for intracellular life styles or lack thereof, because no genes (or patterns of genes) are known that perfectly correlate with intracellularity. At present, only experiments can detect intracellularity with confidence. What can be said is that the gene content of the basal lineages hints that those lineages are not intracellular, which would be in line with the intracellular-late hypothesis. It does not support it. The revised manuscript addresses this subtle but important distinction in the abstract and in lines 468/469 of the Discussion. The paragraph line 503-523 still wrongly treats gene content as support for the hypothesis, however. Calling it "indirect" support is in my opinion just confusing. This paragraph should thus be toned down.

In paragraph of line 490, an "envisioned" scenario is described, which in line 495 is than considered support for the late origin of obligate association with host. A hypothetical scenario can not serve as evidence or support for another hypothesis. This should be adjusted.

OTHER GENERAL REMARKS

In Supplementary Text 5 and the Methods, the Phylobayes model used (CAT) is still not mentioned

Reviewer #2 (Remarks to the Author):

This is a revised version of a major comparative genomic study of the Rickettsiales.

The authors have done a good job of revision in line with previous comments. I found the piece interesting also on a second reading. I now have only minor comments.

i) There are some words/expressions which aren't quite standard English, and should be revised (representativity and numerosity stuck out here).

ii) In the section on nucleotide synthesis and nucleotide transport, the authors note these are almost perfectly complementarity. I think it is worth saying they are complementary except for one case, a Midichloria with low synthetic capacity and no retrieved tlc, and this single case is of weak evidential value as it has the lowest BUSCO completeness and is a MAG. I'd also be inclined to note past literature on intracellular parasites losing nucleotide biosynthesis, e.g. in microsporidia.

iii) The loss of amino acid synthesis capacity in intracellular strains is interesting, and it may be worth drawing out a little more biology here; one of the things a microbial symbiont can do for a host is amino acid anabolism, but the results presented here suggest this is not the case with any regularity in the Rickettsiales, and may more be a trait of other microbial groups (e.g. gammaproteobacteria).

We thank again both the Reviewers for the careful evaluation of our work and the insightful suggestions, which we followed in order to improve our manuscript.

Specifically, answering Reviewer 1, we:

- Clarified the usage of the constrained tree according to the reviewer's suggestions
- Removed the ALE analysis entirely and clarified the usage of "homology groups" for manual inspection
- Expanded the explanation of the results on amino acid and nucleotide biosynthesis and transport. Specifically, we clarified the general strategy and discussed the most interesting cases, but we did not detail each one indicated by the Reviewer, as we felt the flow of the text would have suffered from such a long listing.

Finally, we answered all the minor points raised by Reviewer 2.

See the detailed point by point answers below.

REVIEWER COMMENTS

Reviewer #1 (Remarks to the Author):

I apologize in advance for my verbose review reports. Being verbose helps me organize my thoughts.

SPECIES TREE RECONSTRUCTION

In my original review report, I was mainly concerned about the constraints that were used during the tree search. The search was constrained such that the Pelagibacters could only branch with Rhodobacterales/Rhizobiales/Caulobacterales, and the Holosporales could only branch with Rhodospirillales. Both Pelagibacters and Holosporales are known to artefactually branch with Rickettsiales, and constraining the tree search this way would ensure that these artefacts would not occur. However, this strategy cannot guarantee that other artefacts are suppressed, as the biased sites responsible for the abovementioned artefacts remain part of the analysis. I suggested to use a more conventional site-removal strategy.

Despite my concerns, the authors have not revised their methodology. Essentially, two reasons are given:

- Loss of phylogenetic signal. Indeed, the natural consequence of removing data is the reduction of signal. Their dataset is also affected by this, as shown in their new trees shown in Supplementary Figure 2. Starting at 20% site removal, key bipartitions ((DDG,Ana),(all_others)) and ((DDG,Ana,Midi,Rick,Atha),(all_others)) drop below the 90% ultra-fast bootstrap threshold, and as more sites are removed, more bipartitions (or branches) follow suit. However, this does not necessarily mean that these branches can no longer be trusted. The main aim of the site removal strategy is to identify artefactual branches. If branches merely lose support, and are not replaced by other strongly supported branches, we can have some faith that they are real. This branches discussed above seem to behave this way, and thus we can have some faith in them.

- The unintentional introduction of new artefacts as sites are removed. As far as I am aware, this is not a phenomenon recognized by the current phylogenetic literature. Perhaps I have missed it, in which case please list me the references. But from the literature I know and my own experiences, removing sites that poorly fit the used phylogenetic model have never introduced new, strongly supported artefacts.

We apologise that our reasoning was not clear previously.

As discussed in the previous rebuttal letter, we observed that applying “compositional-site-trimming” approaches frequently led to low supports both in preliminary exploratory tests (not shown) and in the final dataset, which led us to reason on their suitability for the current study dataset and aims. We did not find any reference clearly demonstrating the unsuitability of site-removal approaches for tackling composition biases (which would have otherwise refrained us from using these methods at all). However, neither we could find references testing comprehensively the performance of those methods and recommending cases of usage in relation to the non-relevance of possible drawbacks (if any). This seems not surprising, since, to our knowledge, these methods have been applied in a relatively limited number of studies, however valuable. We noticed that the accuracy of these methods has been debated (<https://doi.org/10.1038/s41559-020-1239-x>, although the specific alternatives therein proposed were also debated in following reports on their own drawbacks), and, more in general, we found indications that approaches tackling compositional heterogeneity may sometimes decrease accuracy of the reconstructions (e.g. <https://doi.org/10.1093/sysbio/syac042>). All of the above prompted us to adopt a more “cautious” approach in this regards, which eventually led us to the herein presented study design, as described in the manuscript and discussed in the previous rebuttal letter.

I have closely studied the new trees shown in Supplementary Figure 2.

On a side note, these trees are very difficult to read. Different clades are not annotated and/or colored, and branch lengths are not rendered. It was quite a chore to properly assess these trees and extract meaning out of them. The authors need to improve this in their next revision.

We apologise for the poor readability of those trees, we have now edited the trees in order to improve their readability by adding branch lengths and evidencing the members of each family.

As expected, the Holosporales and Pelagibacterales branch artifactually with maximum support in the untrimmed (and unconstrained analysis), but are resolved as soon as 10% (in case of Pelagibacterales) or 20% (in case of the Holosporales) of the sites are removed. This shows that, for this dataset, only a small fraction of the sites need to be removed to resolve these artefacts. Unfortunately, at 20%, two key bipartitions within the Rickettsiales drop to respectively 86% and 85% ufbboot support, below the 90% threshold. However, (as explained above), because these branches were significantly supported in the untrimmed analysis, and were not replaced by other strongly supported branches with more progressive trimming, we can have some faith that these branches are real.

It appears that the topologies at the class/family level are exactly the same between the constraint tree of Figure 1 / Supplementary Figure 1 and the the 20% site removal tree in Supplementary Figure 2. Since I lacked the newick files, I could not check whether the more fine-grained branching patterns are also identical.

If they are, I would suggest that the authors rewrite the narrative of their manuscript such that they select the 20% site removal tree as their species tree. This means no new analyses need to be done. But it would prevent the need to describe the constraint strategy as their tree inference strategy. While here it appears that the constraint strategy did not lead to other artefacts, I fear it sets a dangerous precedent to the field that it is OK to use constraints for inferring species trees.

Constraints should strictly be used for tree topology testing. Therefore, if it is possible to present the species tree without mentioning the constraint strategy, I strongly encourage the authors to do so. If they are not exactly identical, it would be OK with me if the authors do not change their narrative. A change in the species tree means re-running ALE for all single gene trees and re-interpreting the results, which altogether is a massive amount of work. And as the 20% trimmed tree would in that case only be slightly different from the constraint tree, it would not lead to strikingly different results. However, I would like to see a revision of the main text -at the start of paragraph line 126- that includes (1) a brief explanation of how the species tree was inferred, including an explanation of why the constraint strategy was chosen over a site-trimming strategy and (2) a note that explains that the constraint strategy itself does not ensure that the entire tree is free of artefacts. I think it is important that the field understands that constraints for species tree inferences should be avoided if possible. Then, to show that no new major artefacts were present in the constraint tree, the authors could present their Supplementary Figure 2 analysis and show that the major bifurcations in the tree remained the same.

We thank the Reviewer for the suggestion. However, the 20%-trimmed tree is not 100% identical in all its branching patterns to the untrimmed and constraint tree (7 bipartitions differ, all at low taxonomic levels), which prevented us to follow the first option suggested. Thus, we followed the second option suggested, namely adding a brief explanation in the manuscript (line 125-133 of the track-mode manuscript, reported below):

“The selected phylogeny of the final dataset (Fig. 1; Supplementary figure 1) was inferred by partially constraining the topology search for the relationships between some alphaproteobacterial orders (see Methods and Supplementary text 4 for further details). This was opted in consideration of the presence of well-ascertained large-scale artefacts in alphaproteobacterial phylogeny due to compositional heterogeneity (e.g. Muñoz-Gómez et al. 2019; Martijn et al. 2018; Viklund et al. 2012)), and that, on the other hand, heterogeneous-sites trimming strategies that could counteract such artefacts would also reduce the phylogenetic information for inferring finer phylogenetic relationships (Supplementary figure 2). Thus, the employed approach allowed us to “balance” between those two aspects, but in principle the presence of additional artefacts cannot be excluded.”

Regarding the decision to keep Holosporales and Pelagibacters in the species tree and single gene trees, this is understandable. It would indeed be interesting to see HGT events between Holosporales and Rickettsiales, and keeping the Pelagibacters for representation sake makes sense as well. However, a note should be included in the ALE part of the main text Results section that known phylogenetic artefacts affecting the placements of Pelagibacters and Holosporales relative to Rickettsiales could affect single gene trees (this is very apparent in the single gene trees of the biosynthesis genes) and in turn the reconciliations as well. Any potential grouping of a Holosporales with a Rickettsiales in a single gene tree could be explained by a HGT, but also by a phylogenetic artefact. It should also be mentioned that MarineAlpha lineages (and possibly other non-Rickettsiales alphaproteobacterial MAG lineages) were not included, and that this lack in representation could also affect reconciliations.

We added the requested clarification by the Reviewer, which, for the sake of text smoothness and readability, was included in the Methods section (lines 709-715 of the track-mode manuscript, reported below):

“For the sake of phylogenetic representativeness, this final dataset included sequences from the AT-rich *Holosporales* and *Pelagibacterales* genomes, and lacked, also due to computational limits, sequences from non-*Rickettsiales* MAG-only alphaproteobacterial lineages (e.g. the MarineAlpha by Martijn et al. 2018). Accordingly, the reconstructions of the evolutionary history of *Rickettsiales* genes (see below) could have been potentially affected. Nevertheless, this was taken

into account in the interpretation of the results, and potential artefacts due to the AT-rich sequences were directly addressed (see below).”

In my report, I argued that the increased taxon sampling most likely did not cause the shift from the Schon et al topology -(('DG'),(Ana,Midi))- to the one observed by the authors - (('DDG',Ana),(Midi))- . In their revision, the authors still argue that their increased taxon sampling may be responsible. There is no support this. First, this topology is, as they mention themselves, in line with most other studies with fewer taxa. Hence, adding more taxa did not alter this topology. Second, Schon et al also obtain the (('DG',Ana),(Midi)) tree in their ML untrimmed analysis. Hence, adding the Diomedesiaceae and the additional Gami's and Deia's did not alter the topology. The topology only changed when biased sites were removed. In both their own analyses and that of Schon et al, the (('DDG',Ana),(Midi)) tree fell below significance threshold. Of course, that does not mean the addition of taxa is not valuable. If we get the same tree with better taxonomic representation, our faith in that tree increases.

I do agree with the authors that whether (('DG'),(Ana,Midi)) or (('DDG',Ana),(Midi)) is correct is not yet clear and is outside the scope of this study. Still, I think that the study contributes some insight to this issue. It suggests that, regardless of the level of taxon sampling, one topology occurs with untrimmed alignments, and the other (albeit unsupported in ML analyses, but supported in Bayesian analyses) when biased sites are removed. This could be an important clue for future studies and should be made more clear.

We thank the Reviewer for this comment. We overall agree on a potential clue for future studies, and we made it more clear in the manuscript (lines 166-170 of the track-mode manuscript, reported below). Nevertheless, we believe that, based on the available data, it is not yet possible to conclusively exclude some other factors that could influence the reconstructions of the inner relationships within (Ana, Midi, DDG). While trimming has clearly an important role, differences may also depend on other reasons, including taxon sampling, considering that the set of analyses herein performed on the larger dataset do not fully correspond to those performed by Schon et al. (in particular, due to computational constraints, we did no PMSF and no Bayesian analyses), which does not allow, for the moment, a complete comparison of the results.

“Therefore, compositional trimming could be a major reason behind those different reconstructions, which may also have been potentially significantly influenced by other factors, such as the progressive and large increase of available members of the DDG-clade in successive studies (in particular, this is the first study considering *Diomedesiaceae*)”

RECONSTRUCTION OF "HOMOLOGOUS GROUPS"

In my previous report, I outlined my concerns regarding the methods used to reconstruct orthologous groups (or 'orthogroups'). One distinct orthogroup includes all genes of a given group of species that have evolved from a single ancestral gene in either the last common ancestor of that group or one of the more recent ancestors within that group. It turns out however that the authors did not intend to reconstruct orthogroups, but what they call 'homology groups'. This means that any two genes within a homology group can have a last common ancestor gene that was present in an ancestor that predates the last common ancestor of the given group of species. For example, let's say a gene in an ancient bacterial ancestor experienced a duplication. One copy was retained only in the lineage to Firmicutes, and the other was retained only in the lineage to Alphaproteobacteria and Rickettsiales. Then, the Firmicute copy was recently transferred to a Rickettsiales lineage. The Firmicute (transferred) copy and the Rickettsiales (inherited) copy belong to different orthogroups at the Alphaproteobacteria or Rickettsiales level, but belong to the same homology group. The authors should more clearly define what they mean exactly with 'homology group' to aid the reader in understanding their methodology.

I understand from the author's rebuttal that they do believe that you should ideally aim to reconstruct the true orthogroups, but that this is not possible when you use a pre-existing dataset of orthogroups as a base. A number of arguments are given:

First, they argue that this approach inherently leads to wrong assignments. It is not fully clear to me why this approach is "inherently" wrong. As far as I understand, the eggNOG-mapper was explicitly designed for identifying orthogroups of proteins of novel organisms, more or less precisely what the authors would like to do. This statement should be explained further.

Second, some Rickettsiales protein queries could be wrongly assigned to non-alphaproteobacterial (e.g. Metazoa, Firmicutes, Sphingomonadales) eggNOGs. In their revision, the authors list four scenarios that could explain this:

(i) the Rickettsiales gene was recently acquired through HGT from a non-alphaproteobacterial lineage. The firmicute and sphingomonadales hits for the Aquarickettsia protein could potentially be explained this way. With the simple eggNOG-mapper approach for orthogroup reconstruction, this gene would be entirely missed by downstream analyses and this is indeed problematic. However, it does correctly indicate that this gene does not belong to any of the pre-existing orthogroups. The conventional approach here is to de novo cluster such unmapped sequences into groups that approximate such new orthogroups. The HGT scenario is thus not necessarily an issue with the simple eggNOG-mapper approach.

(ii) high sequence evolutionary rates in Rickettsiales cause distantly related non-alphaproteobacterial sequences to have better scoring diamond hits than more closely related alphaproteobacterial sequences (they have acquired so many substitutions since their divergence with the reference eggNOG taxa that the sequences have essentially become saturated, and true close relatives appear as or more distant compared to true distant relatives). The authors write in their rebuttal that this scenario is reasonably likely in multiple potential cases. However, no examples are given. I would like to see some examples, and some indication that this is a pervasive enough issue that it is a major issue for orthogroup reconstruction. As I explained in my first report, almost none of the species tree lineages seem to have extraordinarily long branches relative to Rickettsiales and other alphaproteobacterial lineages present in the reference eggNOGs to warrant such saturation worries. In addition, I would like to re-iterate here that a more sensitive search algorithm (hmmer) could be used with the eggNOG-mapper, compared to the default diamond one, to minimize the saturation issue.

(iii) the non-alphaproteobacterial hit is a taxonomically mislabelled reference sequence (in other words, the hit is actually from an alphaproteobacteria, but this bacterium contaminated the reference genome of the non-alphaproteobacteria). This is indeed an issue, but can be circumvented by restricting the eggNOG search to alphaproteobacteria NOGs (a hit to an alphaproteobacterial ortholog is still sufficient to identify the correct orthogroup, even if it is not the best hit in the entire eggNOG database)

(iv) the protein query itself is not actually alphaproteobacterial, but a contamination. This is however most likely not an issue because the novel genomes have been thoroughly checked. In summary, it seems to me the conventional 'eggNOG-mapper + de novo clustering unmapped queries' approach is quite adequate for the task of orthogroup reconstruction in this context.

In the rebuttal, the authors also state that the gene tree species tree reconciliation with ALE would have been redundant if they had used fully ascertained orthologous groups. I do not understand what they mean with this. If you have managed to reconstruct perfectly an orthogroup, you still do not know for each possible pair of genes inside that orthogroup what their exact relationship is. It could be an orthologous relationship (i.e. their last common ancestor was a speciation event), a paralogous one (the lca was a duplication event) or even a xenologous one (at least one of the pair experienced an HGT in its past since the lca of the pair) (see also Gabaldon & Koonin 2013, Nature Review Genetics). To be clear, not necessarily all pairwise relationships in an orthogroup are

orthologous ones. One can use ALE to infer to sort out pairs of paralogs, pairs of orthologs etc etc. ALE is thus not redundant.

Due to the issues perceived by the authors (outlined above), they implement their so-called 'telescopic approach' to reconstruct homology groups. If I understand the 'telescopic' approach correctly, homology groups are created at different levels. The last common ancestor of all genes in a homology group can be either at the Root (i.e. LUCA) level, the Bacteria level, the Proteobacteria level, the Alphaproteobacteria level, or the Rickettsiales level. Homology groups at the root, bacteria and proteobacteria level may thus in effect contain multiple orthogroups at the Alphaproteobacteria and Rickettsiales levels. The homology groups are constructed with the ultimate aim to be used for gene tree species tree reconciliation analyses with ALE, to infer ancestral copy numbers, duplications, transfers, originations and losses, and as a base for later phylogenetic analyses of genes of interest.

For protein queries that, because of reference genome contamination or sequence saturation, hit non-alphaproteobacterial seed-orthologs during the preliminary eggNOG-mapping, this telescopic approach would indeed ensure that the queries group with their true orthogroup (alphaproteobacteria or rickettsiales levels) relatives in the same homology group. For queries that hit non-alphaproteobacterial seed-orthologs because of horizontal gene transfer, the approach will merge multiple true orthogroups into a single homology group.

As far as I'm aware, ALE has only been used in conjunction with orthogroups in the literature. The reconciliation method for ancestral reconstruction is still in its infancy and whether the use of homology groups that contain multiple orthogroups is appropriate here is unclear. It could lead to some unforeseen artefacts, if one is not careful. However, we can perhaps predict some behaviour with logical reasoning. In theory, if one reconstructs a tree from a homology group with multiple orthogroups, each orthogroup should form its own monophyletic group. In reality, each of these orthogroups has (by definition) a separate, independent origin in the species tree. Now the question is whether the reconciliation of this homology group tree with the species tree can actually recognize these independent origins. By definition, ALE only recognizes one 'origination' event per reconciliation. However, the other true originations (one for each orthogroup) could still be recognized as 'transfer' events or even as 'duplication' events in specific cases (see below). I can envision several types of errors:

(i) two orthologous groups branch as sisters in the gene tree, and their independent origins are in sister branches of the species tree. Hence, the gene tree fits perfectly into the species tree by accident. The origination event is falsely inferred in the parent branch of the species tree, and both true origination events are missed.

(ii) two orthologous groups branch as sisters in the gene tree, and their independent origins happen to be on the same branch of the species tree. A single origination event is correctly inferred on this branch, but the other is missed. Instead, a false duplication event will be inferred.

(iii) two or more orthogroups branch as sisters in the gene tree, but have independent origins in unrelated areas of the species tree. The behaviour ALE is more difficult to predict, which is unfortunate because this seems to me the most frequent scenario. Even though there are multiple true origination events, only one will be inferred per reconciliation. ALE reports a 100 replicate reconciliations per gene tree-species tree comparison, and one can envision that each true origination event is recognized as such in only a subset of these 100. Its signal is thus 'diluted'. If a true origination event is not recognized as one, it may still be recognized as an incoming transfer. Each of these transfers are not representative of the true historical transfers, as their origin (from outside the species tree) is entirely missed. Recognition of a transfer origin can still be useful though, even if the source is not clear. However, each of the true origination events may still be difficult to recognize (as origination or transfer) in the ALE result, as the signal is heavily diluted among the 100 ALE replicates.

In summary, the use of homology groups with multiple orthogroups can lead to unforeseen and difficult to predict ancestral reconstruction artefacts. If the authors wish to stick to their telescopic approach, I would be OK with that as long as it is made clear to the reader what the pro's and cons of homology groups with multiple possible orthogroups are in this context, and how it affects their interpretation of the results.

We thank the Reviewer for the further detailed insight on this aspect. To our understanding, the majority of (if not all) the Reviewer's concerns on "homology groups" are related to their usage in gene-to-species tree reconciliation by ALE, rather than to "homology groups" *per se*. Accounting for this issue and for another Reviewer's argument on the limited added value of ALE analyses (see below), we opted to remove those ALE reconciliation analyses completely. The manuscript now says that the copy numbers of each "homology groups" were manually inspected, providing a preliminary comprehensive insight on the gene content evolution in *Rickettsiales*, and instructing following analyses (lines 207-210 of the track-mode manuscript, reported below, and Supplementary text 5, in tracking-mode). We believe that this adequately and "faithfully" replaces the role of inspection of ALE results as preliminary step of our workflow, as well as addressing the Reviewer's concerns.

Following the recommendations received, we also added further description of the "homology groups" and their potential advantages and disadvantages *per se* with respect to more conventional orthogroups (line 734-741 of the track-mode manuscript, reported below). Please see further comments below.

We acknowledge the Reviewer's point and experience on the overall validity of a conventional approach. Nevertheless, we would like to further explain the reasoning behind the usage of homology groups: we were very particularly concerned about HGT of any origin, in particular those with distant donors which may fall outside the taxonomic range of the dataset. We also aimed to test those and all other possibly non-ortholog cases by phylogenetic methods on selected genes in our dataset, rather than sticking to the direct assignment of each sequence performed by the eggNOG-mapper according to individual sequences similarity.

Regarding our consideration on the possible misleading effects of substitution saturation on eggNOG assignments, this is largely based on our previous experiences of examining the sequence homology (and homology-based assignments with other tools) as opposed to phylogeny in comparable datasets, chiefly when newly examining the genome sequence of *Deianiraea* (10.1038/s41396-019-0433-9). Here, due to the size of the output of eggNOG-mapper, we could not examine it fully, but we reasoned that it is quite probable that this same issue could impair the assignments in a non-negligible number of cases. This line of thinking prompted us to be cautious and adopt a strategy that could counteract this, among other, potential "confounding" factors.

We thus added the two following sentences, the first in the results section, the second in the methods section.

"For this purpose, we reconstructed "homology groups" from the annotated genes and manually inspected the copy numbers in order to get insight into their evolutionary patterns (see Methods for details on the construction of "homology groups" and the differences with respect to the original eggNOG orthogroups)"

"The aim of this step was to merge the orthogroups resulting from eggNOG assignments that, based on eggNOG itself, shared significant homology. While this step obviously led to the loss of information about distinct paralog eggNOG orthogroups, it also allowed a comprehensive overview of the homologous genes in the dataset, for a *de novo* inference on the possible duplication and/or transfer events specific to the *Rickettsiales* (including from distantly related organisms falling

outside the taxonomic range here directly investigated), as well as for coping with potential “partial” mis-assignments by eggNOG (see Supplementary text 5 for details).”

ALE ANCESTRAL RECONSTRUCTION

In my original report I noted that the authors did not make use of the 'fraction_missing' option of ALE, despite the good chance that many new genomes are not complete. The authors reply in their rebuttal that assessing completeness is inaccurate for Rickettsiales, even for known complete genomes. This is a fair point. This argument should be incorporated in the next revision of the manuscript, however.

Adding this argument to the manuscript is now unnecessary, since, accounting for other comments by the Reviewer, we have now removed the ALE analyses (see above and below).

The authors also stress that their ALE analysis merely functions as a preliminary analysis, a comprehensive screening, to identify and select (sets of) genes for more detailed downstream analyses. It is however very much unclear what criterion, derived from the ALE analysis, was used for selection. Supplementary Figure 3 is referenced in the main text where the selection is made, but this figure just shows the number of inferred ancestral events and copy numbers. What patterns exactly lead the authors to select their systems of interest (secretion attachment motility, nucleotide and amino acid metabolism and transporters)? These systems were already of interest prior to this study, so in principle the ALE analysis was not necessary to select them. Furthermore, the only ALE result discussed was the global trend of genome reduction in the Rickettsiales, something that is already well established. In other words, what was the added value of the ALE analysis?

We thank the Reviewer for this comment. While we believe that the ALE analyses would have some added value here, we agree with the Reviewer that this is limited, and we understand that they can be misleading to readers for a number of reasons. Therefore, we considered that it makes more sense to remove those analyses from our manuscript (please see lines 762-764 of the track-mode manuscript, reported below, and Supplementary text 5, in tracking-mode). As discussed also above, we believe that reporting that we carefully examined the presence/absence and copy numbers of “homology groups” can meaningfully and adequately replace the manual inspection of ALE results for its significance as a preliminary and comprehensive step in our workflow.

“The presence/absence and copy number patterns of each homology group were carefully manually inspected, in order to get a general overview of the functional repertoire of the investigated *Rickettsiales*, and to instruct more specific in-depth analyses (see below).

PHYLOGENETIC ANALYSIS OF NUCLEOTIDE BIOSYNTHESIS GENES

The authors are interested in the evolutionary origin of the nucleotide biosynthesis genes. If it can be confidently inferred to be present in the Rickettsiales ancestor, it gives more weight to the idea that it was a free-living organism. To do so, the authors identified the homology groups associated with these biosynthesis genes, and concatenated their alignments into two supermatrices, one for the purine biosynthesis pathway, and one for the pyrimidine synthesis pathway. Given that these are metabolic genes, I was concerned that individual genes may have had alternate gene histories, and could therefore not be concatenated. The authors have now provided single gene trees for each of the 30 genes (15 for purine, 15 for pyrimidine) to show that their histories are largely congruent with each other, and to that of the inferred species tree.

Upon close inspection I agree that indeed, the trees mostly agree with the species tree. That is, I could not see any strongly supported major incongruencies. The one exception being the placement of the Rickettsiaceae (this is also visible in the concatenated phylogenies of Supplemental Figure 9, see discussion below). However, I have also noted some, perhaps minor, issues that should be resolved.

The *pyrF*, *pyrB*, *purL* and *purB* still include paralogs. For *pyrF*, this includes all taxa of the clade on the top of the tree with Magnetococcae and Gamibacteraceae. For *pyrB*, it is all taxa in the clade with Gamibacteraceae, Deianireaceae, Athabascaceae, Dasania, Congregibacter, Alteromonas and Enterobacter. For *purL* and *purB*, it is the outgroups. The branch connecting the ingroup with the outgroup is vastly longer than all other branches in the tree, indicating that these are most likely not orthologs of the Rickettsiales and Alphaproteobacteria in the ingroup. All these paralogs can lead to significant errors in the concatenated tree and should therefore be removed.

For the position of *Rickettsiaceae* in concatenated trees, see the response to a more detailed comment below.

Below the Reviewer made a comparable comment on amino acid biosynthesis regarding single potentially non-orthologous sequences, and asked us to identify and remove paralogs/xenologs, and to keep fast-evolving/misplaced putative orthologs, in order to strengthen the manuscript, listing a series of cases to be investigated (see below). We will answer here to both comments, explaining why, while we appreciate the Reviewer's indication, we feel it is overall unnecessary to edit our datasets.

As discussed in the previous rebuttal, we started by the consideration that single-gene phylogenies may inevitably provide only limited signal resolution, thus we focused on producing the concatenated ones. Prior, we carefully examined our single-gene phylogenies, and removed all those cases that, in our view, clearly represented non-orthologs. We left all those cases for which we could not interpret a fully reliable inference, also to avoid unintentional biasing of the dataset. We underline that during our manual curation, in order to filter out "clear" paralogs, to the best of our efforts, we retained only sequences belonging to clades with "best hits" sequences and/or NCBI domain hits on the sequences/profiles consistent with the "desired" ortholog.

We do agree that specific cases indicated by the Reviewer (and potentially others) may represent suspected paralogs (or xenologs), but they may also be fast-evolving and/or misplaced true orthologs (especially for lineages that display high rates such as the DDG). We feel that the probability of each of the two is variable case by case. Thus, we still believe that we simply do not have enough information to interpret and decide case by case. Actually, in our view the strength of our conclusions is heightened rather than weakened by the fact that the phylogenies overall clearly indicate vertical inheritance, despite some of those few residual undetermined cases left in the final dataset may in fact be non-orthologs.

Please see also our reply below regarding the "organism selection" and its function on potential non-orthologs.

In sum, we believe that it is adequate to provide also the readers with the "concatenated" trees as well as, thanks to the previous Reviewer's report, "single gene" phylogenies, which seem consistent with our general interpretation (and the consequent evolutionary hypotheses). The readers can inspect directly all those trees to gain hints on specific cases of potential non-orthology. We have made this more clear by editing the methods section as follows (lines 793-801 of the track-mode manuscript):

"We noticed some potential "false positives", namely organisms (including *Rickettsiales*) displaying only few genes of a given pathway. The apparently missing steps might be "filled" by additional non-specific enzymes, as hypothesised for other pathways in *Rickettsiales* (Driscoll *et al.* 2017). These sequences could also represent "residual" non-clearly discernible paralogs/xenologs, with

potential distinct metabolic roles, despite the herein ascertained similarity with the target genes. True inclusion of paralogs/xenologs could mislead the inference on the concatenated alignments. Anyway, in all cases when organisms lack many genes of a pathway, the “unbalanced” availability of sites could potentially hamper the accuracy of the phylogenetic inferences.”

The adk tree lacks support values.

We apologise for this omission, now corrected.

The authors use the homology groups to construct two types of datasets. An "all organisms" or "full organisms" (quick side note: I think it would benefit the manuscript if one name would consistently be used instead of two) and an "organism selection" or "selected organism" dataset.

We thank the Reviewer for this suggestion. We have edited consistently into “organism selection” and “all organisms”, respectively.

The former simply includes all taxa with at least one gene present, whereas the latter only includes the taxa that have at least 50% or some significant number of genes of the pathway present. The reason for this second dataset is explained as follows: some organisms displayed incomplete pathways according to their predicted gene complement. The genes that are present may either still function in that pathway, with the missing steps being filled in by other enzymes, or they may themselves have other 'non-specific' roles in other pathways. The authors regarded them as possible 'false positives'. Furthermore, incomplete pathways lead inherently to missing data in the phylogenetic datasets which the authors believe could hamper phylogenetic accuracy. To resolve these issues, the authors select only those taxa that have a 'significant' proportion of genes of the pathway present for their second dataset.

In my opinion, this second dataset is entirely unnecessary. The biochemical role the current day genes play in their respective organisms is not relevant to the question of whether or not they were present in the *Rickettsiales* ancestor. What is important is that they are orthologs to the biosynthesis genes in question, and this has been confidently shown with the single gene trees. They are thus not 'false positives' in an evolutionary sense. Also, missing data in phylogenetic dataset, as far as I know, does not lead to phylogenetic inaccuracies (that is, strongly supported inaccurate topologies), only to reduction in signal. At the worst, these taxa may, because of missing data, have uncertain placements in the tree. The trees generated with the second dataset look cleaner and are easier to read, but at the cost of not using perfectly valid data. This could lead to wrong conclusions. I would simply omit this dataset and all of its associated phylogenies.

This was something that was already visible in the original manuscript. My apologies for not bringing this up in my original report.

We understand this Reviewer’s concern, and we apologise that we did not show our view on this clearly enough previously, now edited (see lines 793-805 of the track-mode manuscript, reported below).

In our view, any reduction of signal may indeed have a negative impact on the strength and clarity of the reconstructions, causing uncertain placements and possible low supports, potentially affecting also “relevant” parts of the tree, namely those pertaining the main *Rickettsiales* clade(s). We also think that there is no reason to consider the “organism selection” as non-valid, as it simply represents a subselection of the *Rickettsiales* (ideally, the most representative for the given pathway), based on an obviously arbitrary, but in our view still reasonable, criterion (see also below for additional reasons). Accordingly, we still think that our paired presentation of the results of “all organisms” and “organism section” is convenient to account to the readers of these more ample “subtle” alternatives on data selection, without an absolute preference for either (we examined and considered both, with their pros and cons, as noticed by the Reviewer and here discussed).

There is another relevant reason that prompted us to adopt the “organism selection” approach, namely the presence of potential “residual” paralogs/HGT-acquisition in the single-gene datasets. As explained above and below in response to other Reviewer’s comments, we felt that the examination of the single-gene phylogenies sometimes left uncertainties about whether certain sequences represent orthologs or not. Accordingly, we opted not to discard them abruptly, but still aimed to take this possibility into account. Interestingly, we also noticed that in many of those cases (including several of those mentioned by the Reviewer, in particular regarding the amino acids below), the “uncertain” genes belonged to organisms that otherwise lacked (most of) the other genes of the same pathway, which could be seen as supportive for a non-ortholog status. Therefore, as now explicitly stated in the manuscript (lines 793-805 of the track-mode manuscript, see below), we considered that our paired “all organisms” and “organism section” approach was convenient also to account for alternative selections due to this aspect.

“We noticed some potential “false positives”, namely organisms (including *Rickettsiales*) displaying only few genes of a given pathway. The apparently missing steps might be “filled” by additional non-specific enzymes, as hypothesised for other pathways in *Rickettsiales* (*Driscoll et al. 2017*). These sequences could also represent “residual” non-clearly discernible paralogs/xenologs, with potential distinct metabolic roles, despite the herein ascertained similarity with the target genes. True inclusion of paralogs/xenologs could mislead the inference on the concatenated alignments. Anyway, in all cases when organisms lack many genes of a pathway, the “unbalanced” availability of sites could potentially hamper the accuracy of the phylogenetic inferences. Therefore, for each concatenated alignment we opted for two alternative strategies in parallel, namely phylogeny on “all organisms” dataset, or on a “organism selection” dataset, keeping only those organisms displaying at least 50% of the included genes, or, alternatively, at least a significant proportion of selected sub-branches of the pathway (Supplementary table 6)”

The trees inferred from the concatenated alignments of both the purine and pyrimidine genes placed the Rickettsiaceae in a different spot relative to the species tree. Whereas in the species tree, they are placed as the first diverging lineage among the classical Rickettsiales families, in these biosynthesis trees they are more nested within a clade consisting of DDG and Anaplasmataceae (with strong support). This placement persists with progressive z-score filtering. While this would not interfere with the main conclusion that the biosynthesis genes were most likely present in the Rickettsiales ancestor, it is a major observation and worth mentioning. What could explain this phylogenetic incongruency?

We thank the Reviewer for this comment. It is true that the position of the *Rickettsiaceae* family within the crown *Rickettsiales* in the concatenated trees for nucleotide synthesis is not fully consistent with the organismal phylogeny (namely, sister group of the Anaplasmataceae+Midichloriaceae+DDG together). Specifically, they are frequently associated with the DDG clade in the pyrimidine trees, mostly with relatively high supports, and with the DDG alone, DDG+*Midichloriaceae*, or DDG+*Anaplasmataceae*, in each of the purine trees, mostly with relatively lower supports (i.e. below 90 UFB). We believe that a possible explanation is an artefact due to long-branch attraction (now added to the manuscript, lines 302-307 of the track-mode version, see below). Namely, the DDG sequences, evolving faster than their closer relatives belonging to *Anaplasmataceae* and *Midichloriaceae*, would have been attracted by the more “basal” *Rickettsiaceae* sequences.

“Relatively minor exceptions pertain to the inner relationships among *Rickettsiales* families, namely the placement in some trees of the DDG clade (as well as sometimes *Midichloriaceae* or *Anaplasmataceae*) as close relatives of *Rickettsiaceae*, though often not with very high support,

which potentially represents an artefact, due to the fast-evolving rate of the sequences, similarly to the respective organismal phylogeny”

It may also be worth mentioning that the Pelagibacteraceae remain (most likely) artefactually placed with Rickettsiales even with the most stringent z-score filtering.

Regarding *Pelagibacterales*, they are “nested” within the *Rickettsiales* in the purine trees, although not so stably (while at 50% trimming they are “again” nested, they are not at 30% organismal selection). This is most probably an artefact, but, even if true, it would indicate a gene transfer FROM *Rickettsiales* to *Pelagibacterales*, which is not the target of this study (conversely, it would have been if vice versa), thus is not worth a specific mention in our view. We have specified this general approach in the methods section of the manuscript (lines 826-831 of the track-mode manuscript, reported below):

“While inspecting the phylogenies, we focused on verifying the support for the monophyly of the whole *Rickettsiales* and of each *Rickettsiales* family, or alternatively to reconstructions suggesting possible HGT events with *Rickettsiales* as recipients (while potential events with *Rickettsiales* as donors, such as sequences of other lineages nested within a clade of *Rickettsiales* sequences, were not highlighted, being non-target for the aims of this study).”.

PHYLOGENETIC ANALYSIS OF AMINO ACID BIOSYNTHESIS GENES

The authors were interested in the origin and evolution of the amino acid biosynthesis genes as well. As per my request, they have included all the single gene trees in Supplementary Figure 10. I have also checked these trees carefully, and it indeed seems that most of these genes that share the same pathway are concatenable (i.e., they seem to have largely the same underlying tree). However, many gene groups seem to include possible distant paralogs or isolated cases of gene transfers. I have flagged many suspicious cases below. A good deal of them probably are indeed non-orthologs. The authors should investigate them and decide which ones are paralogs or transfers (to be removed), and which ones are orthologs with much higher substitution rates and/or artefactual placements (to be kept). It would strengthen the manuscript if the quality of these datasets were enhanced.

Please see above the answer to the same comment on the nucleotides for our view and general approach on this point. Below are reported our comments to individual cases that we believe may deserve further specifications.

Serine:

- The clade consisting of Gami's and Deia's seems to be very distant from the other *Rickettsiales* relatives and other alphaproteobacteria, indicating they may be paralogs

This is potentially true. Alternatively, given the fast-evolving rate of this organisms (compared to other *Rickettsiales*), it might be simply be an artefact due long-branch attraction, and as, such, not a clearly unambiguous indication of non-orthology in our view . We also noticed that, in several of our trees (Suppl. Fig 9) the bipartitions separating Deia+Gami from the other *Rickettsiales* have low (or very low) support (with respect to the UFB 90% threshold). We have now mentioned this in the manuscript (lines 352-375 of the track-mode manuscript, see paragraph reported below at the end of the replies to the amino acid section).

Tyrosine, Tryptophan, Phenylalanine:

- *aroK*: the clade consisting of an unnamed rickettsiaceae, an unnamed deianireaceae and two *Wolbachia*'s seem very suspicious, as well as the two *Magnetococcae*. These two taxa are usually placed near the outgroup and have branch lengths comparable to all non-*Rickettsiales* taxa

Please see above regarding the effect of the “organism selections” on potential paralogs/HGTs, as well as below concerning the “residual” cases of potential non-orthologs in non-target (i.e. non-*Rickettsiales*) taxa.

- trpD: the clade consisting of Mitibacters, Gammaproteobacteria, Sphingo's and Parvularcula. It is very odd to see the Gammaproteo's there, as well as the Mitibacters far removed from the other *Rickettsiales*

- tyrA: the unnamed rickettsiaceae nested within the Mitibacters. Also this tree is not properly rooted.

We apologise for the lack of rooting, now edited (Suppl. Fig. 10).

Isoleucine, Valine, Leucine

- there is a clade consisting of Caulobacters, Sphingo's and Parvularcula that has an odd taxonomic composition, a longer branch, and is often placed relatively far away from the other non-*Rickettsiales* alphaproteobacteria. These may be some paralogs that perhaps have replaced the true orthologs in these taxa. The clade occurs in a number of these genes.

We agree with the Reviewer in considering reasonable that those genes of non-target taxa could be potential paralogs. Nevertheless, we believe that this hypothesis should be verified by dedicated analyses, which were clearly outside of the aims of the present study, and we opted not to focus on this further.

See below some additional suspects:

- ilvIB: Occidentia

- ilvIH: Two unnamed Gamibacters

- ilvIC: Two Anaplasmataceae and a Midichloriaceae

Regarding these three cases, please see also above of the discussion on the “organism selection” approach.

- cimA: *Deianiraea vastatix*

although the branch is long, the sequence is associated to its expected closest relatives in this dataset, namely *Diomedesiaceae* and the only other *Deianiraeaceae* included.

- leuB: Sphingo's in the outgroup

Proline

- proB: Midichloriaceae endosymbiont of *Peranema trichophorum*, Rickettsiaceae endosymbiont of *Amblyomma cajennense* Ac37b and Ca *Deianiraea vastatix*. It may be that these are contaminations of their eukaryotic hosts? I've seen cases of the *Amblyomma* genome contaminations before.

- proA: the same taxa, as well as the freshwater MAG, and the two Gammaproteo's

- pro/03_COG0345@2: the clade of Pelagibacters and Bartonella.

Besides considerations on the quality of each assembly, we think that it is highly unlikely that the sequences from *Rickettsiales* assemblies mentioned by the Reviewer could be contaminations (or HGTs) from any eukaryote, since they display extremely low identities with any eukaryotic sequence (at most ~40-50%). At most, they could potentially be HGTs from other bacteria (e.g. *Legionellales*) or paralogs, although their identities with any other sequences are relatively low nevertheless (<70%). These hypothetical conditions fall within the general possibilities accounted and discussed in previous replies above, and the now edited in the manuscript (lines 352-375 of the track-mode manuscript), as reported below at the end of the replies to the amino acid section. Please see also above for our considerations on “residual” potential paralogs/HGTs in non-target taxa.

Cysteine

- The single gene trees seem to be very incongruent with each other. As the authors note in line 330

of Main Text, HGT may be the underlying cause for incongruent patterns. It should be noted however that the inclusion of paralogs and/or contaminating sequences can not be excluded either. In any case, concatenation of these genes is unjustified. I would therefore omit the concatenation tree in the next revision. Oddly, the authors still conclude "global" vertical inheritance, even though for Cysteine this is clearly not the case.

We only partly agree with the Reviewer in this case. Specifically, we agree on the potential paralogs and/or, in minority of cases, contaminants, we have now made it clear in the novel version of the manuscript (lines 352-375 of the track-mode manuscript, reported below at the end of the reply to the amino acid section). It is also true that the pattern of single-gene trees is frequently incongruent but, besides the non-monophyly of *Rickettsiales* sequences (see below), in terms of deep topology this phenomenon seems mostly restricted to the first 3 genes, which also presented low supports in most deep nodes. Thus, incongruency seems to be possibly imputable, at least partly, to insufficient signal. Accordingly, we still believe that, also in this case of cysteine synthesis, concatenation is justified and useful to address this aspect. Regarding "global" vertical inheritance, please see the answer to the following Reviewer's point.

- One could still however deduce from the single gene trees that they were present in the *Rickettsiales* last common ancestor, however: the copies of Mitibacters, Jistubacters, Atha's and Ark's always branch together, implying that their last common ancestor, which is essentially the last *Rickettsiales* common ancestor, had them.

We apologise that our phrasing on the "global" vertical inheritance was unclear/imprecise, we have now rephrased it (lines 352-375 of the track-mode manuscript, please see the text reported at the end of the replies to the amino acid section) We considered that in the concatenated trees both the main lineages outside *Rickettsiales*, and each main *Rickettsiales* lineage are mostly monophyletic. In particular, the pattern of basal *Rickettsiales* seems indicative of the presence of those genes in the last *Rickettsiales* common ancestor and its following vertical inheritance, as noticed by the Reviewer. Whereas, the presence, depending on which concatenated tree, of 1-2 monophyletic lineages of crown *Rickettsiales* sequences (with inner branching pattern overall consistent with the respective organismal phylogeny) suggests that those sequences were vertically inherited as well (possibly representing the result of a HGT or of a vertically inherited paralog).

Histidine

- Most single gene trees seem to be somewhat congruent with each other and with that of the species tree. The authors see evidence of possible HGT, however. I suspect that this is deduced (although it is not explained) from the odd placement of the clade consisting of several *Deianireaceae*, *Gammaproteobacteria*, *Caedimonas*, an unnamed *Gamibacteraceae*, several *Caulo*'s, *Puniceispirillum*, *Geminicoccus* and *Parvularcula* in the concatenated tree. It seems to me that this clade represents a distant paralog. The unusual composition, the inclusion of an outgroup, the longer branch, and the placement relatively away from close species tree relatives are strong hints. This paralog clade, or a subset of it, is visible in all histidine single gene trees

We believe that the Reviewer has correctly interpreted our thoughts here. We have now made them more explicit in the manuscript (lines 352-375 of the track-mode manuscript, reported below). Specifically, we agree that the conditions noticed by the Reviewer could be indicative of paralogy, but the scenario remains more complex here in our view. Specifically, we could envision a vertically inherited paralog, horizontal transfer(s) of orthologs or paralogs, plus, focusing on the few *Rickettsiales* involved (*Deianiraeaceae* and *Gamibacteraceae*), in principle it cannot also be excluded a long-branch attraction artefact of vertically inherited orthologs in those fast-evolving organisms.

The following text includes all the changes discussed for this section (lines 352-375 of the track-mode manuscript).

“Specifically, our phylogenies indicate an overall vertical descent of these pathways, with few exceptions listed below. The most notable case pertains to cysteine biosynthesis, in which several single-gene phylogenies are quite poorly supported, and, both in single-gene and concatenated phylogenies, “basal” and crown *Rickettsiales* form distinct clades (Supplementary figure 9; 10). Additional partial incongruences pertain to the histidine pathway trees (*Deianiraeaceae* and one *Gamibacteraceae* placed in a separate clade with respect to other *Rickettsiales*), serine pathway trees (*Deianiraeaceae* and *Gamibacteraceae* in an early-diverging position among *Alphaproteobacteria*, not always with good supports), and proline pathway trees (few *Rickettsiales* from multiple families placed in an early-diverging position among *Alphaproteobacteria*) (Supplementary figure 9; 10). These cases may be seen as indicative of possible paralogs and/or horizontal transfer events (with some *Rickettsiales* as recipients), or potentially even contaminations in single assemblies. They might also represent artefactual phylogenetic positions of true orthologs, especially when considering the fast-evolving sequences of the DDG clade.

In all such potential cases of lineage-specific acquisition of amino acid biosynthesis genes by *Rickettsiales* presented above or previously proposed (Weyandt et al 2022), at most a few (or even a single) HGT event(s) would be implied. Therefore, in all cases, the analyses presented here provide support for “general” vertical inheritance of amino acids biosynthesis in the *Rickettsiales*, mostly, though possibly non-exclusively, directly from the last common *Rickettsiales* ancestor”

- hisG: the clade sister to the abovementioned clade most likely represents a paralog as well
Please see comments above on paralogs/HGTs involving non-target taxa.

Other general remarks for the single gene trees

- *Rickettsia* sp. GCA 015657545, *Rickettsia prowazekii* and *Rickettsiales bacterium* GCA 014132315 are not colored

- Support values are indicated at the nodes, they should be indicated at the branches. Support values indicate numerical support for bipartitions, i.e. branches of the tree, not the nodes. (this is also true for the tlc single gene tree)

- Several taxa are solely indicated by some accession numbers. Taxonomic labeling is missing. This does not seem to be an issue in the species tree.

We thank for the indications on the supplementary figures’ layout, and apologise that they were incomplete/inconvenient for reading. We have now edited the figures accordingly.

ANALYSIS OF NUCLEOTIDE AND AMINO ACID TRANSPORTERS

In my original report I was curious whether the multiple independent origins of nucleotide and amino acid transporters was also reflected in the orthogroups. As the authors point out in their rebuttal, their groupings of gene do not actually constitute orthogroups, but 'homology groups'. Hence, the one origin - one orthogroup rule can not be used to estimate the number of independent origins from the homology groups alone. The authors inferred the number of independent origins/acquisitions by comparing the tree inferred from the homology group with that of the species tree by eye. This is OK, but perhaps the authors could briefly explain how each independent origin was inferred from this gene tree - species tree comparison.

ON THE INTRACELLULARITY LATE HYPOTHESIS

In my original report I was concerned that the authors had concluded too much from their results. The manuscript occasionally regarded the predicted gene content of the newly discovered genomes and MAGs (specifically genes related to nt/aa biosynthesis, nt/aa transporters etc) as independent support for the 'intracellular-late' hypothesis put forward by Castelli et al 2019. The idea being

that the gene content of the basal Rickettsiales (Atha's, Ark's, Miti's & Jistu's) indicates that they are not intracellular and therefore the Rickettsiales ancestor was not intracellular, and hence the paraphyletic intracellular lineages must have had independent intracellular origins. Unfortunately gene content alone can not serve as independent evidence or support for intracellular life styles or lack thereof, because no genes (or patterns of genes) are known that perfectly correlate with intracellularity. At present, only experiments can detect intracellularity with confidence. What can be said is that the gene content of the basal lineages hints that those lineages are not intracellular, which would be in line with the intracellular-late hypothesis. It does not support it. The revised manuscript addresses this subtle but important distinction in the abstract and in lines 468/469 of the Discussion. The paragraph line 503-523 still wrongly treats gene content as support for the hypothesis, however. Calling it "indirect" support is in my opinion just confusing. This paragraph should thus be toned down.

In paragraph of line 490, an "envisioned" scenario is described, which in line 495 is than considered support for the late origin of obligate association with host. A hypothetical scenario can not serve as evidence or support for another hypothesis. This should be adjusted.

We thank the Reviewer for this comment, which we partly agree with. We apologise that our phrasing was not yet clear on this point, we have now edited the manuscript to better convey our view, also accounting for the Reviewer's point (lines 549-568 of the track-mode manuscript, reported below).

We fully support the Reviewer's statement on the only hypothetical lifestyle of the "basal" *Rickettsiales*, as these are only MAGs (we also explicitly accounted for this concept in multiple sections of the text. See lines 528-534 and 601-606 of the track-mode manuscript). However, this is not the foundation of our inference on the "late" host-association and intracellularity. In case these "basal" members are truly not intracellular, this would tell little on the "early" or "late" evolution of intracellularity among the monophyletic crown *Rickettsiales*, since in any case we can expect that any obligate intracellular bacterium will have some close or distant relative that is not. Our point here is different.

We started from inferences on metabolic traits of *Rickettsiales*. Accordingly, with (in our view) satisfying reliability even for MAGs, our data and evolutionary reconstructions indicate that dependence on the host evolved "late" and independently among crown *Rickettsiales*. We believe this to be our main finding. Then, we simply reasoned that, by definition, obligate intracellular bacteria are a subset of those that are obligatorily host-associated (at most, they are coincident, if there are no obligatorily host-associated bacteria that are not obligatorily intracellular). Thus, in other words, inferring a "late" obligate host-association also implies inferring a "late" intracellularity (while not vice versa), although it does not tell anything about the potential existence of obligatorily host-associated that are not (obligatorily) intracellular.

In sum, we believe that our novel data represent additional support for the intracellularity "late" hypothesis, independent from our previous discovery of the extracellular and obligatorily host-associated *Deianiraea*. We also underline that the existence of at least one extracellular and phylogenetically-derived crown *Rickettsiales* bacterium was the first basis to infer the intracellularity late hypothesis, but is not "*per se*" implied/required by that hypothesis, which, in principle, could have been newly inferred on the basis of the herein presented genomic analyses (while obviously the joint evidence of *Deianiraea* reinforces this scenario and allows to add further aspects).

In any case, we acknowledge the Reviewer's comment that the inherently uncertain intracellular/extracellular status of MAGs (not for what concerns basal ones, but for the members of the DDG-clade other than *Deianiraea*) does not allow to directly infer further on their evolution.

"Obligate intracellularity is a well-documented condition in most of the characterised crown *Rickettsiales*, namely the non-monophyletic assemblage of *Rickettsiaceae*, *Midichloriaceae*, and *Anaplasmataceae*. Conversely, we previously showed that *Deianiraea*⁸, while obligatorily host-

associated, is not intracellular, providing the first basis to infer a possible alternative “intracellularity late” hypothesis for *Rickettsiales*. Obligate intracellular bacteria are inherently obligatorily host-associated, and, as such, intracellularity is one among the possible features evolved by bacteria living in obligate host association. Thus, the data and analyses above supporting a “late” obligate host-association similarly represent additional support for the previously proposed⁸ “intracellularity late” hypothesis. Accordingly, it seems most probable that obligate intracellularity would have evolved multiple independent times and with differential features only in some of the obligatorily host-associated crown *Rickettsiales*. The members of the DDG clade other than *Deianiraea*, all represented by MAGs, showed quite comparable genome features to this extracellular bacterium (e.g. metabolic pathways and putative adhesins such as those of the exoprotein family; Supplementary figure 7, 8, 9, 10, 12), and, based on genomics, could be deemed as extracellular as well, thus being consistent with such a scenario.”

OTHER GENERAL REMARKS

In Supplementary Text 5 and the Methods, the Phylobayes model used (CAT) is still not mentioned

We apologise for this inaccuracy. In any case, since this analysis was part of the ALE workflow, it has now been removed from the manuscript, accounting for other Reviewer’s comments (please see above).

Reviewer #2 (Remarks to the Author):

This is a revised version of a major comparative genomic study of the Rickettsiales.

The authors have done a good job of revision in line with previous comments. I found the piece interesting also on a second reading. I now have only minor comments.

i) There are some words/expressions which aren't quite standard English, and should be revised (representativity and numerosity stuck out here).

We apologise for our incorrect/non-idiomatic expressions, we have carefully revised the manuscript according to the suggestion, for example changing “representativity” into “representativeness”, and “numerosity” into “number”.

ii) In the section on nucleotide synthesis and nucleotide transport, the authors note these are almost perfectly complementarity. I think it is worth saying they are complementary except for one case, a *Midichloria* with low synthetic capacity and no retrieved tlc, and this single case is of weak evidential value as it has the lowest BUSCO completeness and is a MAG.

Following the Reviewer’s recommendation, we added to the manuscript the explanation on the *Midichloriaceae* MAG (lines 312-316 of the track-mode manuscript):

“Indeed, the presence/absence pattern of tlc nucleotide translocases almost perfectly inversely correlates with nucleotide biosynthesis (Fig. 2; Supplementary figure 8), with as noticeable exception a *Midichloriaceae* bacterium MAG (GCA_013288625) that is devoid/depleted in both, but also presents a low BUSCO estimated completeness”.

I'd also be inclined to note past literature on intracellular parasites losing nucleotide biosynthesis, e.g. in microsporidia.

We thank the Reviewer for this suggestion. Indeed, it is intriguing to note the convergent evolution of some metabolic traits in multiple lineages of bacterial and eukaryotic intracellular parasites (including microsporidia and others, e.g. *Leishmania*). We now mention this in the revised manuscript (lines 434-439 of the track-mode manuscript):

“It is a generally accepted notion that metabolic dependence on the hosts is a key feature in obligate associations such as those involving multiple independent lineages of bacterial and eukaryotic intracellular parasites (Muñoz-Gómez et al. 2019; Dean et al. 2016; Carter et al. 2008), including the *Rickettsiales* (Driscoll et al. 2017; Min et al. 2008; Schön et al. 2022; Castelli et al. 2019), that evolved as the consequence of the possibility to efficiently acquire metabolites (including precursors and, for energy, ATP)”

iii) The loss of amino acid synthesis capacity in intracellular strains is interesting, and it may be worth drawing out a little more biology here; one of the things a microbial symbiont can do for a host is amino acid anabolism, but the results presented here suggest this is not the case with any regularity in the *Rickettsiales*, and may more be a trait of other microbial groups (e.g. *gammaproteobacteria*).

We thank the Reviewer for the insight, which we now considered for the revision of the manuscript (lines 591-594 of the track-mode manuscript, reported below). Actually, based on the present study and on multiple previous studies on *Rickettsiales*, it is suggested that, in many cases, they do not do anything *for* the host, rather they parasitically exploit it for their own benefit, and were for this reason included in “professional symbionts” (10.1016/j.cub.2021.05.049). While this condition does not preclude the evolution of beneficial roles for the hosts (although probably unlinked with amino acids), such as *Wolbachia* in filarial nematodes and in some arthropods, it overall differentiates *Rickettsiales* from nutritional symbionts providing e.g. amino acids to their insect hosts. These encompass multiple *Gammaproteobacteria*, as well as representatives of several other bacterial lineages, including the alphaproteobacterium “*Ca. Hodgkinia cicadicola*”.

“We can also observe significant differences in the metabolic interchanges with their hosts, with nutritional symbionts providing metabolites such as amino acids to their hosts, while *Rickettsiales* (and other “professional symbionts”) “stealing” the same metabolites from the hosts.”

REVIEWER COMMENTS

Reviewer #1 (Remarks to the Author):

SPECIES TREE RECONSTRUCTION

-SITE REMOVAL METHODS-

In my previous report I provided critique on the assertion of the authors that removal of heterogeneous/biased sites can unintentionally lead to new artefacts. I stated that this was not a phenomenon recognized by the current phylogenetic literature. The author's replied in their rebuttal that on the one hand they could indeed "not find any reference clearly demonstrating the unsuitability of site-removal approaches for tackling composition biases", but on the other hand could also not find references that comprehensively tested the performance of said site removal methods. Indeed, to my knowledge there have not been in any studies where site-removal methods have been tested on simulated datasets, where the true tree is known. Such a study would indeed be very valuable to the deep phylogenomics community. However, there have been several studies that have shown that site-removal can lead to dramatic improvements in model fit for empirical datasets. In Bayesian methods for example, the ability of the model (with parameter settings optimized for the dataset at hand) to capture across-taxa and across-site heterogeneity adequately can be assessed with posterior-predictive tests. In Nesnidal et al (2010), Zaremba-Niedzwiedzka et al (2017), Martijn et al (2018), Martijn et al (2020), removing the most biased sites have led to dramatic improvements in model fit. Even Fan et al (2020) (cited by the authors in their rebuttal), who attempt to discredit site-removal methods, show in their Figure 1 that all investigated site-removal methods drastically improve model fit. Trees with better model fit should always be preferred over trees with poorer model fit. Moreover, in Munoz-Gomez et al (2021), a tree reconstructed after removing the most AT/GC biased sites was strongly preferred by a more realistic model of evolution over a tree reconstructed from an untreated dataset. Hence, with the current knowledge, site-removal methods appear to be a powerful tool to improve model fit and disentangle phylogenetic artefacts, with no known side effects that induce new artefacts. Finally, I would like to comment that the Foster et al (2022) paper cited by the authors purely investigates recoding strategies, and not site-removal strategies. These are vastly different methodologies, each with their own intricacies. Hence, that study can not be used as an argument against site removal strategies.

-THE CONSTRAINT TREE-

In my previous report I strongly encouraged the authors to omit the constraint tree in favor of the 20% trimming tree, in case both trees turned out to be identical. Constraint trees should strictly be used for tree topology testing. However, if the trees did not turn out to be identical, this would mean that the authors would have to do a complete rerun of the computationally and labour intensive ALE analysis, now with the 20% trimming tree. As both species tree would only be minimally different, the new ALE results would not be too different either. In that case, I stated I would be OK with leaving in the

constraint tree. As it turned out, the trees were not exactly identical (only 7 different bipartitions). Hence, the authors kept their constraint tree.

When I wrote this comment, I expected the authors to keep their ALE analysis. However, due to other issues, the authors have decided to completely omit their ALE analysis from the manuscript. This means that, in my view, there is no more reason to keep the constraint tree. As explained above and earlier reports, using constraints to infer species trees should be avoided if possible. The authors seem to be weary of site removal strategies, mainly because in their view they (1) reduce phylogenetic signal and can lead to unsupported bipartitions, and (2) may lead to unintentional new phylogenetic artefacts. However, as I explained in my previous reports, bipartitions that are significantly supported w/o site removal, but lose support as sites are removed can still be trusted as long as they are not replaced by other significantly supported bipartitions. In addition, while total phylogenetic signal may indeed decrease, the signal-to-noise ratio (where noise is responsible for systemic errors) typically increases substantially. And as explained above, site removal methods have -as of yet- never led to new artefacts. It seems to me that in their desire to be "cautious" (in and of itself a good idea), the authors have actually ended up choosing a less cautious approach, that is, using the constraints.

I therefore insistently recommend that the authors choose the 20% trimmed tree to show as their main species tree, rather than the constraint tree, for the reasons above, but also to prevent setting a dangerous precedent. This should not alter any of their conclusions, nor force them to redo any analyses.

-THE PLACEMENT OF DDG RELATIVE TO MIDI AND ANA-

The authors observe that their obtained topology is largely in line with previously published Rickettsiales trees, with the exception of the placement of the 'DDG' clade. They recover (('DDG',Ana),(Midi)), whereas Schon et al recovered (('DG'),(Ana,Midi)). They highlight this difference and speculate on what could cause the difference. They point out in their manuscript that Schon et al also recovered (('DG',Ana),(Midi)), but that this topology lost support and was replaced by (('DG'),(Ana,Midi)) when compositional trimming was applied. Currently, the way this section is written, the readers are led to believe that Schon et al obtained a supported (('DG',Ana),(Midi)) tree using a ML analyses based on a compositional trimmed dataset. It should be made clear that this topology was only significantly supported with the Bayesian inference, and that the Schon et al ML trees in fact show a similar pattern as the authors ML trees, namely that support for (('DDG/DG',Ana),(Midi)) is lost and not replaced by another significantly supported topology when compositional heterogeneous sites are removed.

In my previous report I explained that the increased taxon sampling was most likely not responsible for the shift from (('DG'),(Ana,Midi)) to (('DDG',Ana),(Midi)). However, the authors still consider it a possibility in their revision. I must insist here that there is no sign or hint that additional taxa are responsible for the topological shift. In fact, recent ML analyses (Castelli et al 2019; Schon et al 2022) already showed (('DG/D',Ana),(Midi)) trees. (('DG'),(Ana,Midi)) was only recovered by Schon et al with a combination of compositional trimming and Bayesian tree inference. In other words, compared to

similar ML analyses (albeit not perfectly comparable, due to slight alterations in evolutionary models and algorithms), there was no topological shift. Additional taxa can not be responsible for a shift in topology, because there was no shift to begin with. Of course, it is possible that if the authors were to repeat their ML analyses without the new taxa, a topology incongruent with (('DDG',Ana),(Midi)) (be it - (('DG'),(Ana,Midi))- or something else) would be recovered. Actually, if the authors would like to back up their statement, this should be a fairly straightforward analysis to carry out. However, in light of the currently available ML analyses, this seems quite unlikely to me.

RECONSTRUCTION OF HOMOLOGOUS GROUPS

In my previous report I requested that the authors should more clearly define what they mean exactly with the concept of a 'homology group', to aid the reader in understanding their methodology. The current manuscript provides an operational description of homology groups, that is, how they are reconstructed, but not a biological definition, that is, how each gene is related to each other gene in the group. It should be made more clear that any pair of genes within a group of sequences (that aims to approximate a true homology group) does not necessarily have a last common ancestor that is younger than the last common ancestor of the given group of species. Being clear on the definition of evolutionary datasets like this aids the reader to follow and understand the analyses better.

PHYLOGENETIC ANALYSES OF NUCLEOTIDE AND AMINO ACID BIOSYNTHESIS GENES

In my previous report I outlined a fair number of sequences in the datasets of nucleotide and amino acid biosynthesis genes that were suspicious of being (out-)paralogs or (out-)xenologs. Such sequences can at best be not informative on the presence/absence state of a gene in a particular ancestor and at worst interfere with the accuracy of trees inferred from concatenated datasets. The authors have decided to keep of all these suspicious sequences in their datasets. They argue that for all these sequences, it is not 100% clear whether these indeed represent (out-)paralogs and/or (out-)xenologs or fast-evolving/misplaced true orthologs. Unintentional removal of true orthologs may "bias" the dataset. I disagree with this line of reasoning. First, in my view many of the sequences that I've highlighted are clear non-orthologs for one reason or another (see below as well as previous report), not mere edge coin-toss cases. Second, I do not see how unintentional removal of a true ortholog biases the dataset. The only thing that happens is that the dataset loses a little bit of signal. What biases a dataset is the inclusion of non-orthologs! Hence, removing a suspicious sequence in the worst case (i.e. removing a true ortholog) leads to a small reduction in signal, whereas the best case (i.e. removing a non-ortholog) removes a relative large source of noise. Finally, the authors argue that the fact that their concatenated trees reflect vertical inheritance despite the likely inclusion of non-orthologs strengthens rather than weakens the weight of their conclusions. This in my view is flawed reasoning as well. The weight of your conclusion should depend on the quality of the tree, and hence the quality of the dataset, regardless of what the shape of the inferred tree is. In other words, the tree inferred from a dataset that includes

known sources of noise should be taken with a grain of salt, regardless of what the topology of that tree is.

On a side note, the authors mention that, "in order to filter out clear paralogs, (...), we retained only sequences belonging to clades with best hits sequences and/or NCBI domain hits on the sequences/profiles consistent with the desired ortholog". I might have missed it but I could not find this curation step anywhere in the manuscript. Please include it.

-CLEAR CASES OF NON-ORTHOLOGS-

pyrB: The clade consisting of Gami's, a Deia, two Atha's, a Jistu and four Gammaproteobacteria.

- The entire clade is maximally supported and consists of taxa with branches much longer relative to the rest of the tree, and is situated on a long branch itself. This is suggestive of either lba or a set of non-orthologs.
- The clade contains Gami's and a Jistu that are separated from their other family members by several maximum or significantly supported branches. This is suggestive of non-orthologs only.
- The clade contains Gammaproteobacteria outgroups. This is a strong sign that this clade is a part of another orthologous group (i.e. non-orthologs relative the other sequences in the pyrB group)

purL and purB: The outgroup consisting of Beta, Gamma and Magnetococcae

- The branch separating the Alphaproteobacteria and the outgroups is exorbitantly extremely long. Such a branch length can not be explained by simple accelerated or fast-evolving sequences. The most likely explanation is that in- and outgroup are homologous to one another, but diverged long before the last common Proteobacteria ancestor. One copy was retained in Alpha's but lost in outgroups, and the other copy vice versa. It may also be that the outgroup sequences and the ingroup sequences share some homologous domains, but that other domains not shared between in- and outgroup were falsely aligned with one another, leading to such extreme branch lengths. Either way, both gene datasets must be curated.

serB: The clade consisting of Gami's and a Deia

- The branch separating this clade from all other sequences is maximally supported and significantly longer than all other sequences. Much longer than can be explained by higher evolutionary rates as suggested by the species tree and many other single gene trees. These sequences most likely diverged from the other sequences in the group prior to the last common ancestor of the taxa in this group.

By extension, the sequences corresponding to these taxa in the serA and serC are highly suspicious too. In serA in particular, this clade is separated from the other Rickettsiales sequences by another maximally supported, long branch.

hisI: The clade containing Geminicoccus, several Caulo's, Gammaproteobacteria, a Gami and several Deia's

- I highlighted this clade in my previous report as well. The author's replied that in their view, the placement of this clade could be explained by vertically inherited paralogs (i.e. in-paralogs), horizontal transfers, or long branch attraction induced by fast-evolving but vertically inherited orthologous sequences. I disagree. First, the taxa involved are typically not fast-evolving, with perhaps the exception of the Gami's and to a greater extent the Deia's. Caulobacters, Gammaproteobacteria and Geminicoccus are generally not fast-evolving, nor are they involved in long-branch artefacts. Second, the odd placement and odd composition of this clade could indeed be explained by multiple horizontal gene transfers, in which case they should be removed anyway. Third, a duplication after the divergence of the last common ancestor of all taxa in this dataset does not explain the Gammaproteobacteria inside this clade. In my view, this clade is a clear case of a non-ortholog clade for one more reason: the Caulobacterales representatives and the Gamibacter are separated from the other family members by several significantly supported branches. This pattern is much more likely to be explained by this clade being an out-paralog. I.e. it can trace its common ancestry with the other sequences to a duplication event that predates the last common ancestor of all taxa in the dataset. This non-ortholog clade is also present in all other histidine gene datasets and should be removed.

hisG: the abovementioned clade as well as its sister clade

- The placement of this clade should be re-evaluated after the removal of the clade discussed above, but in the current single gene tree I find it highly suspicious of being yet another non-ortholog clade. This is mainly motivated by the much longer branches of the taxa in this clade relative to the other sequences, with all of them situated on a maximum supported long branch. With exception of the Pelagibacters, none of these taxa are typically involved in any known long branch attraction artefacts. In addition, the placement of Magnetococcales in this clade is highly reminiscent of a typical Alphaproteobacteria species tree, suggesting that this clade diverged from the other sequences prior to the last common ancestor of all taxa in this dataset.

ilvB: the clade consisting of occidentia, citromicrobium, parvularcula and several caulobacterales

- This suspicious clade is also present in several other gene trees (ilvH, ilvD, leuB) and was also highlighted in my previous report. The authors replied that whether these taxa were non-orthologs or not was outside the scope of the study and should be verified with dedicated analyses. I argue that it is very much inside the scope of your study. The conclusions drawn from observations depend on the quality of the trees and hence the quality of the dataset. These sequences are, in my view, clear non-orthologs and thus reduce the quality of the dataset, and that of the overall conclusions.

In ilvB, the non-ortholog identity of this clade is supported by the fact that their branches are much longer, and the clade itself is situated on a significantly supported long branch. Members of the clade include representatives of Caulobacterales and closely related Maricaulales and Parvularculales (taxa not

known for LBA artefacts), which in *ilvB* are separated from other Caulobacterales by several significantly supported branches. If one were to root this tree with this clade, the ingroup is extremely reminiscent of a typical Alphaproteobacteria species tree with Magnetococcales and Beta/Gammaproteobacteria as outgroups. These patterns are very strongly indicative of this clade being an out-paralog.

ilvH: the two unnamed Gamibacters

- The placement on the much longer branch inside the outgroup, and the separation of these Gami's with all other Gami's in the tree by several significantly supported branches strongly suggests that these two sequences diverged from all other sequences prior to the last common ancestor of all taxa in this dataset.

-THE CONCATENABILITY OF CYSTEINE BIOSYNTHESIS GENES-

In my previous report, I noted that the single gene trees of the cysteine biosynthesis genes were very incongruent to each other, and that it was therefore not justified to concatenate these genes together. The authors essentially replied that the incongruencies were poorly supported and could thus be explained by a lack of signal. I disagree. I have noted here many significantly supported incongruencies: Perhaps the clearest examples are in the *cysI* gene. Here, the Rickettsiales are separated by all other sequences with a branch that is far longer than you would expect from their faster rate of evolution, and branch together with a Betaproteobacterium and a Rhodospirillaceae with maximum support, far away from the other Betaproteobacteria and Rhodospirillaceae in the tree. This is a sure sign that these long branches separate out-paralogs and should be treated separately. In addition, the separation of *Nitrospira* and *Magnetospirillum* from the other Betaproteobacteria and Rhodospirillaceae respectively, and the maximally supported grouping of 'DDG' taxa and Miti/Jistu/Ark/Atha are not seen in any of the other cysteine biosynthesis single gene trees.

The next clear example is in the *cysH* gene. Here, the Beta- and Gammaproteobacteria are not monophyletic. Instead, the Betaproteobacteria and a single Gammaproteobacterium branch with the 'DDG' Rickettsiales with maximum support. That *Dasania* is separated from the other Gammaproteobacteria with a long branch with maximum support strongly suggests that the Beta/Gamma/DDG clade is an out-paralog relative to the other sequences in the group and should therefore be treated separately. In addition, this branching pattern disagrees with maximum support from all other single gene trees, except perhaps *cysN* where a similar but not identical pattern is observed.

In *CysC*, *Magnetococcus marinus* is situated on a long, maximally supported branch with two Rickettsiaceae and a Gamibacterium, away from *Magnetofaba*, the other Magnetococcales. This strongly suggests that these taxa constitute an out-paralog. In addition, the separation of Magnetococcales is a significantly supported incongruency with e.g. *cysKM* and *cysE* where they branch together with maximum support.

Zymomonas mobilis, Burkholderia cepacia and Magnetococcus marinus all branch in inconsistent places relative to all other taxa in different genes. Zymomonas branches with the other Spingomonadales in cysE, with Gammaproteobacterium Enterobacter in cysJ, cysC and cysN, and in a clade away from both Spingomonadales and Enterobacter in cysD. Burkholderia branches with other Betaproteobacteria in cysE, cysI, cysH and cysD, while nesting deep within Alphaproteobacteria in cysN, cysJ and cysC.

In CysE, the Acetobacteraceae are represented in three highly distinct places in the tree whereas in e.g. cysKM they branch neatly together.

In addition to the described examples above, there are many other less dramatic but not less valid strongly supported incongruencies peppered throughout the trees. The overall pattern suggests that cysteine genes had different evolutionary histories, i.e. a different underlying tree and can therefore not be justifiably concatenated. I implore the authors to remove the suspected out-paralogs from the dataset, redo the single gene trees and omit the concatenation analysis for the cysteine biosynthesis genes. As explained in the previous report, presence of these genes in the last common ancestor of Rickettsiales can still be confidently inferred from individual gene trees. Concatenation is not necessary.

-THE NEED FOR AN "ORGANISM SELECTION" DATASET-

The authors wish to infer the presence of nucleotide and amino acid biosynthesis genes in the last common Rickettsiales ancestor and see if they were vertically inherited since then. They constructed datasets for this purpose that sought to, per biosynthesis gene, gather all genes present in all the selected genomes that had putatively descended from the respective ancestral gene in the Rickettsiales ancestor. This constitutes the "all organisms" datasets. The authors further make "organism selection" datasets, that comprises solely of taxa that have at least 50% of the considered pathway genes present. In my previous report, I argued that this second dataset was unnecessary. Having taxa in a dataset with less than 50% of the genes present (i.e. with a relatively large amount of missing data) are not expected to lead to phylogenetic inaccuracies, only most likely to poorly supported placements of those taxa with fewer genes. I also realize now that any branches leading up to those "less-than-half" taxa may also have reduced support values as a consequence. An "organism selection" dataset may thus have some value for getting a better sense of statistical support for all branches, and evaluating whether the resulting trees indicate vertical descent from the ancestor.

However, the authors provide two additional arguments that in my view are flawed.

They argue that "unbalanced" availability of sites could hamper the accuracy of the phylogenetic inference. I am not aware of such a phenomenon in phylogenetics. As far as I know, taxa with few gene data do not induce strongly supported phylogenetic inaccuracies. Perhaps the authors could provide a reference to back up their statement in line 785?

Further, they argue that the genes present in those "less-than-half" taxa could represent residual non-clearly discernible paralogs/xenologs. Could the authors back this up with some examples? For the Isoleucine/Valine/Leucine analysis for example, all Rickettsiaceae were removed in the "organism

selection" dataset despite being fairly clear true-orthologs in *ilvA* (the only gene where Rickettsiaceae were represented). Similarly, the some Rickettsiaceae and Anaplasmataceae (only in *aroA*) were removed in the "organism selection" dataset of *Trp_Tyr_Phe*, despite again seeming to consist of typical orthologs (with exception of suspicious *Wolbachia* and Rickettsiaceae bacterium GCA_018062985.1 in *aroK*). Most of the Rickettsiaceae in *Lys_Thr_Met* also were removed, despite consisting of typical orthologs in all genes they featured. There is however clade consisting of a mixture of a Rickettsiaceae, Midichloriaceae, an Anaplasmataceae and several Deianiraea in the *lysC/thrA* gene that is very suspicious of being an out-paralog: the Deianiraeaceae are branched far away from other Deianiraea and other DDG taxa with significant support, the Anaplasmataceae same story. Hence all sequences on this branch, including the Rickettsiaceae and Midichloria are likely out-paralogs as well. Despite being very likely paralogs, these Deianiraea and Midichloria are not removed in the "organism selection" dataset. In summary, the creation of the "organism selection" dataset often removes true orthologs, and keeps some non-orthologs. It is at best a very inefficient strategy to deal with residual out-paralogs and xenologs.

Thus, although I now think the "organism selection" dataset has some value (see above), it can't be used as a strategy to deal with residual non-orthologs or to prevent phylogenetic inaccuracies. It is in my view much better to simply remove all sequences that have at least a moderate level of suspiciousness (see my reasoning above) of being non-orthologs by inspection of single gene trees, and then regenerate the "organism selection" dataset from the re-curated "all organisms" dataset.

Finally, despite putting quite a bit of work in these "organism selection" datasets, their results are not discussed in the main text at all. Are there any insights that could be made from the "organism selection" trees, but not from the "all organisms" trees? In other words, what is the added value of the 'organism selection' dataset in the end?

-THE PERSISTENT ARTEFACTUAL GROUPING OF PELAGIBACTERS WITH RICKETTSIALES-

In my previous report, I noted that in trees inferred from the concatenated purine datasets, the Pelagibacters remained (most likely) artefactually placed with the Rickettsiales, even with the most stringent z-score filtering. This is in contrast to most other concatenated datasets (pyrimidine and other amino acid biosynthesis pathways) where the Pelagibacteriales+Rickettsiales artefact is eventually alleviated after some degree of z-score filtering. The authors replied that if this placement were not an artefact, it would indicate a Rickettsiales-to-Pelagibacteriales gene transfer, which is not in the scope of their study and thus not worth mentioning. I agree, if it were indeed not an artefact. But it most likely is, as this particular placement is extremely reminiscent of typical artefact trees in which both groups branch together. In that case, it is worth mentioning, because it indicates that the z-score filtering was, for this particular dataset, not sufficient to alleviate the artefact. Hence, the reader should be made aware that artefacts may still persist in these trees, even with the most stringent z-score filtering.

ON THE INTRACELLULARITY LATE HYPOTHESIS

In my previous report, I believed that the authors based their support for the intracellularity-late hypothesis originally put forward by Castelli et al 2019 on the inference that the basal Rickettsiales (Atha's, Ark's, Miti's and Jistu's) were not intracellular, and that this inference in turn was based on the gene content of these lineages. It turns out that I misunderstood their line of reasoning. The basis of support for intracellularity-late is instead the, in their view, strong support for the independent emergence of host dependence (that is, obligate host-association) in the Rickettsiales. Because intracellular symbionts are by definition obligate host-associated, independent origins of obligate host-association means automatically that intracellularity evolved independently in multiple different lineages as well. This is a solid line of reasoning, but depends on whether the hypothesis of independent origins of obligate host-association is correct or not.

Honestly I am not convinced that the data supports this hypothesis (independently from the Deianiraea vastatrix work) without a doubt. The idea of an independent origin of obligate host-association rests basically on two things: prediction with a high degree of certainty of whether a taxon is obligate host-associated or not, and the non-monophyly of obligate host-associated lineages. After all, if all obligate host-associated lineages are monophyletic, a single origin scenario is the most likely evolutionary history. Now, we know that the Rickettsiaceae, Midichloriaceae and Anaplasmataceae (at least those that have been studied beyond genome sequencing) are obligate symbionts. In addition, Deianiraea vastatrix was previously shown to be an obligate symbiont. Hence, the question is whether any of the other 'DDG' taxa can be confidently predicted to be not obligate symbionts. Because it is the 'DDG' clade, that breaks the monophyly of the Rick/Midi/Ana lineages. The authors do not explicitly state this, but based on their independent origins of obligate host-association conclusion, it is strongly implied that at least some of these DDG taxa are considered to be not-obligate symbionts or even free-living. As far as I can tell, the only* data that suggests that is the richness of aa/nt biosynthesis genes and the lack of an nt transporter and poor-ness of aa transporters in the Diomedesiaceae and Gamibacteraceae MAGs as well as in the Deianiraeaceae MAG that diverges the earliest from all other Deianiraeaceae. The problem with this assertion is basically that, at least within the Rickettsiales, there are no known examples that are known to be not-obligate and have this pattern of being rich in biosynthesis and poor in transporters. The fact that the known obligates are poor in biosynthesis and rich in transporters does not necessarily mean that those that have the reverse pattern are not obligates. For example, *D. vastatrix* is relatively rich in aa transporters but is in fact obligate. I do think the link between the pattern and obligate lifestyle makes a lot of biological and evolutionary sense, but it is I think not enough to confidently predict that these lineages are not obligates. Perhaps if the authors could name some examples of lineages outside the Rickettsiales where the loss of biosynthesis and gain of transporters coincides with a well accepted transition from free-living/facultative host-association to an obligate one, we could be more confident about the prediction of the lifestyle of the DDG taxa, and in turn the hypotheses that obligate host-association and intracellularity evolved independently.

* The other points raised in the Discussion (distribution of secretion systems and putative effectors, as well as other pathways described in SuppText2) seem to me not particularly useful in predicting lifestyles. The T4SS for example is present in nearly all Rickettsiales, and as pointed out by the authors, true host-influencing effectors may still be discovered in many lineages including the DDG clade and basal Rickettsiales. The "taken together" line (I505) implies that all the above points are supporting the non-obligate lifestyles of DDG and in turn the 'late' condition of obligate host-association, but to me only the biosynthesis/transporter argument support it.

In the discussion it is stated that all Deianiraeaceae MAGs have similar genome features (i.e. gene content) in terms of metabolic pathways and putative adhesins, and could therefore be deemed extracellular as well. I think this may indeed be true for the 4 closely related MAGs, but not for the earliest diverging MAG. It is rich in biosynthesis genes, does not have tlc, and is poor in aa transporters as well, while the other Deia's have the exact opposite pattern. It may thus be facultative host-associated or even free-living. Those are technically also 'extracellular' of course, but I'm guessing the authors mean extracellular here as in extracellular symbionts.

MINOR COMMENTS

SuppFig9 still has bootstrap support values at the nodes, should be more like SuppFig10. Also many support values are missing. I have deduced these are actually 100/100 supports, but this was not explained in the figure legend. I recommend to simply show them.

In Figure 3, it may be useful to indicate that the lifestyle cartoons for the basal Rickettsiales, the DDG (but not *D. vastatrix*) and the hypothetical ancestors are hypotheses, and not known, like all other cartoons.

In the paragraph of line 505, a hypothetical facultative host-associated lifestyle for an early Rickettsiales ancestor is discussed. It is stated also that the lifestyle of Deianiraea (I guess *D. vastatrix* is meant here) could be reminiscent of the lifestyle of the Rickettsiales ancestor. However, the lifestyle of *D. vastatrix* is later stated to be an obligate host-associated one, while the hypothetical ancestor is discussed to have a facultative one. Hence, it is an odd comparison.

In paragraph of line 527, an "envisioned" scenario is described, which in line 532 is then considered support for the late origin of obligate host-association. A hypothetical scenario can not serve as evidence or support for another hypothesis. I also raised this issue in my previous report, but it was not addressed.

REVIEWER COMMENTS

Reviewer #1 (Remarks to the Author):

We thank the Reviewer for the comments received. We largely accounted for those, including modifying several analyses and the manuscript accordingly, while, for a limited set of issues, we explained in detail the reasons behind our respectful disagreement (see below). For the sake of clarity, we opted to provide, whenever possible, comprehensive answers to the points included in each section of the Reviewer's report (based on the provided headings), or even comprehensive answers to interconnected sections (i.e. site removal methods+constraint tree, and several sections pertaining to the phylogenies on amino acid and nucleotide biosynthesis genes).

SPECIES TREE RECONSTRUCTION

-SITE REMOVAL METHODS-

In my previous report I provided critique on the assertion of the authors that removal of heterogeneous/biased sites can unintentionally lead to new artefacts. I stated that this was not a phenomenon recognized by the current phylogenetic literature. The author's replied in their rebuttal that on the one hand they could indeed "not find any reference clearly demonstrating the unsuitability of site-removal approaches for tackling composition biases", but on the other hand could also not find references that comprehensively tested the performance of said site removal methods. Indeed, to my knowledge there have not been in any studies where site-removal methods have been tested on simulated datasets, where the true tree is known. Such a study would indeed be very valuable to the deep phylogenomics community. However, there have been several studies that have shown that site-removal can lead to dramatic improvements in model fit for empirical datasets. In Bayesian methods for example, the ability of the model (with parameter settings optimized for the dataset at hand) to capture across-taxa and across-site heterogeneity adequately can be assessed with posterior-predictive tests. In Nesnidal et al (2010), Zaremba-Niedzwiedzka et al (2017), Martijn et al (2018), Martijn et al (2020), removing the most biased sites have led to dramatic improvements in model fit. Even Fan et al (2020) (cited by the authors in their rebuttal), who attempt to discredit site-removal methods, show in their Figure 1 that all investigated site-removal methods drastically improve model fit. Trees with better model fit should always be preferred over trees with poorer model fit. Moreover, in Munoz-Gomez et al (2021), a tree reconstructed after removing the most AT/GC biased sites was strongly preferred by a more realistic model of evolution over a tree reconstructed from an untreated dataset. Hence, with the current knowledge, site-removal methods appear to be a powerful tool to improve model fit and disentangle phylogenetic artefacts, with no known side effects that induce new artefacts. Finally, I would like to comment that the Foster et al (2022) paper cited by the authors purely investigates recoding strategies, and not site-removal strategies. These are vastly different methodologies, each with their own intricacies. Hence, that study can not be used as an argument against site removal strategies.

-THE CONSTRAINT TREE-

In my previous report I strongly encouraged the authors to omit the constraint tree in favor of the 20% trimming tree, in case both trees turned out to be identical. Constraint trees should strictly be used for tree topology testing. However, if the trees did not turn out to be identical, this would mean that the authors would have to do a complete rerun of the computationally and labour intensive ALE analysis, now with the 20% trimming tree. As both species tree would only be minimally different, the new ALE results would not be too different either. In that case, I stated I would be OK with leaving in the constraint tree. As it turned out, the trees were not exactly identical (only 7 different bipartitions). Hence, the authors kept their constraint tree.

When I wrote this comment, I expected the authors to keep their ALE analysis. However, due to other issues, the authors have decided to completely omit their ALE analysis from the manuscript. This

means that, in my view, there is no more reason to keep the constraint tree. As explained above and earlier reports, using constraints to infer species trees should be avoided if possible. The authors seem to be weary of site removal strategies, mainly because in their view they (1) reduce phylogenetic signal and can lead to unsupported bipartitions, and (2) may lead to unintentional new phylogenetic artefacts. However, as I explained in my previous reports, bipartitions that are significantly supported w/o site removal, but lose support as sites are removed can still be trusted as long as they are not replaced by other significantly supported bipartitions. In addition, while total phylogenetic signal may indeed decrease, the signal-to-noise ratio (where noise is responsible for systemic errors) typically increases substantially. And as explained above, site removal methods have -as of yet- never led to new artefacts. It seems to me that in their desire to be "cautious" (in and of itself a good idea), the authors have actually ended up choosing a less cautious approach, that is, using the constraints. I therefore insistently recommend that the authors choose the 20% trimmed tree to show as their main species tree, rather than the constraint tree, for the reasons above, but also to prevent setting a dangerous precedent. This should not alter any of their conclusions, nor force them to redo any analyses.

We did not mean here to set any kind of “precedent”, we apologise that it could have looked as if so. Rather, we aimed to follow a cautious and pragmatic approach according to our considerations and interpretations on the methods and results obtained. We acknowledge the concern by the Reviewer on the constrained tree, accordingly we have entirely removed it from the manuscript.

Regarding compositional heterogeneity-based site removal approaches, we obviously do agree on their sense and overall merit. However, as extensively presented in the previous replies, while we acknowledge the insight and expertise by the Reviewer, we think that their potential limits are not yet a fully understood matter, which “per se” suggests caution. Such cases include inference of finer evolutionary events, such as the placement of DDG herein (see also reply to the dedicated section below).

It was under this reasoning that we meant to provide some literature reference on potential drawbacks even of methods based on similar assumptions (i.e. recoding), though technically different. We also consider that any metric such as model fit (mentioned by the Reviewer) can be used at best as a “proxy” for phylogenetic accuracy (the ultimate goal of inferences), but cannot be treated as a direct “index” for that. Direct measuring of accuracy is obviously impossible on real world datasets, while benchmarking on simulated datasets seems to be still lacking for the herein discussed methods, at least according to our and the Reviewer’s knowledge.

Taking all the above into account, we judge that the most sensible selection as our “main” tree (Fig. 1) is the untrimmed (and unconstrained) tree, rather than the 20%-trimmed tree suggested by the Reviewer. We underline that, after previous removal of the ALE analyses, this is mostly a matter of presentation choices. All the compositionally trimmed trees are included as supplementary, and this point is accounted in the manuscript, with further appropriate edits after the removal of the constrained tree (lines 124-148 and 737-738 of the track-mode manuscript, and the track-mode supplementary text 4).

-THE PLACEMENT OF DDG RELATIVE TO MIDI AND ANA-

The authors observe that their obtained topology is largely in line with previously published Rickettsiales trees, with the exception of the placement of the 'DDG' clade. They recover (('DDG',Ana),(Midi)), whereas Schon et al recovered (('DG'),(Ana,Midi)). They highlight this difference and speculate on what could cause the difference. They point out in their manuscript that Schon et al also recovered (('DG',Ana),(Midi)), but that this topology lost support and was replaced by (('DG'),(Ana,Midi)) when compositional trimming was applied. Currently, the way this section is

written, the readers are led to believe that Schon et al obtained a supported (('DG',Ana),(Midi)) tree using a ML analyses based on a compositional trimmed dataset. It should be made clear that this topology was only significantly supported with the Bayesian inference, and that the Schon et al ML trees in fact show a similar pattern as the authors ML trees, namely that support for (('DDG/DG',Ana),(Midi)) is lost and not replaced by another significantly supported topology when compositional heterogeneous sites are removed.

In my previous report I explained that the increased taxon sampling was most likely not responsible for the shift from (('DG'),(Ana,Midi)) to (('DDG',Ana),(Midi)). However, the authors still consider it a possibility in their revision. I must insist here that there is no sign or hint that additional taxa are responsible for the topological shift. In fact, recent ML analyses (Castelli et al 2019; Schon et al 2022) already showed (('DG/D',Ana),(Midi)) trees. (('DG'),(Ana,Midi)) was only recovered by Schon et al with a combination of compositional trimming and Bayesian tree inference. In other words, compared to similar ML analyses (albeit not perfectly comparable, due to slight alterations in evolutionary models and algorithms), there was no topological shift. Additional taxa can not be responsible for a shift in topology, because there was no shift to begin with. Of course, it is possible that if the authors were to repeat their ML analyses without the new taxa, a topology incongruent with (('DDG',Ana),(Midi)) (be it -(('DG'),(Ana,Midi))- or something else) would be recovered. Actually, if the authors would like to back up their statement, this should be a fairly straightforward analysis to carry out. However, in light of the currently available ML analyses, this seems quite unlikely to me.

We apologise for our inaccuracy, in that we did not clearly state that, according to the results by Schon et al., branch support for different reconstruction is also significantly dependent of the inference methods (Bayesian inference vs maximum likelihood). This is now stated clearly in the manuscript (see lines 168-193 of the track-mode manuscript). At the same time, considering that taxon selection is in general recognised for its impact on phylogenetic inferences (e.g. doi: 10.3724/SP.J.1002.2008.08016; doi: 10.1093/bib/bbr014), and its “contribution” here cannot be at present excluded, we still believe that it is worth mentioning this as well. As previously agreed by the Reviewer in a past report, we believe that further analyses to clarify this point are beyond the aim of the current study.

RECONSTRUCTION OF HOMOLOGOUS GROUPS

In my previous report I requested that the authors should more clearly define what they mean exactly with the concept of a 'homology group', to aid the reader in understanding their methodology. The current manuscript provides an operational description of homology groups, that is, how they are reconstructed, but not a biological definition, that is, how each gene is related to each other gene in the group. It should be made more clear that any pair of genes within a group of sequences (that aims to approximate a true homology group) does not necessarily have a last common ancestor that is younger than the last common ancestor of the given group of species. Being clear on the definition of evolutionary datasets like this aids the reader to follow and understand the analyses better.

The Reviewer is right, biological meaning of this choice is needed. We have clarified this in the manuscript (lines 753-770 of the track-mode manuscript).

PHYLOGENETIC ANALYSES OF NUCLEOTIDE AND AMINO ACID BIOSYNTHESIS GENES

In my previous report I outlined a fair number of sequences in the datasets of nucleotide and amino acid biosynthesis genes that were suspicious of being (out-)paralogs or (out-)xenologs. Such sequences can at best be not informative on the presence/absence state of a gene in a particular ancestor and at worst interfere with the accuracy of trees inferred from concatenated datasets. The authors have decided to keep of all these suspicious sequences in their datasets. They argue that for all these sequences, it is not 100% clear whether these indeed represent (out-)paralogs and/or

(out-)xenologs or fast-evolving/misplaced true orthologs. Unintentional removal of true orthologs may "bias" the dataset. I disagree with this line of reasoning. First, in my view many of the sequences that I've highlighted are clear non-orthologs for one reason or another (see below as well as previous report), not mere edge coin-toss cases. Second, I do not see how unintentional removal of a true ortholog biases the dataset. The only thing that happens is that the dataset loses a little bit of signal. What biases a dataset is the inclusion of non-orthologs! Hence, removing a suspicious sequence in the worst case (i.e. removing a true ortholog) leads to a small reduction in signal, whereas the best case (i.e. removing a non-ortholog) removes a relative large source of noise. Finally, the authors argue that the fact that their concatenated trees reflect vertical inheritance despite the likely inclusion of non-orthologs strengthens rather than weakens the weight of their conclusions. This in my view is flawed reasoning as well. The weight of your conclusion should depend on the quality of the tree, and hence the quality of the dataset, regardless of what the shape of the inferred tree is. In other words, the tree inferred from a dataset that includes known sources of noise should be taken with a grain of salt, regardless of what the topology of that tree is. On a side note, the authors mention that, "in order to filter out clear paralogs, (...), we retained only sequences belonging to clades with best hits sequences and/or NCBI domain hits on the sequences/profiles consistent with the desired ortholog". I might have missed it but I could not find this curation step anywhere in the manuscript. Please include it.

-CLEAR CASES OF NON-ORTHOLOGS-

pyrB: The clade consisting of Gami's, a Deia, two Atha's, a Jistu and four Gammaproteobacteria.
- The entire clade is maximally supported and consists of taxa with branches much longer relative to the rest of the tree, and is situated on a long branch itself. This is suggestive of either lba or a set of non-orthologs.

- The clade contains Gami's and a Jistu that are separated from their other family members by several maximum or significantly supported branches. This is suggestive of non-orthologs only.

- The clade contains Gammaproteobacteria outgroups. This is a strong sign that this clade is a part of another orthologous group (i.e. non-orthologs relative the other sequences in the pyrB group)
purL and purB: The outgroup consisting of Beta, Gamma and Magnetococcae

- The branch separating the Alphaproteobacteria and the outgroups is exorbitantly extremely long. Such a branch length can not be explained by simple accelerated or fast-evolving sequences. The most likely explanation is that in- and outgroup are homologous to one another, but diverged long before the last common Proteobacteria ancestor. One copy was retained in Alpha's but lost in outgroups, and the other copy vice versa. It may also be that the outgroup sequences and the ingroup sequences share some homologous domains, but that other domains not shared between in- and outgroup were falsely aligned with one another, leading to such extreme branch lengths. Either way, both gene datasets must be curated.

serB: The clade consisting of Gami's and a Deia

- The branch separating this clade from all other sequences is maximally supported and significantly longer than all other sequences. Much longer than can be explained by higher evolutionary rates as suggested by the species tree and many other single gene trees. These sequences most likely diverged from the other sequences in the group prior to the last common ancestor of the taxa in this group. By extension, the sequences corresponding to these taxa in the serA and serC are highly suspicious too. In serA in particular, this clade is separated from the other Rickettsiales sequences by another maximally supported, long branch.

hisI: The clade containing Geminicoccus, several Caulo's, Gammaproteobacteria, a Gami and several Deia's

- I highlighted this clade in my previous report as well. The author's replied that in their view, the placement of this clade could be explained by vertically inherited paralogs (i.e. in-paralogs), horizontal transfers, or long branch attraction induced by fast-evolving but vertically inherited orthologous sequences. I disagree. First, the taxa involved are typically not fast-evolving, with perhaps the exception of the Gami's and to a greater extent the Deia's. Caulobacters, Gammaproteobacteria and Geminicoccus are generally not fast-evolving, nor are they involved in

long-branch artefacts. Second, the odd placement and odd composition of this clade could indeed be explained by multiple horizontal gene transfers, in which case they should be removed anyway. Third, a duplication after the divergence of the last common ancestor of all taxa in this dataset does not explain the Gammaproteobacteria inside this clade. In my view, this clade is a clear case of a non-ortholog clade for one more reason: the Caulobacterales representatives and the Gamibacter are separated from the other family members by several significantly supported branches. This pattern is much more likely to be explained by this clade being an out-paralog. I.e. it can trace its common ancestry with the other sequences to a duplication event that predates the last common ancestor of all taxa in the dataset. This non-ortholog clade is also present in all other histidine gene datasets and should be removed.

hisG: the abovementioned clade as well as its sister clade

- The placement of this clade should be re-evaluated after the removal of the clade discussed above, but in the current single gene tree I find it highly suspicious of being yet another non-ortholog clade. This is mainly motivated by the much longer branches of the taxa in this clade relative to the other sequences, which are themselves situated on a maximum supported long branch. With exception of the Pelagibacters, none of these taxa are typically involved in any known long branch attraction artefacts. In addition, the placement of Magnetococcales in this clade is highly reminiscent of a typical Alphaproteobacteria species tree, suggesting that this clade diverged from the other sequences prior to the last common ancestor of all taxa in this dataset.

ilvIB: the clade consisting of *occidentia*, *citromicrobium*, *parvularcula* and several *caulobacterales*

- This suspicious clade is also present in several other gene trees (*ilvIH*, *ilvID*, *leuB*) and was also highlighted in my previous report. The authors replied that whether these taxa were non-orthologs or not was outside the scope of the study and should be verified with dedicated analyses. I argue that it is very much inside the scope of your study. The conclusions drawn from observations depend on the quality of the trees and hence the quality of the dataset. These sequences are, in my view, clear non-orthologs and thus reduce the quality of the dataset, and that of the overall conclusions. In *ilvIB*, the non-ortholog identity of this clade is supported by the fact that their branches are much longer, and the clade itself is situated on a significantly supported long branch. Members of the clade include representatives of *Caulobacterales* and closely related *Maricaulales* and *Parvularculales* (taxa not known for LBA artefacts), which in *ilvIB* are separated from other *Caulobacterales* by several significantly supported branches. If one were to root this tree with this clade, the ingroup is extremely reminiscent of a typical Alphaproteobacteria species tree with *Magnetococcales* and *Beta/Gammaproteobacteria* as outgroups. These patterns are very strongly indicative of this clade being an out-paralog.

ilvIH: the two unnamed *Gamibacters*

- The placement on the much longer branch inside the outgroup, and the separation of these *Gami*'s with all other *Gami*'s in the tree by several significantly supported branches strongly suggests that these two sequences diverged from all other sequences prior to the last common ancestor of all taxa in this dataset.

-THE CONCATENABILITY OF CYSTEINE BIOSYNTHESIS GENES-

In my previous report, I noted that the single gene trees of the cysteine biosynthesis genes were very incongruent to each other, and that it was therefore not justified to concatenate these genes together. The authors essentially replied that the incongruencies were poorly supported and could thus be explained by a lack of signal. I disagree. I have noted here many significantly supported incongruencies: Perhaps the clearest examples are in the *cysI* gene. Here, the *Rickettsiales* are separated by all other sequences with a branch that is far longer than you would expect from their faster rate of evolution, and branch together with a *Betaproteobacterium* and a *Rhodospirillaceae* with maximum support, far away from the other *Betaproteobacteria* and *Rhodospirillaceae* in the tree. This is a sure sign that these long branches separate out-paralogs and should be treated separately. In addition, the separation of *Nitrospira* and *Magnetospirillum* from the other *Betaproteobacteria* and *Rhodospirillaceae* respectively, and the maximally supported grouping of 'DDG' taxa and *Miti/Jistu/Ark/Atha* are not seen in any of the other cysteine biosynthesis single gene trees.

The next clear example is in the *cysH* gene. Here, the Beta- and Gammaproteobacteria are not monophyletic. Instead, the Betaproteobacteria and a single Gammaproteobacterium branch with the 'DDG' Rickettsiales with maximum support. That *Dasania* is separated from the other Gammaproteobacteria with a long branch with maximum support strongly suggests that the Beta/Gamma/DDG clade is an out-paralog relative to the other sequences in the group and should therefore be treated separately. In addition, this branching pattern disagrees with maximum support from all other single gene trees, except perhaps *cysN* where a similar but not identical pattern is observed.

In *CysC*, *Magnetococcus marinus* is situated on a long, maximally supported branch with two Rickettsiaceae and a Gamibacterium, away from *Magnetofaba*, the other Magnetococcales. This strongly suggests that these taxa constitute an out-paralog. In addition, the separation of Magnetococcales is a significantly supported incongruency with e.g. *cysKM* and *cysE* where they branch together with maximum support.

Zymomonas mobilis, *Burkholderia cepacia* and *Magnetococcus marinus* all branch in inconsistent places relative to all other taxa in different genes. *Zymomonas* branches with the other Spingomonadales in *cysE*, with Gammaproteobacterium *Enterobacter* in *cysJ*, *cysC* and *cysN*, and in a clade away from both Spingomonadales and *Enterobacter* in *cysD*. *Burkholderia* branches with other Betaproteobacteria in *cysE*, *cysI*, *cysH* and *cysD*, while nesting deep within Alphaproteobacteria in *cysN*, *cysJ* and *cysC*.

In *CysE*, the Acetobacteraceae are represented in three highly distinct places in the tree whereas in e.g. *cysKM* they branch neatly together.

In addition to the described examples above, there are many other less dramatic but not less valid strongly supported incongruencies peppered throughout the trees. The overall pattern suggests that cysteine genes had different evolutionary histories, i.e. a different underlying tree and can therefore not be justifiably concatenated. I implore the authors to remove the suspected out-paralogs from the dataset, redo the single gene trees and omit the concatenation analysis for the cysteine biosynthesis genes. As explained in the previous report, presence of these genes in the last common ancestor of Rickettsiales can still be confidently inferred from individual gene trees. Concatenation is not necessary.

-THE NEED FOR AN "ORGANISM SELECTION" DATASET-

The authors wish to infer the presence of nucleotide and amino acid biosynthesis genes in the last common Rickettsiales ancestor and see if they were vertically inherited since then. They constructed datasets for this purpose that sought to, per biosynthesis gene, gather all genes present in all the selected genomes that had putatively descended from the respective ancestral gene in the Rickettsiales ancestor. This constitutes the "all organisms" datasets. The authors further make "organism selection" datasets, that comprises solely of taxa that have at least 50% of the considered pathway genes present. In my previous report, I argued that this second dataset was unnecessary. Having taxa in a dataset with less than 50% of the genes present (i.e. with a relatively large amount of missing data) are not expected to lead to phylogenetic inaccuracies, only most likely to poorly supported placements of those taxa with fewer genes. I also realize now that any branches leading up to those "less-than-half" taxa may also have reduced support values as a consequence. An "organism selection" dataset may thus have some value for getting a better sense of statistical support for all branches, and evaluating whether the resulting trees indicate vertical descent from the ancestor.

However, the authors provide two additional arguments that in my view are flawed. They argue that "unbalanced" availability of sites could hamper the accuracy of the phylogenetic inference. I am not aware of such a phenomenon in phylogenetics. As far as I know, taxa with few gene data do not induce strongly supported phylogenetic inaccuracies. Perhaps the authors could provide a reference to back up their statement in line 785?

Further, they argue that the genes present in those "less-than-half" taxa could represent residual non-clearly discernible paralogs/xenologs. Could the authors back this up with some examples? For the Isoleucine/Valine/Leucine analysis for example, all Rickettsiaceae were removed in the "organism selection" dataset despite being fairly clear true-orthologs in *ilvA* (the only gene where Rickettsiaceae

were represented). Similarly, the some Rickettsiaceae and Anaplasmataceae (only in aroA) were removed in the "organism selection" dataset of Trp_Tyr_Phe, despite again seeming to consist of typical orthologs (with exception of suspicious Wolbachia and Rickettsiaceae bacterium GCA_018062985.1 in aroK). Most of the Rickettsiaceae in Lys_Thr_Met also were removed, despite consisting of typical orthologs in all genes they featured. There is however clade consisting of a mixture of a Rickettsiaceae, Midichloriaceae, an Anaplasmataceae and several Deianiraea in the lysC/thrA gene that is very suspicious of being an out-paralog: the Deianiraeaceae are branched far away from other Deianiraea and other DDG taxa with significant support, the Anaplasmataceae same story. Hence all sequences on this branch, including the Rickettsiaceae and Midichloria are likely out-paralogs as well. Despite being very likely paralogs, these Deianiraea and Midichloria are not removed in the "organism selection" dataset. In summary, the creation of the "organism selection" dataset often removes true orthologs, and keeps some non-orthologs. It is at best a very inefficient strategy to deal with residual out-paralogs and xenologs.

Thus, although I now think the "organism selection" dataset has some value (see above), it can't be used as a strategy to deal with residual non-orthologs or to prevent phylogenetic inaccuracies. It is in my view much better to simply remove all sequences that have at least a moderate level of suspiciousness (see my reasoning above) of being non-orthologs by inspection of single gene trees, and then regenerate the "organism selection" dataset from the re-curated "all organisms" dataset.

Finally, despite putting quite a bit of work in these "organism selection" datasets, their results are not discussed in the main text at all. Are there any insights that could be made from the "organism selection" trees, but not from the "all organisms" trees? In other words, what is the added value of the 'organism selection' dataset in the end?

We thank the Reviewer for the detailed report on several points related to the phylogenies of nucleotide and amino acid biosynthesis genes. Accounting for the suggestions received, we made several changes to our analyses and manuscript (see below for details). In particular, we removed all the suspected non-orthologs indicated by the Reviewer from the concatenated alignments, fully removed the concatenated phylogenies for cysteine, and also removed the "organism selection" analyses for all pathways.

We also realise that we probably failed in properly describing some relevant matters of our reasoning and study design. We apologise for that, and hope that these are now clear here and in the revised manuscript (see lines 793-837 of the track-mode manuscript).

Starting from "homologous groups", here we aimed to understand whether the presence of biosynthesis genes for nucleotide and amino acids could be traced back to the ancestor of *Rickettsiales* (case 1), or to more recent acquisitions (case 2). The latter (case 2) would hold true in case of recent HGT (i.e., after the last *Rickettsiales* ancestor) from whichever origin. Whereas, case 1 would hold true in all other possibilities, including orthologs, in-paralogs specific to *Rickettsiales*, ancient paralogs vertically inherited from the last *Rickettsiales* ancestor (even if received by an ancient HGT from whichever origin).

To do so, we must have a dataset that also includes suspect genes, with the specific aim of understanding what they are. Orthologs indicate case 1, non orthologs could fall in case 1 or 2, and our approach is specifically designed to understand where they fall.

The exclusion of suspected non-orthologs is what would represent a "bias" of dataset selection towards case 1 (a bias that would support our inferences on late host-association and intracellularity). For this reason, in respectful disagreement with the Reviewer, we think it is meaningful to keep the evidenced suspected non-orthologs, to be presented in our single gene trees. At the same time, we do acknowledge the Reviewer's concern on the bias that any non-ortholog would have on the inference of concatenated phylogenies. Accordingly, we removed all suspected non-orthologs indicated by the Reviewer for the concatenated alignment of the respective pathway (synthesis of pyrimidines, purines, serine, histidine, branched-chain aa, aromatic aa, "Lyt+Met+Thr") and newly performed phylogenies only on those concatenated "confirmed" orthologs. Removed genes are indicated in a

new supplementary file (Supplementary table 4). Following the Reviewer's indication, we removed the phylogenies on cysteine pathway genes, due to too many incongruences possibly caused by non-orthologs, as well as those on the "organism selection" datasets for all pathways, which seem indeed dispensable with the newly "polished" concatenation of "confirmed" orthologs. We judge that such "pairing" of phylogenies on single genes (including potential non-orthologs) + phylogenies of concatenated "polished" orthologs (with no further edits) works adequately for presenting clearly and concisely the findings of our study, and in compliance with the Reviewer's requests.

According to all the above, the manuscript has been changed (see lines 308-332, 365-403, and 793-837 of the track-mode manuscript). As suggested by the Reviewer, details on manual inspection for paralogs were also added (see Supplementary text 5, in track-mode)

“”

-THE PERSISTENT ARTEFACTUAL GROUPING OF PELAGIBACTERS WITH RICKETTSIALES-

In my previous report, I noted that in trees inferred from the concatenated purine datasets, the Pelagibacters remained (most likely) artefactually placed with the Rickettsiales, even with the most stringent z-score filtering. This is in contrast to most other concatenated datasets (pyrimidine and other amino acid biosynthesis pathways) where the Pelagibacterales+Rickettsiales artefact is eventually alleviated after some degree of z-score filtering. The authors replied that if this placement were not an artefact, it would indicate a Rickettsiales-to-Pelagibacterales gene transfer, which is not in the scope of their study and thus not worth mentioning. I agree, if it were indeed not an artefact. But it most likely is, as this particular placement is extremely reminiscent of typical artefact trees in which both groups branch together. In that case, it is worth mentioning, because it indicates that the z-score filtering was, for this particular dataset, not sufficient to alleviate the artefact. Hence, the reader should be made aware that artefacts may still persist in these trees, even with the most stringent z-score filtering.

We agree with the Reviewer that this is most likely artefactual. We understand the Reviewer's point regarding that the presence of residual artefact, though not directly impactful for the hypotheses examined, could be useful as being informative on the limits of the compositional trimming as applied in the specific dataset. This is now specified in the manuscript (lines 327-332 of the track-mode manuscript).

ON THE INTRACELLULARITY LATE HYPOTHESIS

In my previous report, I believed that the authors based their support for the intracellularity-late hypothesis originally put forward by Castelli et al 2019 on the inference that the basal Rickettsiales (Atha's, Ark's, Miti's and Jistu's) were not intracellular, and that this inference in turn was based on the gene content of these lineages. It turns out that I misunderstood their line of reasoning. The basis of support for intracellularity-late is instead the, in their view, strong support for the independent emergence of host dependence (that is, obligate host-association) in the Rickettsiales. Because intracellular symbionts are by definition obligate host-associated, independent origins of obligate host-association means automatically that intracellularity evolved independently in multiple different lineages as well. This is a solid line of reasoning, but depends on whether the hypothesis of independent origins of obligate host-association is correct or not.

Honestly I am not convinced that the data supports this hypothesis (independently from the Deianiraea vastatrix work) without a doubt. The idea of an independent origin of obligate host-association rests basically on two things: prediction with a high degree of certainty of whether a taxon is obligate host-associated or not, and the non-monophyly of obligate host-associated lineages. After all, if all obligate host-associated lineages are monophyletic, a single origin scenario is the most likely

evolutionary history. Now, we know that the Rickettsiaceae, Midichloriaceae and Anaplasmataceae (at least those that have been studied beyond genome sequencing) are obligate symbionts. In addition, *Deianaraea vastatrix* was previously shown to be an obligate symbiont. Hence, the question is whether any of the other 'DDG' taxa can be confidently predicted to be not obligate symbionts. Because it is the 'DDG' clade, that breaks the monophyly of the Rick/Midi/Ana lineages. The authors do not explicitly state this, but based on their independent origins of obligate host-association conclusion, it is strongly implied that at least some of these DDG taxa are considered to be not-obligate symbionts or even free-living. As far as I can tell, the only* data that suggests that is the richness of aa/nt biosynthesis genes and the lack of an nt transporter and poor-ness of aa transporters in the Diomedesiaceae and Gamibacteraceae MAGs as well as in the *Deianiraeaceae* MAG that diverges the earliest from all other *Deianiraeaceae*. The problem with this assertion is basically that, at least within the Rickettsiales, there are no known examples that are known to be not-obligate and have this pattern of being rich in biosynthesis and poor in transporters. The fact that the known obligates are poor in biosynthesis and rich in transporters does not necessarily mean that those that have the reverse pattern are not obligates. For example, *D. vastatrix* is relatively rich in aa transporters but is in fact obligate. I do think the link between the pattern and obligate lifestyle makes a lot of biological and evolutionary sense, but it is I think not enough to confidently predict that these lineages are not obligates. Perhaps if the authors could name some examples of lineages outside the Rickettsiales where the loss of biosynthesis and gain of transporters coincides with a well accepted transition from free-living/facultative host-association to an obligate one, we could be more confident about the prediction of the lifestyle of the DDG taxa, and in turn the hypotheses that obligate host-association and intracellularity evolved independently.

* The other points raised in the Discussion (distribution of secretion systems and putative effectors, as well as other pathways described in SuppText2) seem to me not particularly useful in predicting lifestyles. The T4SS for example is present in nearly all Rickettsiales, and as pointed out by the authors, true host-influencing effectors may still be discovered in many lineages including the DDG clade and basal Rickettsiales. The "taken together" line (l505) implies that all the above points are supporting the non-obligate lifestyles of DDG and in turn the 'late' condition of obligate host-association, but to me only the biosynthesis/transporter argument support it.

In the discussion it is stated that all *Deianiraeaceae* MAGs have similar genome features (i.e. gene content) in terms of metabolic pathways and putative adhesins, and could therefore be deemed extracellular as well. I think this may indeed be true for the 4 closely related MAGs, but not for the earliest diverging MAG. It is rich in biosynthesis genes, does not have tlc, and is poor in aa transporters as well, while the other *Deia*'s have the exact opposite pattern. It may thus be facultative host-associated or even free-living. Those are technically also 'extracellular' of course, but I'm guessing the authors mean extracellular here as in extracellular symbionts.

It seems that the reasoning and foundations of our hypothesis were not yet clear, we apologise for that. With the novel edits to the manuscript (see lines 461-470, 502-507, 555-556, 580-585, and 592-598 of the track-mode manuscript), we hope this is now clear. We are sincerely happy that the Reviewer judges that our findings linking patterns of transporters and biosynthesis with the lifestyle are biologically and evolutionarily meaningful. We also understand that the Reviewer might have some legitimate doubts on our host-association and intracellularity hypotheses, which we hope can be at least mitigated by the points here clarified (see below). In any case, we posit that it is inherently impossible to prove any reconstruction on evolutionary history, which can never be treated as conclusive "prediction". Thus, while we present here our hypothesis with, in our view, reasonable supporting elements (see below), it seems well appropriate that these may not be found fully convincing by other researchers, perhaps (and hopefully!) fostering future research that could shed further (still never conclusive!) light on this.

We respectfully disagree with the Reviewer on one point.

If all obligatorily host-associated (crown) *Rickettsiales* lineages formed a monophyletic clade, an obligatorily host-associated ancestor would be “a priori” most parsimonious, until additional data come into the picture. When such data are found, they must be taken into account, and can point towards alternative reconstructions being more likely. Specifically, the main point of this manuscript is precisely that, in our view, multiple elements emerging from genomic analysis suggest an alternative reconstruction: “late” independent evolution of host association in different crown *Rickettsiales* lineages. This hypothesis is not based on, and does not require, that some DDG *Rickettsiales* MAGs are free-living, and we never implied that!

Regarding other elements to be “taken together” in support of the host-association late hypothesis, as extensively described in Supplementary text 2, and now also indicated in the manuscript (lines 580-585 of the track-mode manuscript), we identified presence/absence pattern of reduction comparable to those of nucleotides and amino acids in many other biosynthetic and energy metabolism pathways (i.e. presence in basal *Rickettsiales* and early diverging members of some crown *Rickettsiales* families), thus being suggestive of multiple independent reductions. Our in-depth analyses were focused on amino acid and nucleotide biosynthesis in consideration that these were deemed as more “predictive”, and, especially for nucleotides, transport counterparts could be paired, but all these other pathways are taken into account to “complete” the scenario. Similar considerations on presence/absence and putative independent reduction/losses hold true for genes involved in “Secretion, attachment, and motility” (see lines 271-297 of the track-mode manuscript), e.g. flagella, chemotaxis, T2SS/type IV pili. This has been now made more explicit in the discussion (lines 502-507 of the track-mode manuscript). In this context, the conservation of T4SS, mentioned by the Reviewer, seems peculiar rather than exemplifying.

We thank for the suggestion of finding supportive non-*Rickettsiales* examples indicating evolutionary coincidence of loss of biosynthesis and gain of transporters with transition towards obligate associations. To our knowledge, it is a general notion that loss of metabolic independence involved in such transitions results from a number of steps, among which biosynthesis of amino acids and nucleotides are presumably among the most significant, but not necessarily coincident with the transition (hard to trace), nor each strictly necessary (even among *Rickettsiales*, some have lost just “nucleotides” or “amino acids”, see also lines 574-577 of the track-mode manuscript). Accordingly, in the present study we treated those steps as “indicative” rather than ultimate determinants for such transitions. In any case, the review by Dean and coauthors on Microsporidia (<https://doi.org/10.1371/journal.ppat.1005870>) points towards a concept quite similar to ours, suggesting that “a critical innovation for adapting to intracellular life was the acquisition by lateral gene transfer of nucleotide transport (NTT) proteins [=tlc translocases] that are now present in multiple copies in all microsporidian genomes” because these proteins “allow microsporidia to steal ATP and other purine nucleotides for energy and biosynthesis from their host”. This has been added to the manuscript (lines 461-470 of the track-mode manuscript).

The Reviewer is right that our consideration on the lifestyle of *Deianiraeaeceae* was too general and thus inaccurate, we have now rephrased it (lines 592-598 of the track-mode manuscript).

MINOR COMMENTS

SuppFig9 still has bootstrap support values at the nodes, should be more like SuppFig10. Also many support values are missing. I have deduced these are actually 100/100 supports, but this was not explained in the figure legend. I recommend to simply show them.

We apologise that the omission of 100/100 supports was not mentioned in the tree or its legend. Following the Reviewer's suggestion, we have now added all supports in the novel version of this supplementary file.

In Figure 3, it may be useful to indicate that the lifestyle cartoons for the basal Rickettsiales, the DDG (but not *D. vastatrix*) and the hypothetical ancestors are hypotheses, and not known, like all other cartoons.

We thank for this suggestion. We have now "increased" the style differences between the ascertained and hypothetical lifestyles (wholly coloured vs grey). Please see novel version of Figure 3 and its legend.

In the paragraph of line 505, a hypothetical facultative host-associated lifestyle for an early Rickettsiales ancestor is discussed. It is stated also that the lifestyle of *Deianiraea* (I guess *D. vastatrix* is meant here) could be reminiscent of the lifestyle of the Rickettsiales ancestor. However, the lifestyle of *D. vastatrix* is later stated to be an obligate host-associated one, while the hypothetical ancestor is discussed to have a facultative one. Hence, it is an odd comparison.

In this case, we considered that *D. vastatrix* could share (perhaps by direct inheritance) some but not all lifestyle traits with the hypothesised ancestor (in terms of being "predators" of unicellular eukaryotes, but being already obligatorily host-associated). Accordingly, by "reminiscent" we did not mean "fully equivalent". We have rephrased to make this point clearer (lines 555-556 of the track-mode manuscript).

In paragraph of line 527, an "envisioned" scenario is described, which in line 532 is then considered support for the late origin of obligate host-association. A hypothetical scenario can not serve as evidence or support for another hypothesis. I also raised this issue in my previous report, but it was not addressed.

We thank the Reviewer for this comment, which we respectfully disagree with. We apologise for not making clear our views previously. We posit that "per se" there is no illegitimacy in basing (partly or entirely) a hypothesis on another one, although of course the strength of the second hypothesis will be dependent on the previous one's. We note that this seems a well-accepted practice in evolutionary biology studies, where past events cannot be demonstrated and are thus inherently hypothetical (please see also above in the response to Reviewer's comments of the "INTRACELLULARITY LATE" section). The most obvious example is any hypothesis on evolution of characters based on an inferred species tree, which is itself an evolutionary history hypothesis based on extant sequence data.

REVIEWERS' COMMENTS

Reviewer #1 (Remarks to the Author):

The authors have addressed my most important concerns in a satisfying manner.

The only comment I have left pertains to the discussion on the intracellular late hypothesis. The current state of the Discussion is in principle good enough for publication, but if the authors wish to clarify it a bit more after my feedback (see below), I encourage them to do so.

If I understand their rationale correctly, they find that the presence absence patterns of genes encoding proteins associated with glycolysis, gluconeogenesis, pentose-phosphate, Krebs cycle, electron transport chain, synthesis of cofactors, lipids, peptidoglycan, lipopolysaccharide, polyhydroxyalkanoate granules, but also of flagella, chemotaxis and T2SS/type IV pili in context of the Rickettsiales species tree are suggestive of gene content reductions that happened independently in multiple Rickettsiales lineages. This is taken as another line of support for independent origins of intracellularity within the Rickettsiales, because these independent reductions are indicative of independent transitions from facultative to obligate host associated lifestyles, and as noted before, independent origins of obligate host association automatically means independent origins of intracellularity. I think the link between independent gene losses and support for independent origins of intracellularity, that is, that the gene losses are signs of transitions towards obligate host associations, should be made more clear in these discussions.

Answer to Reviewer's comments

Reviewer #1 (Remarks to the Author):

The authors have addressed my most important concerns in a satisfying manner.

The only comment I have left pertains to the discussion on the intracellularity late hypothesis. The current state of the Discussion is in principle good enough for publication, but if the authors wish to clarify it a bit more after my feedback (see below), I encourage them to do so.

If I understand their rationale correctly, they find that the presence absence patterns of genes encoding proteins associated with glycolysis, gluconeogenesis, pentose-phosphate, Krebs cycle, electron transport chain, synthesis of cofactors, lipids, peptidoglycan, lipopolysaccharide, polyhydroxyalkanoate granules, but also of flagella, chemotaxis and T2SS/type IV pili in context of the Rickettsiales species tree are suggestive of gene content reductions that happened independently in multiple Rickettsiales lineages. This is taken as another line of support for independent origins of intracellularity within the Rickettsiales, because these independent reductions are indicative of independent transitions from facultative to obligate host associated lifestyles, and as noted before, independent origins of obligate host association automatically means independent origins of intracellularity. I think the link between independent gene losses and support for independent origins of intracellularity, that is, that the gene losses are signs of transitions towards obligate host associations, should be made more clear in these discussions.

We thank the Reviewer for the additional feedback. The manuscript was modified accordingly (lines 449-454 of the track-mode manuscript):

“Thus, in contrast with the more traditional views^{10,20,24}, our analyses provide a clear indication that processes of pathway reduction/loss have not taken place just once in *Rickettsiales*, but instead occurred (and are still possibly occurring) multiple times independently in different crown *Rickettsiales* lineages, also in relation with the host features. This hints towards an independent origin of obligate host-associations, and thereby intracellularity, among the *Rickettsiales* (see below section “Evolutionary trajectories of the interactions”).”